# Retrieval of UV-Visible aerosol absorption using AERONET and OMI-MODIS synergy: Spatial and temporal variability across major aerosol environments

Vinay Kayetha[1,2], Omar Torres[2], Hiren Jethva[2,3]

[1]Science Systems and Applications Inc., Lanham, Maryland-20706, USA
[2]NASA Goddard Space Flight Center, Greenbelt, Maryland-20771, USA
[3]Universities Space Research Association, Columbia, Maryland-21046, USA

*Correspondence to*: Vinay Kayetha (vinay.k.kayetha@nasa.gov)

**Abstract.** Measuring spectral aerosol absorption remains a challenging task in aerosol studies, especially in the UV region, where ground and airborne measurements are sparse. In this paper, we introduce an algorithm that synergizes ground measurements with satellite observations for the derivation of spectral single scattering albedo (SSA, $\omega_o$) of aerosols in the UV to visible wavelength range (340-670 nm). The approach consists in explaining satellite measured near-UV radiances (340, 354, 388 nm) by the Ozone Monitoring Instrument (OMI), and visible radiances (466, 646 nm) by the MODerate Imaging Spectrometer (MODIS), given the collocated ground-based Aerosol Robotic Network (AERONET) measurements of total column extinction aerosol optical depth (AOD, $\tau$), in terms of retrieved total column wavelength dependent SSA using radiative transfer calculations. Required information on aerosol particle size distribution is adopted from AERONET-based aerosol type-dependent seasonal climatologies specifically developed for this project. The inversion procedure is applied to about 110 AERONET sites distributed worldwide, for which continuous, long-term AERONET measurements are available. Using the derived data set, we present seasonal and regional climatology of $\omega_o(\lambda)$ for carbonaceous, dust, and urban/industrial aerosols. The resulting UV-Visible spectral dependence of $\omega_o$ obtained for these three major aerosol types is found to be both qualitatively and quantitatively consistent with independent measurements reported in the literature. A comparison to standard AERONET SSA product at 440 nm shows absolute differences within 0.03 (0.05) for 40% (65%) of the compared observations. The derived aerosol $\omega_o(\lambda)$ data set provides a valuable addition to the existing aerosol absorption record from AERONET by extending it to the near-UV region. Furthermore, SSA retrievals from our method at visible wavelengths and around satellite overpass time also complements the equivalent inversion available during early morning/late afternoon from AERONET. In addition to improving our understanding of spectral aerosol absorption properties, the combined

UV-Visible data set, also offers wavelength-dependent dynamic aerosol absorption models for use in the satellite-based aerosol retrieval algorithms.

# 1 Introduction

Through scattering and absorption of solar radiation atmospheric aerosols play a significant role in the radiation balance of the Earth's climate system. The ratio of the amount of the light scattering to the total extinction referred to as single scattering albedo (SSA, $\omega_o$). It is a fundamental variable used to gauge the absorbing nature of aerosols. Mie theory indicates the value of $\omega_o$ is 1.0 for purely scattering aerosols, and less than one towards zero for the increasingly absorbing nature of aerosols. Studies show that the estimates of net aerosol radiative forcing is sensitive to the aerosol $\omega_o$, and small changes to it could potentially alter the forcing on atmosphere (Chyacutelek and Coakley, 1974; Hansen et al., 1997). General circulation models are often fed with essential aerosol properties to estimate the aerosol radiative forcing effects on the atmosphere. These properties include aerosol optical depth (AOD, $\tau$), complex refractive index, and scattering phase function. Here, the knowledge on spectral dependence of such properties is crucial in quantifying the overall effects of aerosols. For example, absorbing aerosols can lead up to a 50% decrease in the near-UV solar irradiance compared to the similar load of only scattering aerosols in the atmosphere (Bais et al., 2005). A report by Intergovernmental Panel on Climate Change (IPCC) suggests that the lack of data on spectral aerosol absorption is one of the major contributors leading to significant uncertainties in quantifying the net aerosol radiative effects on the Earth's climate (IPCC, 2013).

Developments in ground-based and satellite aerosol retrieval techniques have greatly improved our understanding of atmospheric aerosols over the last two decades. However, knowledge on spectral aerosol absorption properties is limited due to difficulties in measurements (e.g., Heintzenberg et al., 1997) and larger uncertainties in remote sensing retrievals (e.g,. Dubovik et al., 2000). Direct measurements of aerosol absorption can be obtained by using instruments that measure aerosol scattering and extinction coefficients. Such measurements are limited to discrete wavelengths and associated with fewer ground stations, laboratory measurements, or airborne field campaigns. In addition, in situ techniques often require making corrections of measurements to overcome instrumental challenges (e.g., Weingartner et al., 2003; Virkkula et al., 2005; Collaud Coen et al., 2010). Aerosol absorption can also be inferred from the combined sky radiance and extinction measurements that rely on fitting ground observations to radiative transfer calculations (Nakajima et al., 1996; Dubovik et al., 1998; Cattrall et al., 2003). The accuracy of the aerosol absorption retrieval through ground-based remote sensing techniques primarily relies on the instrument calibration, whereas ancillary information such as the characterization of surface reflectivity has only secondary effect on the overall accuracy of the retrievals. Detailed reviews of

measurements and techniques to retrieve aerosol absorption are available in the literature (e.g., Clarke et al., 1967; Bond and Bergstrom, 2006; Moosmüller et al., 2009). Among ground-based sensors, AERONET (Aerosol Robotic Network) provides the longest aerosol absorption record at four discrete wavelengths from the visible

(Vis) to near Infrared (NIR) spectral region over many sites distributed worldwide. Known limitation of the currently available AERONET inversion product (Version 3) is the lack of single scattering albedo at near-UV wavelengths, and the aerosol load threshold ($\tau_{440} > 0.4$) required to obtain reliable absorption in Vis-NIR spectrum. Like AERONET, a network of ground-based radiation measurement network with sites in Asia and Europe referred as SKYNET (Sky Radiometer Network) provides aerosol optical depth and single scattering

albedo in the near UV-NIR spectrum (Nakajima et al., 2007). However, the accuracy of the SKYNET aerosol absorption product is affected by temporally constant and spectrally invariant surface reflectance used in the inversion procedure (Jethva and Torres, 2019). These limitations restrict our ability of complete characterization of aerosol absorption as a function of both wavelength and aerosol load.

For a few decades now, satellite remote sensing has been used as an essential tool to gain a global perspective of aerosols distribution in the atmosphere. The physical basis of satellite aerosol retrievals is that under cloud-free conditions after accounting for Rayleigh (molecular) scattering, gaseous absorption effects, and surface reflectance, the upwelling top-of-the-atmosphere (TOA) reflectance is a function of aerosol optical depth, particle size and composition (i.e., complex refractive index). Mathematically, for a cloud-free atmosphere overlying an

Lambertian surface the upwelling TOA reflectance ($L_{TOA}$) received by a nadir viewing satellite can be expressed in normalized units as (Chandrasekhar, 1960):

$$L_{TOA}(\theta, \varphi, \lambda) = L_0(\theta, \varphi, \lambda) + \frac{\rho(\lambda) \cdot T(\tau, \theta, \lambda)}{(1 - s(\lambda) \cdot \rho(\lambda))} \tag{1}$$

Where $\theta$, $\varphi$, and $\lambda$ are the zenith, azimuthal angles of the direction of propagation and wavelength of light,

$L_0$ is the atmospheric path radiance,

$\tau$ is the optical thickness of the atmosphere,

$\rho$ is the surface reflectivity,

T is the total direct and diffuse transmittance of the light in the atmosphere, and

$s(\lambda)$ is the spherical albedo of the atmosphere when it is illuminated from below.

The first and second terms on the right side of equation (1) represents the atmospheric path radiance and the amount of light that is reflected to the sensor after encountering the surface, respectively. The satellite measured

TOA reflectances are sensitive to both $\tau$ and $\omega_o$, in addition to the surface reflectance. Therefore, separating the contributions of atmosphere and surface is of utmost importance to retrieve aerosols from satellite measurements. For satellites with single-view measurements, aerosol retrieval algorithms rely on prior assumptions on particle sizes (scattering phase function) and $\omega_o$ to retrieve $\tau$. On the other hand, several efforts have been made to estimate aerosol $\omega_o$ from direct satellite measurements at visible wavelengths (e.g., Kaufman, 1987; Kaufman et al., 2002; Satheesh and Srinivasan, 2005; Zhu et al., 2011) and near-UV wavelengths (Torres et al., 1998, 2007, 2013). However, the variety of natural surface types, choice of wavelengths, and aerosol models pose limitations on such techniques. In terms of wavelength, enhanced molecular scattering in the near-UV region acts as a strong attenuating background below aerosol layer and helps identify absorbing aerosols. However, to retrieve aerosol absorption using near-UV measurements, quantitative information on aerosol layer height (ALH) is required. Existing satellite aerosol retrieval techniques that rely on observations in the visible spectrum assume a temporally constant value of $\omega_o$ that varies regionally (Remer et al., 2005; Levy et al., 2007), and for a few algorithms, it is still assumed wavelength-independent (Hsu et al., 2013). A review of commonly used satellite aerosol products singled out aerosol absorption as an inherent problem common to all sensors (Li et al., 2009). Studies using the evolving ground-based aerosol record provides evidence that satellite retrieved $\tau$ can lead to large biases if the assumed aerosol imaginary index, which drives $\omega_o$, is wavelength-independent (Jethva and Torres, 2011), and seasonally invariant (Lyapustin et al., 2011; Eck et al., 2013). These studies highlight the importance of using wavelength-dependent aerosol $\omega_o$ and accounting for its spatial and temporal variability in the retrieval of satellite aerosol products.

In the past, few studies used both ground and satellite measurements to retrieve aerosol absorption properties. Li et al., (1999) used visible band radiances from AVHRR (Advanced Very High Resolution Radiometer) and insitu measured $\tau$ during SCAR-B (Smoke Clouds Aerosol Radiation–Brazil) experiment to derive absorption from biomass burning aerosols. Sinyuk et al., (2003) used UV-radiances from TOMS (Total Ozone Monitoring Station) and aerosol extinction from AERONET to derive the imaginary refractive index of dust particles over a few stations in the Saharan region. Lee et al., (2007) estimated the aerosol SSA across a few stations over China using combined ground and satellite (MODIS–Moderate resolution imaging spectrometer) measurements at visible wavelengths. Nonetheless, these studies are limited and do not provide a comprehensive, long-term characterization of absorbing aerosols in the UV-Visible spectral region.

The objectives of the present work are to derive columnar aerosol $\omega_o(\lambda)$, and its spectral dependence in the UV-Visible part of the spectrum. The proposed inversion procedure makes use of AERONET measured wavelength dependent $\tau$ and retrieved particle size distribution in conjunction with satellite measured radiances at UV and visible wavelengths by A-train constellation sensors Aqua-MODIS and Aura-OMI (Ozone Monitoring Instrument). The near-simultaneous measurements from these sensors provide an excellent opportunity to combine satellite and ground measurements during the overpass times (local time, ~13:30 hrs) over the AERONET sites.

The organization of the remaining paper is as follows: section 2 describes the ground-based and satellite measurements used in this work; section 3 describes the methodology adopted to derive aerosol $\omega_o(\lambda)$; section 4 discusses results of sensitivity analysis aimed to estimate the expected accuracy of the proposed aerosol absorption retrievals; section 5 presents a comparison of the resulting SSA product from this work to the AERONET aerosol absorption product; section 6 presents the seasonal variability in regional aerosol $\omega_o(\lambda)$ derived for sites across major aerosol environments worldwide; section 7 provides a discussion of the regional variability of the UV-Vis aerosol absorption product derived in this work. Finally, section 8 provides a summary of the work, along with the key findings and outlook for further studies.

## 2 Data sets

The details of ground-based and satellite data sets used in this work are provided in Table 1. Our usage of satellite data is strictly limited to the measured TOA reflectances by the Aqua-MODIS and Aura-OMI sensors, the associated viewing geometry, and other ancillary information such as quality flags, and the OMI near-UV Aerosol Index (UVAI).

### 2.1 AERONET

AERONET employs an automatic sun-tracking photometer (CIMEL Electronique CE-318) to measure sun and sky radiances (Holben et al., 1998). The direct sun measurements are made with a 1.2° full field of view at nine nominal wavelengths of 340, 380, 440, 500, 675, 870, 940, 1020 and 1640 nm typically for every ~5 to 15 minutes interval. Columnar extinction $\tau$ is computed from these measurements for all wavelengths except for the 940 nm, which is used to retrieve water vapour amounts. The extinction $\tau$ obtained from these measurements has an estimated uncertainty of ±0.01 (±0.02) at the visible (near-UV) wavelengths, primarily due to calibration

uncertainty (Eck et al., 1999). The currently available AERONET Version 3 Level 2 AOD product uses improved cloud screening and quality checks to provide reliable data to the user community (Giles et al., 2019). In addition to the direct sun measurements, the photometer also measures multi-angular diffuse sky radiances along the almucantar plane (plus hybrid scans to lower solar zenith angles) at four distinct wavelengths from visible to the near-Infrared spectrum (440, 675, 870 and 1020 nm) with near-hourly frequency. In recent years the newer model

of instruments also include sky radiance measurements at 380, 500 and 1640 nm (Sinyuk et al., 2020). An inversion procedure that uses both direct sun and angular sky radiances together is implemented to derive aerosol particle size distribution and complex refractive indices (Dubovik and King, 2000; Dubovik et al., 2006). The uncertainty in the derived spectral aerosol SSA provided by the AERONET inversion Level 2 product is estimated to be $\pm 0.03$ for $\tau_{440} > 0.4$ (Dubovik et al., 2000). Since 2018, the release of Version 3 inversion product

implements several changes to the traditional AERONET aerosol absorption retrievals. A complete description of the changes implemented in Version 3 inversion products along with the updated uncertainty estimates are available in Sinyuk et al., (2020). It should be noted here that in the currently available AERONET inversion products the shortest wavelength of aerosol SSA is 440 nm. In this work, we use AERONET Version 2 inversion product for constructing a representative aerosol model for the associated sites. For the AOD inputs to our

retrieval algorithm and for the comparison of SSA we use the latest Version 3 products. Figure 1 shows the location of the total 110 AERONET sites selected in this work for which long-term (> 7 years) quality assured measurements are available.

**2.2 OMI**

Launched in July 2004, the OMI on board NASA's EOS Aura satellite is a nadir-viewing hyper-spectral imaging

radiometer (Levelt et al., 2006). OMI measures the TOA radiances in the wavelength range 270-500 nm with a ground pixel spatial resolution of 13 km x 24 km at nadir. OMI achieves daily global coverage in 14-15 orbits with a swath of 2600 km scanning the entire earth's surface. In this work, we use OMI radiances (340, 354, and 388 nm) provided in the in-house product OMLERWAVE and publicly accessible OMI near UV OMAERUV Level 2 aerosol product (Version 1.8.9.1, Torres et al., 2018). The OMLERWAVE product reports radiances and

Lambertian equivalent reflectivity (LER) at several discrete wavelengths in the near-UV and visible parts of the spectrum. Additionally, we also use ancillary information on the quality of pixel (cloud contamination, land/sea mask, etc.), UVAI, LER, surface pressure and ALH data set used in the operational OMAERUV product. Since mid-2007, OMI suffers from an external obstruction that affects the quality of radiance measurements in a few rows (cross-track pixels). This is referred to as 'row anomaly' that restricts the current usage of OMI observations

for the scientific purpose to about half in a total of 60 cross-track rows (Schenkeveld et al., 2017). The impact of reduced spatial coverage as a result of the row anomaly on the OMAERUV aerosol product is discussed by Torres et al., (2018).

## 2.3 MODIS

The MODIS sensor on board NASA's EOS Aqua and Terra satellites are nadir-viewing, multi-spectral radiometer. MODIS measures the TOA radiances in 36 wavelength bands ranging from 0.41-14.23 μm with a ground pixel spatial resolution between 250-1000 m (King et al., 1992). MODIS scans the earth's surface with a 2300 km wide swath to provide near-global coverage daily. In this work, we use Aqua-MODIS radiances (at 466 and 646 nm) provided in the 10-km aerosol product (MYD04_L2) from the Deep Blue (DB) aerosol algorithm.

This aerosol product provides cloud-free radiances and ancillary information on the terrain height/pressure, quality of pixel and estimated cloud fraction (Hsu et al., 2013). For surface characterization in visible wavelengths we use MODIS MAIAC (Multi-Angle Implementation of Atmospheric Correction) MCD19A1 daily 1-km sinusoidal gridded spectral BRF (Bidirectional reflectance function) or surface reflectance product (Lyapustin and Wang, 2018).

## 3 Methodology


A schematic flowchart shown in Figure 2 illustrates the method adopted in this work to derive wavelength-dependent aerosol single scattering albedo.

### 3.1 Computation of Site-specific Seasonal Look-up tables of TOA reflectances

To start, we compile a seasonal climatology of aerosol particle size distributions and real part of the refractive 200 index (440 nm) for the entire τ range from the AERONET Level-2, Version 2 inversion product for each site considered in the study (see Figure 1). Here, we assume that the spectral variability of the real part of the aerosol refractive index through UV-Visible is minimal and, therefore, values derived at 440 nm were assumed to be wavelength-independent across the UV-Visible spectral range considered in this study. The resulting site-specific climatologies of aerosol size distribution are fed to a radiative transfer model (RTM) to generate look up table 205 (LUT's) of outgoing TOA reflectances at 340, 354, 388, 466, and 646 nm with varying nodal points of satellite-sun geometry (i.e., SZA–solar zenith angle at 0º, 20º, 40º, 60º, 66º, 72º and 80º; VZA–viewing zenith angle at 2º

interval from 0–88º; RAA–relative azimuth angle at 15º interval from 0–180º). Reflectance LUTs are created for two values of surface pressure (1013.25 and 600 mb), seven values of $\tau$ (0.0, 0.1, 0.5, 1.0, 2.5, 4.0 and 6.0), five nodal points on ALH (0, 1.5, 3.0, 6.0 and 10.0 km) for the referenced surface pressure nodes, and eight values of imaginary component of the refractive index (0.000, 0.008, 0.016, 0.024, 0.032, 0.040, 0.048 and 0.056). The aerosol profiles used in the RTM follow a Gaussian distribution centred around the respective modes of ALH. We assumed a total column ozone of 275 Dobson Unit (DU) in the RTM to account for ozone absorption. The Gauss-Seidel radiative transfer code used for the radiative transfer simulations accounts for gaseous absorption ($NO_2$ is not included), molecular and aerosol multiple scattering (Herman and Browning, 1965). Thus, a database of AERONET site-specific seasonal LUT of reflectances for the aerosols observed over each site in the study is created. Figure 3 shows an example of the calculated net aerosol reflectance at the TOA for selected sun-satellite geometry (SZA=20º, VZA=40º, RAA=130º) and varying values of $\tau$ and $\omega_o$ from our LUT developed for the GSFC site (38.92º N, 76.84º W). These results illustrate that for a given satellite-sun geometry, observed radiance, and assumed LUT, multiple combinations of $\tau$ and $\omega_o$ can explain the satellite measurements. In addition, it is noted that the net reflectances are mostly invariant at low optical depths (~0.1) regardless of variations in SSA for all wavelengths. This is a typical scenario for the LUT approach to derive aerosol properties, suggesting retrieval of absorption is likely not reliable at low optical depths. These results also illustrate the *critical reflectance concept* (Kaufman, 1987), as a particular upwelling reflectance value (also associated with a particular value of surface reflectance) at which there is no sensitivity to aerosol optical depth and, therefore, theoretically suitable for the retrieval of aerosol absorption from satellite observations. Nonetheless, to derive the best-fit or unique solution of $\omega_o$ from satellite measurements, an accurate characterization of $\tau$, cloud-free radiances, and surface reflectances are required.

The site-specific LUTs developed here assume spherical particle shapes (Mie theory) for carbonaceous and urban aerosols. However, mineral dust particles are assumed non-spherical and modelled as randomly oriented spheroids (Dubovik et al., 2006; Torres et al., 2018). To account for the non-sphericity of dust particles, a unified dust model LUT is created using particle sizes from selected AERONET sites over Sahara and Middle East region that include: Saada, SEDE_BOKER, Solar_Village, and Tamanrasset_INM. These sites were selected based on the observed prevailing dust aerosol type. The particle sizes and real refractive index obtained at these sites are used with a pre-computed set of kernels that assume a spheroidal shape with a fixed distribution of axis ratio to produce phase function (Dubovik et al., 2006). The obtained phase matrix elements are input to the RTM

to create reflectance LUT's. The process of acquiring a non-spherical unified model LUT is necessary to save a considerable computational time, which otherwise would require to create another set of site-specific LUTs.

## 3.2 Collocation of satellite and ground measurements

We use satellite measurements located within the 50 km radius of each AERONET site. In essence, we treat the overlying atmospheric aerosols within a 50 km radius of the site as a representative of the AERONET measured $\tau$. We look for valid AERONET $\tau$ measurements within $\pm 2$ hours of satellite overpass and assign the $\tau$ closest in time to all the ground pixels. It should be noted here that for the OMI sensor, the OMAERUV product provides cloud-free radiances (340, 354 and 388 nm) in the native pixel resolution of 13 km x 24 km, while the MODIS

sensor DB-product provides cloud-free radiances (466 and 646 nm) at 10 km x 10 km resolution.

## 3.3 Retrieval of aerosol $\omega_o(\lambda)$

The proposed technique to derive aerosol absorption involves obtaining (a) the AERONET AOD at OMI and MODIS wavelengths, (b) aerosol type, (c) aerosol centroid layer height, (d) surface pressure, (e) surface reflectance, and (f) best quality assured MODIS and OMI measured cloud free-TOA reflectances.


### 3.3.1 AERONET AOD at sensor wavelengths

The AODs at the nominal wavelengths measured by AERONET along with the computed Extinction Ångström Exponent (EAE, $\alpha$) for several wavelength ranges (340-440, 380-500, 440-675, 440-875, etc.) are available in the AERONET AOD product. We derive the $\tau$ at our interest of satellite wavelengths using the closest available

measurement and $\alpha$ through the power-law approximation (Ångström, 1929) as shown in equation 2. For OMI wavelengths, the AERONET 340 and 380 nm measurements are readily available while $\tau_{354}$ is obtained with $\lambda_{Ref}$ = 380 nm and $\alpha_{\lambda\_Ref} = \alpha_{340-440}$. For the few sites with older models of AERONET sunphotometer that does not have direct sun measurements at 340, and 380 nm (for example Banizoumbou, Avignon, etc.), we use $\lambda_{Ref}$ = 440 nm and $\alpha_{\lambda\_Ref} = \alpha_{440-675}$. Similarly, for MODIS wavelengths the $\tau$ at 466 and 646 nm are obtained using $\lambda_{Ref}$ = 440

nm and $\alpha_{\lambda\_Ref} = \alpha_{440-675}$.

$$\tau_\lambda = \tau_{\lambda_{Ref}} \left(\frac{\lambda}{\lambda_{Ref}}\right)^{-\alpha_{\lambda\_\lambda Ref}} \tag{2}$$

### 3.3.2 Aerosol type and ALH

Aerosol type information is essential to derive absorption properties. We use a combination of Extinction

Ångström Exponent ($\alpha_{440\text{-}870}$) derived from AERONET and UVAI from OMAERUV product to categorize the observed aerosols into three basic types – dust, carbonaceous, and urban/industrial. Initially, our algorithm uses $\alpha_{440\text{-}870}$ to identify the aerosols as coarse ($\alpha_{440\text{-}870} \leq 0.2$) and fine ($\alpha_{440\text{-}870} \geq 1.2$) mode dominated particles. Threshold $\alpha_{440\text{-}870}$ of 0.2 chosen for coarse mode particles unambiguously identifies dust aerosols. However, the sample of fine mode particles consists of both carbonaceous and urban types of aerosols that has wide range of

absorption depending on the source of emissions. The near UV aerosol index is an excellent indicator to identify the presence of absorbing aerosols. Threshold UVAI value adopted from OMAERUV algorithm is used to separate carbonaceous (UVAI $\geq 0.8$) and urban (UVAI $< 0.8$) aerosols, respectively. Based on extensive tests on the OMI signal strength on all surface types it is determined that a minimum UVAI of 0.8 is required to identify absorbing aerosols (Torres et al., 2007, 2013). Although UVAI is an excellent indicator to identify the presence

of absorbing aerosols, the large OMI footprint (13 km x 24 km) and sub-pixel contamination may occasionally produce an underestimated UVAI, resulting in the misidentification of the observed aerosols as urban type. To derive aerosol absorption for dust, LUT for non-spherical particle shape is selected, while for carbonaceous and urban aerosols site-specific spherical LUTs are used. In addition, for the absorbing types of aerosols i.e., carbonaceous and dust, we choose an estimate of ALH from a CALIOP (Cloud-Aerosol Lidar with Orthogonal

Polarization) based ALH climatology product of absorbing aerosols (Torres et al., 2013). Prescribed uncertainty in the monthly climatology of ALH derived from joint OMI-CALIOP product primarily due to limited sampling of the CALIOP lidar overpasses in the 1° grid (16-day overpass cycle) and day-to-day variability of ALH is expected to be within ±1 km (Torres et al., 2013). For urban aerosols, an exponential aerosol profile peaking at the surface is employed to perform the inversion procedure.


### 3.3.3 Surface reflectance and pressure

For the surface characterization at OMI wavelengths, we use a near-UV surface albedo database at quarter degree grid resolution provided in the OMAERUV product (Torres et al., 2007). The near-UV surface albedo employed by OMAERUV is derived from minimum lambertian equivalent reflectance obtained from available long-term

measurements. The uncertainty in the near-UV surface albedo from these measurements is expected to be within ±0.01 (Torres et al., 2018). At MODIS wavelengths, surface reflectance or BRF provided by MAIAC MCD19A1 product is used. The MAIAC MCD19A1 provides spectral surface BRF over cloud-free and clear-to-moderately turbid atmospheric conditions ($\tau_{466} < 1.5$) for solar zenith angles below 80°. The measurement-based uncertainty in MCD19A1 BRF at visible wavelengths is reported to be in the range of 0.002–0.003 for the combined sources

of errors including uncertainties from gridding, cloud detection, and aerosol model properties (Lyapustin et al., 2018). Additionally, the surface or terrain pressure reported in the OMAERUV, and terrain height (converted to pressure) reported in the MODIS aerosol products are used in our SSA retrievals.

Our retrieval technique gathers all the above-mentioned required inputs including the best quality assured cloud-free radiances reported in OMAERUV (Quality flag = 0) and MODIS-DB (Quality flag = 3) products to perform an inversion for each wavelength independently. The inversion procedure interpolates the LUT radiances linearly for the prescribed satellite-sun geometry, ALH, $\tau$, and logarithmically over the surface pressure nodes. The obtained LUT radiances as a function imaginary refractive index are then fitted with the satellite measured radiances to derive aerosol $\omega_o(\lambda)$.

### 3.3.4 Correction accounting for NO$_2$ gas absorption

Among the trace gases present in the lower atmosphere, ozone (O$_3$) and nitrogen dioxide (NO$_2$) have strong absorption lines in the UV-Visible spectral region of our interest (340-646 nm). As mentioned in section 3.1, the RT model employed in this work accounts for H$_2$O and O$_3$ amounts leaving the simulated radiances unaccounted for NO$_2$ absorption. In general, NO$_2$ amounts present in the lower atmosphere are primarily formed through vehicular and industrial emissions in addition to the small amounts of NO$_2$ through biomass burning emissions. Therefore, over the regions with high vehicular and industrial emissions the retrievals of aerosol SSA could be biased since our simulated radiances do not include the contribution from NO$_2$ amounts. Krotkov et al (2005) investigated the effect of NO$_2$ amounts on the retrieval of aerosol SSA using the measured NO$_2$, aerosol optical depth, and radiance measurements from the UV-Multifilter Rotating Shadowband Radiometer (UV-MFRSR). This study demonstrates that at UV wavelengths the aerosol SSA could be biased low due to unaccounted NO$_2$ gas absorption in the measured radiances and provides a correction (equation 3) to obtain the corrected aerosol single scattering albedo.

$$\omega_a = \omega(no\ NO_2\ corr).\left[1 + \frac{\tau_{NO2}}{\tau_a}\right] \tag{3}$$

Where, $\omega_a$ is the corrected aerosol SSA,

$\omega$ is the aerosol SSA unaccounted for NO$_2$ absorption,

$\tau_{NO2}$ is the optical depth of columnar NO$_2$ amounts, and

$\tau_a$ is the aerosol optical depth after correcting for Rayleigh, and trace gases including NO$_2$.

For the SSA retrievals in this work, we apply $NO_2$ gas correction as a final step in our retrieval algorithm. We use $NO_2$ concentration provided in the AERONET AOD product (determined from monthly climatology of the total column $NO_2$ retrievals from OMI measurements gridded at $0.25°$ x $0.25°$ spatial resolution) and absorption coefficients from Vanadele et al (1998) to determine $\tau_{NO2}$. The obtained spectral $\tau_{NO2}$ is used to estimate the corrected aerosol SSA (shown in equation 3) as demonstrated by Krotkov et al (2005).


### 3.3.5 Illustration of retrieved SSA

    Figure 4 shows the retrieved aerosol SSA over the GSFC site as a function of AERONET measured $\tau$. Located in the vicinity of a metropolitan area, the prevailing aerosols over the GSFC site are the urban industrial types that are relatively more scattering in nature. In general, retrieved SSA increases with aerosol loading, except for a

small decrease at large AOD's at 466 and 646 nm. Particularly notable is the high variability of retrieved SSA in most $\tau$ bins for the visible wavelengths (i.e., MODIS bands). This is due to the diminishing aerosol signal strength for weakly absorbing urban type aerosols at lower aerosol loading in the visible spectrum, where the measured TOA radiances are dominantly contributed by the underlying surface, notably at 646 nm. The mean aerosol SSA retrieved at the GSFC site for observations with $\tau_{440} > 0.4$ at 340, 354, 388, 466 and 646 nm are

0.95, 0.96, 0.96, 0.95 and 0.93, respectively. These results agrees well with the values reported for GSFC site using AERONET products at 440 and 675 nm as 0.96 and 0.95, respectively (Giles et al., 2012). Also shown in Figure 4 is the number of collocated observations that were used in the inversion and the percent of observations for which SSA is retrieved. For about 12 years of the satellite and ground collocated observations used here, the number of observations from OMI is less than MODIS observations. The difference in the number of collocated

observations stems partly from the OMI row-anomaly, cloud contamination, and the coarser pixel resolution. The percent of SSA retrieved observations varies widely even within the corresponding sensor wavelengths (OMI: 340, 354, 388 nm and MODIS: 466, 646 nm). At times depending on the surface reflectance, the computed net aerosol reflectance might exceed the LUT limits and produce SSA values above one or less than the maximum absorption in the LUT, typically referred to as out-of-bounds retrieval. We avoid this by constraining our

inversion procedure within the LUT limits and do not allow for any extrapolation of the radiances. However, this leads to the unequal number of retrieved observations within and between the sensor wavelengths. In other words, for a given observation within the OMI or MODIS sensor, it is possible to have aerosol SSA retrieved at one wavelength and no retrieval (out-of-bounds) at other wavelengths.

To examine the spectral dependence of aerosol absorption, we created a subset of the data that includes observations for which aerosol SSA is retrieved for all the corresponding sensor wavelengths simultaneously (OMI: 340, 354, 388 nm and MODIS: 466, 646 nm) on any given day. This step reduces the sample size drastically but eliminates the need for making prior assumptions on the wavelength dependence of aerosol absorption to fill those gaps. The obtained subset of aerosol SSA in the UV-Visible range is used to compute the resulting spectral dependence of aerosol absorption of the prevailing aerosols over the corresponding AERONET sites in terms of the aerosol Absorption Ångström Exponent (AAE), a measure of the spectral dependence of aerosol absorption optical depth (Bond, 2001) using a power-law approximation, analogous to the Extinction Ångström Exponent (van de Hulst, 1957). The spectral dependence of aerosol absorption AAE, is defined as the slope of aerosol absorption optical depth with wavelengths on a log-log scale. The aerosol absorption optical depth $\tau_{abs}(\lambda)$ is derived as shown in equation (4):

$$\tau_{abs}(\lambda) = \left(1 - \omega_0(\lambda)\right) \cdot \tau_{ext}(\lambda) \tag{4}$$

from which the AAE for wavelength range $\lambda_1$, $\lambda_2$ is calculated as shown in equation (4).

$$AAE(\lambda_1, \lambda_2) = -\frac{\ln(\tau_{abs}(\lambda_1)/\tau_{abs}(\lambda_2))}{\ln(\lambda_1/\lambda_2)} \tag{5}$$

The results presented hereafter include only a data subset that meets the following three conditions: (a) SSA retrievals are available for all five wavelengths on a given day, (b) $\tau_{440} > 0.4$, to ensure reliable accuracy spanning through UV-Visible wavelengths, and (c) there are at least 5 days of observations available per season per aerosol type.

**4 SSA retrieval sensitivity analysis**

The proposed inversion procedure to derive spectral aerosol absorption from the combined ground and satellite measurements is susceptible to several systematic and random errors. These error sources include uncertainties in the following input parameters: (a) aerosol extinction measurements, (b) estimation of particle sizes (volume mean radius-VMR), (c) real part of refractive index (RRI), (d) calibration of satellite measured TOA radiances, (e) sub-pixel cloud contamination, (f) $O_3$ and $NO_2$ gaseous absorption in the RTM, (g) surface reflectance, (h) ALH, (i) surface pressure, and (j) variability in AOD for the satellite observations within ±2 hours and 50 km radius of the AERONET measurement. The retrieved aerosol absorption from our inversion procedure could be affected by all these sources of uncertainties. Errors associated with surface reflectance, ALH, and cloud contaminations on the satellite retrieved optical depths are well documented in the literature (e.g., Fraser and

Kaufman, 1985; Torres et al., 1998; Jethva et al., 2014). In summary, it is known that an: (i) overestimation (underestimation) of surface reflectance leads to lower (higher) aerosol SSA, (ii) overestimation (underestimation) of $\tau$ leads to lower (higher) aerosol SSA, (iii) overestimation (underestimation) of ALH produces higher (lower) aerosol SSA (significantly more pronounced in the UV than in visible wavelengths), and (iv) an increase in TOA reflectance due to sub-pixel cloud contamination produces higher aerosol SSA.

## 4.1 Estimation of theoretical errors in the retrieved aerosol $\omega_o(\lambda)$

Here, we conduct sensitivity tests for all sources of errors in the input variables to derive a theoretical estimate of the error percolated in the aerosol SSA retrieval due to uncertainties in the assumed values of these variables. To have a controlled setup, we performed tests for a representative fixed satellite-sun geometry (SZA=20°, VZA=20°, RAA=130°) over the GSFC, Mongu, and Tamanrasset_INM. These sites were selected to represent three distinct aerosol types as well as surface conditions. We assume a fixed value of $\omega_o$ = 0.9 at 388 nm and aerosol load $\tau_{440}$ = 0.2, 0.3, and 0.4 as our references to estimate errors in the retrieved SSA. To derive corresponding spectral AODs at remaining wavelengths, we assume an $EAE_{340-646}$ of 1.9, 0.2, and 1.9 for carbonaceous, dust and urban aerosols, respectively. Similarly, spectral SSA at other wavelengths is estimated assuming an $AAE_{340-646}$ of 1.7, 2.5 and 0.9 for carbonaceous, dust and urban aerosols, respectively. We calculate the uncertainty of the derived spectral SSA for each aerosol type by perturbing, one at a time, the nominal values of the nine inputs parameters by an assumed or observationally known uncertainty. The absolute error is computed as the SSA obtained with altered input minus the assumed SSA. The combined uncertainty of the derived spectral SSA is given by square root of the summation of the squares of the errors associated with each parameter.

## 4.1.1 Theoretical errors in retrieved $\omega_o(\lambda)$ due to $\tau_{ext}$, VMR and RRI

Figure 5 shows the error analysis of the retrieved SSA as a function of wavelength and optical depth given a change in the input (a) $\tau_{ext}(\lambda)$, (b) volume mean radius of the particles, and (c) real part of refractive index. We perturb the input $\tau$ with an absolute value of ±0.02 for $\lambda$ < 400 nm and ±0.01 for $\lambda$ > 400 nm as prescribed by the AERONET AOD product. As shown in Fig 5a, AOD over-estimations result in SSA under-estimations whereas AOD underestimations yields SSA overestimations for all aerosol types over the considered spectral range. The magnitude of the SSA error ($\Delta\omega_o$) decreases with increasing AOD. It is noted that for all aerosol types an underestimation (overestimation) of AOD, the magnitude of $\Delta\omega_o$ is positive (negative) and increases with wavelength (340 nm to 388 nm, and 466 nm to 646 nm). The higher magnitudes of $\Delta\omega_o$ noted for visible

wavelengths are attributed to lower spectral AODs where the aerosol absorption signal is diminished for a stable retrieval, particularly notable for weakly absorbing urban aerosols. For the reference $\tau_{440} = 0.4$, perturbation of $\pm 0.02 + \tau$ at 340 nm yields an error $\Delta\omega_o$ within $\pm 0.002$, while a perturbation of $\pm 0.01 + \tau$ at 646 nm yields an error within $\pm 0.011$.

Uncertainties in assumed particle sizes are also expected to affect the retrieval of aerosol absorption. To estimate the error incurred in our SSA retrieval, we perturb the particle volume mean radius derived from all AOD observations by 20%. We chose $\Delta VMR = 20\%$ based on examination of seasonal climatology of particle sizes as a function of $\tau_{440}$ and the most frequently occurring $\tau_{440}$ bin. It is noted that an overestimation of particle radii produces higher aerosol SSA leading to positive $\Delta\omega_o$ and vice-versa for carbonaceous and urban aerosols. Spectrally the magnitude of $\Delta\omega_o$ is minimum in the UV and increases towards the visible wavelengths. The magnitude and spectral behavior of $\Delta\omega_o$ noted here suggests aerosol scattering primarily drives the particle size effect. For dust aerosols, the $\Delta\omega_o$ noted are quite small/negligible in UV wavelengths while remains invariant at visible spectral range. This owes to the size of dust particles that are much higher than the considered spectral range where extinction of radiation reaches maximum efficiency and remains less variant with additional increase in particle sizes. It should be noted that at the AODs considered ($\tau_{440} = 0.2$, 0.3, and 0.4), particle size assumptions here have only a small effect on the retrieved SSA for all aerosol types. However, the increase in particles sizes due to processes such as coagulation and condensation at higher AOD levels might likely add additional errors in the estimated $\Delta\omega_o$. For the reference $\tau_{440} = 0.4$, perturbation of $\pm 20\%$ VMR to all aerosol types yields an error $\Delta\omega_o$ within $\pm 0.018$, and $\pm 0.044$ at 340 nm and 646 nm, respectively.

Another aerosol intrinsic property input for our SSA retrieval algorithm obtained from AERONET inversion product is the real part of refractive index (RRI) – which primarily contributes to the magnitude of scattering. The prescribed uncertainty in aerosol RRI is estimated to be $\pm 0.04$ (Dubovik et al., 2000). Our results indicate that an overestimation of RRI produces lower aerosol SSA and vice-versa. The effect of aerosol RRI perturbation on retrieved SSA is noted to be higher in the UV-spectrum than in the visible. This is likely a result of strong competing effects from molecular scattering and aerosol absorption, while aerosol load adds an additional weak dependence. For the reference $\tau_{440} = 0.4$, perturbation of $\pm 0.04 + RRI$ yields an error $\Delta\omega_o$ within $\pm 0.009$ and $\pm 0.002$ at 340 nm and 646 nm, respectively.

### 4.1.2 Theoretical errors in retrieved $\omega_o(\lambda)$ due to OMI/MODIS calibration and cloud contamination

Figure 6 shows the error analysis of the retrieved SSA as a function of wavelength and optical depth given a change in the (a) TOA radiances due to sensor calibration, and (b) sub-pixel cloud contamination. The prescribed uncertainty in the TOA radiance measurements for OMI and MODIS sensors are expected to be ±1.8% (Schenkeveld et al., 2017) and ±1.9% (Guenther et al., 2002; Xiong et al., 2018), respectively. As expected, an overestimation of TOA radiances due to sensor calibration produces lower aerosol SSA and vice-versa. Errors in

the retrieved SSA due to uncertainties in the sensor calibration increase with decreasing aerosol optical depth. For the reference $\tau_{440} = 0.4$, perturbation of ±1.8% in TOA radiances at 340 nm yields an error $\Delta\omega_o$ within ±0.027, while a perturbation of ±1.9% in TOA radiances at 646 nm yields an error within ±0.037. To estimate the error incurred in the retrieval of aerosol SSA due to cloud contamination, we developed LUTs for each aerosol type assuming a cloud of optical thickness 0.5 in our RT simulations. These LUTs are used to compute the sensitivity

of aerosol SSA due to the presence of optically thin cloud layer in the atmosphere. Our results demonstrate that the assumed optically thin cloud ($\tau_{cloud} = 0.5$) produces an overestimation of TOA radiances leading to higher aerosol SSA. The effect of cloud contamination in TOA radiances is more pronounced in the visible than in UV spectrum. For observations with $\tau_{440} = 0.4$, the cloud contamination yields an error $\Delta\omega_o$ within ±0.020 and ±0.056 at 340 nm and 646 nm, respectively.

### 4.1.3 Theoretical errors in retrieved $\omega_o(\lambda)$ due to $O_3$ and $NO_2$ gaseous absorption

As described in section 1, for the retrieval of aerosol properties from satellite measured radiances it is important to separate the TOA radiance signal from the underlying surface and atmospheric constituents including trace gases. It should be noted that the RTM used in this work accounts only for the $H_2O$ and $O_3$ gaseous absorption. In

addition, we applied correction for the retrieved aerosol SSA to account for $NO_2$ absorption. Here we estimate the error incurred in our aerosol SSA retrievals due to uncertainties in the employed $O_3$ and $NO_2$ amounts available through AERONET AOD product. Based on the variability of $O_3$ and $NO_2$ columnar amounts and their uncertainties (not shown here) for the sites used in this work, a perturbation of ±50 DU and ±1 DU for $O_3$ and $NO_2$ respectively are chosen to estimate the error incurred in our retrievals. Our results, as shown in Figure 7,

indicate that overestimation of $O_3$ amounts by +50 DU produces lower SSA (higher absorption) leading to negative $\Delta\omega_o$, while underestimation of $O_3$ amounts produces higher SSA (lower absorption) leading to positive $\Delta\omega_o$. Similar to $O_3$ amounts, an overestimation of $NO_2$ amounts produces lower SSA yielding negative $\Delta\omega_o$ and vice-versa. The error in retrieved SSA due to ±50 DU of $O_3$ is noted to be within 0.008 at 340 nm and increases towards the visible spectrum at 646 nm ranging 0.009-0.018 for observations with $\tau_{440}$ up to 0.4. However, the

error in retrieved SSA due to underestimation of $NO_2$ per DU perturbation increases from 0.023 to 0.040 at 340

nm to 388 nm and decreases towards the visible wavelengths to 0.018 and 0.001 at 466 and 646 nm, respectively for $\tau_{440} = 0.4$. The spectral curvature of $\Delta\omega_o$ due to $O_3$ and $NO_2$ amounts is consistent with the known spectral absorption behavior of $O_3$ and $NO_2$. In addition, as expected the error in $\Delta\omega_o$ for both $O_3$ and $NO_2$ perturbation increases with decrease in $\tau_{440}$. For observations with $\tau_{440} = 0.4$, a perturbation of $\pm50$ DU of $O_3$ amount yields an error $\Delta\omega_o$ within $\pm0.005$ and $\pm0.011$ at 340 nm and 646 nm, respectively, while a perturbation of $\pm1$ DU of $NO_2$ amount yields an error $\Delta\omega_o$ within $\pm0.023$ and $\pm0.001$ at 340 nm and 646 nm respectively.

### 4.1.4 Theoretical errors in retrieved $\omega_o(\lambda)$ due to surface reflectance, ALH and surface pressure

Figure 8 shows the error analysis of the retrieved SSA as a function of wavelength and optical depth given a change in the (a) surface reflectance, (b) ALH, and (c) surface pressure. We perturb the surface reflectance by an absolute $\pm0.01$ for all wavelengths to derive an estimate of error incurred in the SSA retrieval. Our results indicate that $\Delta\omega_o$ increases with increasing wavelength and decreasing $\tau$ due to changes in surface reflectance for all aerosol types. For less-absorbing (urban) aerosols, the surface reflectance becomes increasingly important at the visible wavelengths than compared to absorbing (carbonaceous or dust) aerosols. For the observations with $\tau_{440} = 0.4$, it is noted that $\Delta\omega_o$ are within $\pm0.011$ and $\pm0.050$ at 340 nm and 646 nm, respectively. In contrast to the surface reflectance, the effect of ALH becomes prominent at near-UV wavelengths under the prescribed uncertainty of $\pm1$ km. The $\Delta\omega_o$ due to changes in ALH decreases with wavelength because of the gradually diminishing the intensity of Rayleigh scattering (proportional to $\lambda^{-4}$) and its radiative interactions with aerosols. For the observations with $\tau_{440} = 0.4$, it is noted that $\Delta\omega_o$ due to $\pm1$ km ALH are within $\pm0.028$ and $\pm0.001$ at 340 nm and 646 nm respectively for both absorbing and non-absorbing aerosols.

Another essential input in our SSA retrieval algorithm employed here is the terrain or surface pressure that determines the contribution of molecular scattering in the simulated TOA radiances through pre-computed LUT. We assume an uncertainty of $\pm100$ m in the terrain height equivalent to $\pm12$ mb or hPa pressure to derive an estimate of error incurred in aerosol SSA. Our analysis indicates that an overestimation of surface pressure produces lower aerosol SSA to compensate the higher radiances reaching the TOA. For absorbing (carbonaceous or dust) aerosols, the effect of surface pressure on the retrieved SSA is high in the UV spectrum than in visible, while for less-absorbing (urban) aerosols the error in retrieved SSA in the UV and visible spectral range are comparable exhibiting relatively small slope. Colarco et al., (2017) provides detailed examination of the effect of terrain pressure on OMI measurements and reported these effects are prominent at sites over mountainous regions owing to the coarser OMI footprint. In addition to the surface elevation uncertainty, their study investigates the

differences in OMAERUV (static) and MERRA (6-hourly) surface pressure and report differences up to ±15 hPa over land and higher over oceans. Therefore, the assumed uncertainty of ±100 m terrain height (~ ±12 hPa) in our sensitivity test accounts well for both these effects. For the observations with $\tau_{440} = 0.4$, it is noted that $\Delta\omega_o$ due to

±12 hPa surface pressure is within ±0.011 and ±0.006 at 340 nm and 646 nm, respectively.

### 4.1.5 Theoretical errors in retrieved $\omega_o(\lambda)$ due to variability in AOD around the site

The final variable that could possibly incur error in our SSA retrievals is the point AOD measurement from AERONET that is assumed homogenous/constant for the satellite radiance measurements within ±2 hours and 50

km radius of the site. To estimate the error in our retrieved SSA due to this assumption we initially estimate the variability in AOD derived from OMAERUV and MODIS-DB AOD products, and ±2 hours of AERONET AOD from the satellite overpass times. Based on the variability of AOD (not shown here) for the pixels within ±2 hours and 50 km radius of all sites considered we use a perturbation of ±0.2 for $\lambda < 400$ nm and ±0.1 for $\lambda > 400$ nm to determine the error in our SSA retrievals. As demonstrated previously, an overestimation of AOD produces lower

SSA (negative $\Delta\omega_o$) and vice-versa. For the observations with $\tau_{440} = 0.4$, it is noted that $\Delta\omega_o$ due to ±0.2 perturbation to the measured $\tau$ is within ±0.022 at 340 nm and due to ±0.1 perturbation is within 0.053 at 646 nm, respectively.

### 4.1.6 Combined maximum theoretical errors in retrieved $\omega_o(\lambda)$ due to all input variables

Table 2 summarizes the SSA error analysis 340 nm and 646 nm due to uncertainties in most relevant input variables. Among all input variables used in our algorithm, the $\Delta\omega_o$ at 340 nm arises mostly from (in descending order) the uncertainties in calibration of TOA radiances, spatio-temporal variability in the assumed homogeneous AOD over the site, sub-pixel cloud contamination, ALH, particle sizes and so on. While for the visible wavelength at 646 nm the $\Delta\omega_o$ arises mostly from (in descending order) cloud contamination, surface reflectance,

calibration of TOA radiances, particle sizes, trace gases, and so on. These sensitivity tests clearly indicate that $\Delta\omega_o$ is (a) spectrally dependent due to multiple variables, (b) decreases with increasing $\tau$, and (c) varies with absorbing nature of aerosols. The combined error in retrieved SSA can be now estimated as the square root of the sum of individual error squares (RMSE). Overall, for the observations with $\tau_{440} = 0.4$, the combined error in the retrieved SSA for absorbing (less-absorbing) aerosols are within ±0.051 (±0.043) and ±0.073 (±0.108) at 340 nm

and 646 nm, respectively. However, it should be noted that depending on the reliability of the input variables, the errors stemming from individual sources could be in opposite direction resulting in cancellation of errors. Under

such scenario, the combined error in the retrieved spectral SSA is expected to be much lesser than the maximum combined value reported here with our sensitivity tests.

## 4.2 Estimation of theoretical errors in the derived AAE

Similar to the estimation of errors in the retrieval of SSA, we conduct sensitivity tests to determine the errors in the computed AAE due to certainties in the retrieved SSA. We assume a fixed $\omega_o(388) = 0.9$, aerosol load $\tau_{440} = 0.4$, $EAE_{340-646}$ of 1.9, 0.2, and 1.9 for carbonaceous, dust and urban aerosols, respectively to derive the nominal AAE values at three wavelength pairs 354-388, 466-646, and 340-646. By using fixed $\tau(\lambda)$, we perturb the $\omega_o(\lambda)$ by ±0.01 intervals to compute the AAE. The uncertainty assumed here for the $\omega_o(\lambda)$ includes errors due to all

variables as described in the above section. The errors are reported as difference in AAE from perturbed SSA minus the AAE derived from nominal SSA.

Table 3 presents the theoretical uncertainty in computed AAE due to uncertainties in the SSA. As expected, the $\Delta AAE$ noted for all wavelength pairs increases with increasing $\Delta\omega_o$ for all aerosol types. It is noted that for fine

mode particles (carbonaceous and urban), an overestimation of $\omega_o$ produces lower AAE (negative errors) and underestimation of $\omega_o$ produces higher AAE (positive errors). In contrary for coarse mode particles, an overestimation of $\omega_o$ produces higher AAE (positive errors) and underestimation of $\omega_o$ produces lower AAE (negative errors). This is due to the fact that large particle size drives the scattering effect producing low aerosol absorption optical depths and, therefore, further overestimation of $\omega_o$ yields lower single scattering co-albedo $(1 -$

$\omega_o)$. The magnitude of $\Delta AAE$ is higher for overestimation than those noted for underestimation of SSA. In addition, it is noted that for fine mode particles the errors in $\Delta AAE_{354-388}$ (UV spectral range) are higher, while for coarse mode particles the errors in $\Delta AAE_{466-646}$ (visible spectral range) are higher than the other two wavelength pairs. For the assumed carbonaceous aerosols, perturbation of $\Delta\omega_o = \pm0.04$ yields a $\Delta AAE$ within ±0.13 for all wavelength pairs. For the urban aerosols, perturbation of $\Delta\omega_o = \pm0.04$ yields a $\Delta AAE$ within ±0.70 for all

wavelength pairs. However, for dust a perturbation of $\Delta\omega_o = \pm0.04$ yields a $\Delta AAE$ up to ±1.3 for 354-388 wavelength pair and much higher in the 466-646, and 340-646 wavelength pairs. Additional tests were performed by perturbing only one of the SSA while deriving AAE for any wavelength pair. The resulting $\Delta AAE$ is much higher than for the tests where SSA is perturbed for all wavelengths. It is noted that for even a small perturbation of $\Delta\omega_o = \pm0.01$ at one of the wavelengths in a pair, the $\Delta AAE$ is ±1.2, ±0.2, and ±0.4 for the wavelength pairs at

354-388, 466-646, and 340-646 respectively for all aerosol types. Overall, the errors noted for AAE are

consistent with the wavelength dependence of $\omega_o$ that is function of both size and absorbing nature of the particles.

## 5 Comparison with AERONET SSA product

We compare our aerosol SSA retrievals at visible wavelengths with those from the AERONET inversion data set.
The comparison is limited to AERONET Level 2 reliable retrievals as determined the aerosol load at 440 nm ($\tau_{440}$ > 0.4). We emphasize here that since AERONET SSA is a derived quantity and cannot be considered as 'ground truth', this comparison serves as a consistency check rather than a strict validation exercise. The nearest AERONET wavelength available for comparison with OMI wavelength (388 nm) retrievals is at 440 nm. To facilitate the comparison, we use the AAE computed from our retrievals at 388-466 wavelength pair to transform
the retrieved SSA at 388 nm to 440 nm ($440_{OMI}$). While at the MODIS wavelengths 466 and 646 nm, the AERONET SSA were computed by linear interpolation of the values reported at 440 and 675 nm. This conversion will unlikely introduce notable bias in our comparison as the difference in the nearest wavelengths of both data sets is very low (< 30 nm).

First, we investigate the consistency of retrieved SSA for selected sites for which the prevailing aerosol types and local source environment are well known as documented in several studies in the literature. We use the AERONET SSA from period 2005-2016 for the comparison. Figure 9 shows the comparison of spectral SSA (box plots showing lower and upper quartile of observations with white line representing its mean value) for three distinct aerosol types sampled from the selected sites. The mean AAE derived for the visible wavelength pair at
466-646 nm agrees within 0.5 or less with the AERONET values for all aerosol types and sites considered. However, it should be noted that AAE is highly susceptible to small changes in the retrieved SSA for both data sets. For dust aerosols, the retrieved average SSA show good agreement with AERONET SSA obtained at Dakar and Ouagadougou (within ±0.008), while the differences in SSA (retrieved minus AERONET) at Tamanrasset and Solar Village are 0.02 and 0.05, respectively at 466 nm. Although our sample is limited for few days that met
our criteria for subset, the observed differences in SSA are within the overall uncertainty estimates presented in the previous section. For carbonaceous particles, the retrieved and AERONET SSA agree well with differences of less than 0.015 for Alta_Floresta, Cuiaba, and Mongu. However, notable difference in SSA (0.045) is observed at the Lake_Argyle with high absorption for AERONET than our retrieved value at both 466 nm and 646 nm. For urban/industrial aerosols at GSFC, Avignon, Moldova, and Cairo, the retrieved SSA values agree within the

uncertainty estimates. Particularly notable difference (-0.05) is found for Avignon at 646 nm. It should be noted that, while AERONET Version 3 employs surface reflectance using BRDF parameters from MODIS BRDF/Albedo CMG Gap-Filled Snow-Free Product MCD43GF (Sinyuk et al., 2020), we use MODIS MCD19A1 BRF/surface reflectance product. The use of different surface reflectances data in the two retrieval algorithms likely contributes to some of the observed SSA differences. Although for AERONET SSA retrievals

that uses upward viewing sky radiance measurements surface reflectance is relatively a small source of error, for our retrievals (as discussed in the sensitivity analysis), surface reflectance is the second highest source of errors at visible wavelengths after cloud contamination. Additionally, for our retrievals as well as with AERONET, uncertainties in SSA increases with wavelength for fine mode particles (carbonaceous and urban) with high EAE, notably for weakly absorbing aerosols. For example, the ±0.03 uncertainty in AERONET SSA at 440 nm for $\tau_{440}$

~0.4 is achieved for the NIR wavelength (1020 nm) at $\tau_{440}$ ~0.6 for the fine mode particles observed over the GSFC site (Sinyuk et al., 2020).

Figure 10 shows the absolute difference in retrieved SSA with AERONET as a function of $\tau_{440}$ for all collocated observations. For the SSA at 440 nm$_{OMI}$, the observations within ±0.03 (±0.05) envelopes are 37% (59%), 39%

(63%) and 39% (63%) for dust, carbonaceous, and urban aerosol types, respectively. As expected, the SSA difference is highest for lower AOD's and decreases with increasing aerosol load for all aerosol types. For the SSA at 466 nm, the observations within ±0.03 (±0.05) envelopes are 22% (54%), 33% (59%) and 33% (54%) for dust, carbonaceous, and urban aerosol types, respectively. In terms of particle sizes, the difference in SSA for fine-mode ($\alpha_{440-870}$ > 1.2) are noted to be more scattered than the difference in SSA noted for coarse mode

particles. This could be partially explained by the variability in sizes for fine-mode particles from the seasonal climatology of particle sizes employed in our algorithm. For the SSA at 646 nm, the observations within ±0.03 (±0.05) envelopes are 73% (87%), 38% (60%) and 28% (45%) for dust, carbonaceous, and urban aerosol types, respectively. Spectrally, there is significantly more scatter at 646 nm with SSA differences in the range -0.3 to 0.2 for urban aerosols. The RMSE of the SSA ranges from 0.04 to 0.09 with lowest error for dust particles at 646 nm

and highest error for urban aerosols at the same wavelength. This owes to the high spectral AODs for coarse mode particles through UV-Visible where sufficient absorption signal is available for the retrieval of SSA. While for fine mode particles with the decreasing spectral AODs from UV-Visible the absorption signal becomes weak, particularly notable for the less absorbing (urban) aerosols.

Figure 11a shows the absolute difference in retrieved and AERONET SSA as a function of optical depth by combining all aerosol types together. The differences in SSA for all wavelengths at $440_{OMI}$, 466 and 646 nm are higher for lower $\tau$ and become negligible for higher $\tau$. It is observed that at $440_{OMI}$ and 466 nm the observations with positive differences are relatively more than that at 646 nm. Comparison of the retrieved SSA with AERONET SSA for all aerosol types is shown in Figure 11b. Our retrieved SSA at $440_{OMI}$, 466 and 646 nm

agree within ±0.03 of AERONET SSA for 38% (0.05), 32% (0.06) and 34% (0.08) of observations (RMSE) respectively.

It is important to note here that our retrieval method and that used in the AERONET inversion differ fundamentally in several aspects. (i) different source of surface reflectance data used, (ii) instantaneous particle

sizes derived from sky radiances measurements by AERONET and seasonal climatological average particle sizes used in our retrievals, (iii) the use of multi-spectral, multi-angular sky radiance measurements by AERONET along the almucantar plane or hybrid scan and the single-view TOA radiance measurements by the satellites in our retrievals, and (iv) the use of relatively strong constraint on spectral variation of imaginary refractive index for the fine mode particles in AERONET SSA retrievals, while our retrievals of SSA are carried for each

wavelength independently. The use of relatively strong constraint on spectral variation of imaginary refractive index for the fine mode particles with high $\alpha_{440\text{-}870}$ for AERONET inversion owes to the lower spectral AODs and diminished absorption signal strength at higher wavelengths insufficient for a robust absorption retrieval (Dubovik et al., 2006). The differences noted between our SSA retrievals and that from AERONET at different wavelengths could stem from one or more sources of the differences listed above.

**6 Spectral Aerosol Absorption in Major Aerosol Environments**

In this section, we describe regional average aerosol absorption and AAE derived from our subset of results over worldwide regions dominated by carbonaceous, dust, and urban aerosols. The AERONET sites selected for this analysis are based on the dominant samples observed here and well-known aerosol sources from the literature (e.g., Eck et al., 2010, 2013; Giles et al., 2012 and references therein). The aerosol-typing scheme employed in

this work based on UVAI and particle sizes are only to guide our algorithm to include ALH in the SSA retrieval procedure. Therefore, it should be that aerosol types as described below do not represent a robust characterization and overlap of aerosol samples are observed at few sites especially over the regions where mixtures of aerosols are found.

The average of aerosol SSA and AAE derived for all sites considered in this work are presented in Table S1 as supplementary materials. In addition, the corresponding particle size distributions used for developing the LUT radiances for these sites are presented in supplementary materials as well.

## 6.1 Carbonaceous aerosols

Emissions from biomass burning are one of the major contributors of carbonaceous aerosols found in the
atmosphere. These carbonaceous aerosols are primarily composed of black carbon and organic carbon components in addition to minor fractions of inorganic components (Andreae and Merlet, 2001). Studies show that black carbon amounts in the atmosphere are high absorbers of solar radiation and have near unity AAE due to invariant imaginary part of refractive index in the UV-Visible spectrum (Bergstrom, 1973; Bohren and Huffman, 1983; Bergstrom et al., 2002). The typical spectral behavior for carbonaceous aerosols has decreasing
SSA with increasing wavelength in the visible spectrum (Eck et al., 1998; Reid and Hobbs, 1998). Additionally, the presence of organic carbon amounts shows enhanced absorption in the UV region (Kirchstetter et al., 2004). Our observations for the carbonaceous aerosols from worldwide biomass burning regions depict these characteristics very well. The seasonal average of spectral SSA for carbonaceous aerosols found over major aerosol environments are shown in Figure 12.

For the Missoula site located in Northwestern United States (US), carbonaceous aerosols are observed during JJA in our sample. Aerosols observed over Missoula are primarily emitted from natural forest fires of the northwestern US in the dry season (June through September). The spectral SSA of these aerosols is noted to increase from 340 nm ($0.89\pm0.02$) to 466 nm ($0.94\pm0.03$) followed by a decrease toward the 646 nm ($0.90\pm0.06$).
Average $\tau_{440}$ and $\alpha_{440-870}$ are about 1.2 and 1.8 respectively, while the average $AAE_{340-646}$ is noted as 1.8. Our results are consistent with the insitu measurements of wildfire smoke at the Missoula ground station for 2017 and 2018 summer, that reports an average SSA of 0.93-0.94 at 401 nm and AAE 1.7-1.9 over the spectral range 401-870 nm (Selimovic et al., 2020).

Over South America, our subset of carbonaceous aerosols is observed for sites at the Alta Floresta, Cuiaba, and Ji Parana in Brazil, and Santa Cruz site in Bolivia. In general, carbonaceous aerosols over South America are dominantly emitted from biomass burning during southern hemisphere spring (JJA) and summer (SON), with distinct peaks in August and September. Most aerosol emissions are associated with biomass burning for land and

agricultural management practices. The regional average SSA of these aerosols at 466 nm is noted to be 0.92±0.03 (0.93±0.02) during JJA (SON) months. Spectral SSA increases from 0.90±0.02 at 340 nm to 0.93±0.03 at 388 nm followed by a decrease toward the visible wavelengths. Average $\tau_{440}$ and $\alpha_{440-870}$ are about 1 and 1.8 respectively with mean $AAE_{340-646}$ ranging between 1.5-1.8 for both JJA and SON months. Among the sites considered here, Cuiaba located in the cerrado ecosystem exhibit highest aerosol absorption ($\omega_o$ ~0.90±0.03 at 466 nm), while the remaining sites are surrounded by tropical rainforest exhibits relatively less absorption ($\omega_o$ ~0.93±0.03 at 466 nm). Burning of cerrado (wooded savanna) and rainforest dominantly happens through flaming and smoldering phase combustion respectively, resulting in the noted variation of aerosol absorption over these sites (Schafer et al., 2008).

Over Southern Africa, our subset of carbonaceous aerosols is observed for sites at the Mongu in Zambia, and Skukuza in South Africa. Emissions from biomass burning primarily for agricultural and land management practices are major source of aerosols over Southern Africa (Eck et al., 2001, 2003). In addition, crop residue burning, heavy industrial facilities and episodic dust commonly dictate the aerosol amounts over Southern Africa. Fine mode carbonaceous aerosols noted over these sites shows high average absorption during JJA period. The average SSA for these aerosols increases from 340 nm to 466 nm and then decreases at longer wavelengths. Distinct seasonality in absorption for carbonaceous particles is observed with maximum (minimum) value of 0.87±0.02 (0.90±0.03) at 466 nm for JJA (SON) months. The range of regional average values of $\tau_{440}$ and $\alpha_{440-870}$ for these aerosols are about 0.76 to 1.00 and 1.76 to 1.83 ranges, respectively. Average $AAE_{340-646}$ of these carbonaceous aerosols is noted to be ~1.72 and 1.56 for JJA and SON months.

For the sample obtained over Sahel region, carbonaceous aerosols are observed at Ilorin during DJF. Fine mode aerosols observed over Ilorin are primarily emitted from the biomass burning of the grasslands and savanna in Sahelian and Sudanian zones during the dry season (November through March). Our results show these aerosols exhibit significant absorption with the average SSA ~0.86±0.02 and 0.87±0.03 at 340 nm and 646 nm, respectively. Average $\tau_{440}$ and $\alpha_{440-870}$ are about 1 and 1.3 respectively with mean $AAE_{340-646}$ 1.37 for DJF period. The high aerosol absorption noted here is consistent with the AERONET data analysis that reports high aerosol absorption with increasing fine-mode fraction (FMF) of particles (SSA ~0.80-0.87 and 0.81-0.85 at 440 nm and 675 nm for observations with FMF of 0.75-0.54 at 675 nm) over Ilorin during the dry season (Eck et al., 2010). The relatively low $\alpha_{440-870}$ (1.3) and nearly invariant spectral SSA through UV to visible range, suggests the aerosols noted over Ilorin during DJF are mixtures of black carbon and dust particles. Emissions from burning of

grasslands and savanna in the Sahelian and Sudanian zones dominantly happens through flaming phase combustion producing high amounts of soot (Eck et al., 2010). In addition to the biomass burning, fossil fuel combustion, and vehicular emissions, the vast number of gas flaring stations (> 300) around the Niger Delta produces high emissions (Onyeuwaoma et al., 2015). Highly absorbing black carbon amounts observed at Ilorin is likely a result of such emissions.

720

For the Cairo site in the Middle East, carbonaceous aerosols are noted in our sample. Cairo in the Middle East is one of the highly polluted places among the megacities worldwide. Carbonaceous aerosols over Cairo are primarily emitted from burning of the agricultural waste in the Nile delta during the burning season from September through December (El-Metwally et al., 2008). Emissions from agricultural waste during the burning season adds additional aerosol burden over Cairo to the prevailing high pollution levels throughout the year. The average SSA for these aerosols ranges from $0.89\pm0.03$ to $0.91\pm0.05$ exhibiting weak spectral dependence from 340 nm to 646 nm. Average $\tau_{440}$ and $\alpha_{440-870}$ are about 0.6 and 1.4 respectively with $AAE_{340-646}$ about 1.9 during DJF. The spectral dependence noted for these aerosols likely indicate mixture of black and organic carbon amounts.

730

Over Northeastern China, carbonaceous aerosols are observed throughout the year at the sites Beijing and XiangHe from our sample. Prevailing aerosol loading over these sites constitutes emissions from biomass burning during the SON, episodic dust outbreaks during MAM and the industrial/vehicular emissions throughout the year. The spectral behavior of carbonaceous aerosols at these sites shows increase in SSA from 340 nm to 466 nm and thereafter remains near constant or slightly decreases with an UV-Vis dependence ($AAE_{340-646}$) ranging from 1.60 to 1.74. However, significant seasonality is noted with minimum ($0.96\pm0.03$ at 466 nm) and maximum ($0.92\pm0.04$ at 466 nm) absorption during JJA and DJF, respectively. The increase in SSA is likely result of humidification and secondary aerosol processes during JJA. The high aerosol absorption noted during winter (DJF) is likely contributed from high amounts of local fossil fuel combustion and agricultural waste burning.

740

For the sample obtained over Northern India, carbonaceous aerosols are observed over Kanpur and Gandhi College during SON and DJF. Emissions from crop residue burning during SON and biomass burning for residential heating in DJF prevail over the entire Indo-Gangetic plain and likely result in such absorption. Regionally brick kilns and power plants located in the vicinity contribute to significant aerosol emissions. In addition, other industrial activities and vehicular emissions are observed throughout the seasons. The spectral

behavior of these aerosols shows increase in average SSA from 340 nm (0.91±0.02) to 466 nm (0.93±0.03) and slight decrease till 646 nm (0.91±0.04). The regional average $AAE_{340-646}$ for carbonaceous aerosols ranges 1.2 to 1.5. The weak spectral dependence of SSA noted here is consistent with AERONET SSA analysis. For the aerosols observed over the Kanpur site, spectral dependence of aerosols becomes nearly invariant (SSA ~0.89 at 440 and 675 nm) for high fine mode fraction (FMF ~0.85) of aerosols (Eck et al., 2010).

Over Northern Australia, carbonaceous aerosols are observed at the sites Jabiru and Lake_Argyle during SON. In general, Northern and Western parts of the Australia are covered with savanna grasslands, woodlands, and forests, where biomass burning due to natural fires and land management practices are known to produce high aerosol emissions during the dry season (Scott et al., 1992; Mitchell et al., 2013). Our results indicate the average SSA for carbonaceous aerosols over Northern Australia increases with wavelength from 0.87±0.02 (340 nm) to 0.89±0.03 (388 nm) and then decreases to 0.87±0.06 (646 nm). Average $\tau_{440}$ and $\alpha_{440-870}$ for these aerosols is 0.63 and 1.61, respectively, while the UV-Vis spectral dependence ($AAE_{340-646}$) is noted to be ~1.41. Such behavior of fine-mode particles is likely a result of a mixture of black carbon and organic carbon amounts in the atmosphere, producing stronger (weaker) absorption in the UV (Vis) wavelengths.

## 6.2 Dust

For dust aerosols, minerals such as hematite and other form of oxides play role in scattering/absorption of particles. The absorbing nature of pure dust aerosols close to the source is sensitive to the presence of hematite than other minerals at shorter wavelengths (Sokolik and Toon, 1999). In addition to the sedimentation of coarse aggregates, the dust aerosols observed away from the source are sometimes found to have mixed (internally or externally) with anthropogenic aerosols altering its absorbing nature. Studies show that the typical spectral behavior of dust absorption decreases with increasing wavelength primarily due to attributed to the larger size of the particles (Sokolik and Toon, 1996, 1999). However, the absorption of dust from different sources is known to vary depending on the mineral composition of the soil origin (Di Biagio et al., 2019). The seasonal average of spectral SSA for coarse-mode dust aerosols found over major aerosol environments are shown in Figure 13.

For the dust aerosols sample obtained at Saharan region at the sites Tamanrasset and Saada, the average SSA is ~0.94 at 466 nm. The average SSA for these dust aerosols increase with wavelength from about 0.86±0.03 at 340 nm to 0.97±0.02 at 646 nm. The seasonal average $AAE_{340-646}$ for dust aerosols derived at these sites range from 2.7 to 3.3 with no distinct seasonality in the average spectral SSA. Dust aerosols over the Middle East are

observed for the sites Cairo, Solar Village and SEDE BOKER with an average SSA ~0.95±0.02 at 466 nm. From UV (340 nm) to visible (646 nm), the regional average SSA for dust over the Middle East sites ranges from 0.89±0.03 to 0.98±0.02, while the $AAE_{340-646}$ ranges from 2.7 to 3.8. No distinct seasonality in SSA is found from our sample for the dust aerosols over Middle East. However, a slight increase in SSA at UV wavelengths is noted during winter (DJF). Examining individual sites reveal this feature corresponds to the aerosols over Solar_Village (Table S1). The increase in SSA and high $AAE_{354-388}$ noted for Solar_Village during DJF likely indicates transport of aerosols from neighbouring regions.

Over Sahel, dust aerosols are observed for several sites that include Agoufou, Banizoumbou, Dakar, IER_Cinzana, Ilorin, Ouagauodu, and Zinder_Airport. Located in the middle of Sahelian region through the west Africa, these sites are influenced by both dust and biomass burning emissions (Basart et al., 2009). It should be noted that for the identifying dust in this work, we use $\alpha_{440-870} \leq 0.2$ derived from AERONET. The regional average spectral SSA for dust aerosols derived here resembles typical dust absorption curve (increase in SSA with wavelength, ~0.87-0.91±0.03 at 340 nm to 0.95-0.97±0.02 at 646 nm) with absorption ranging ~0.93-0.94±0.02 at 466 nm. No distinct seasonality in absorption of dust aerosols is noted over Sahel. Average $AAE_{340-646}$ for dust over Sahel ranges 2.0-2.3 for all seasons with an exception during SON where $AAE_{340-646}$ is noted to be relatively less 1.57.

Over Northern India, dust aerosols are observed during spring (MAM) and summer (JJA) months for the sites at Jaipur and Kanpur, where the former site is in proximity to the Thar desert and the later site is influenced by the dust transport. The average SSA shows a steep increase from 340 nm (0.88±0.02) to 466 nm (0.95±0.03), and a relatively smaller increase from 466 nm to 646 nm (0.97±0.02). The dust aerosols noted here has average $\tau_{440}$ 0.73 to 0.77 and exhibit $AAE_{340-646}$ between 2.9 to 3.4. Compared to AERONET the absorption for dust aerosols derived here agrees well. Eck et al (2010) reports the coarse mode particles noted over Kanpur during pre-monsoon (MAM) months exhibit climatological average SSA ~0.89 and 0.95 at 440 and 675 nm, respectively.

For the samples obtained over northeastern China, dust aerosols are observed over Beijing and XiangHe during spring (MAM). The spectral curve of regional average SSA shows an increase from 0.87±0.03 at 340 nm to 0.95±0.03 at 646 nm. The average $AAE_{340-646}$ obtained for the dust aerosols at these sites is 1.56. Among the regional dust observations presented here (Figure 12), northeastern China exhibit high absorption in visible wavelengths ($\omega_o$ ~0.93±0.03 and 0.95±0.03 at 466 and 646 nm) and low $AAE_{340-646}$. It is likely that these coarse

particles are influenced by black carbon components over such highly polluted environments and exhibit anomalously low AAE than dust particles noted for other regions. However, due to large particle size ($\alpha_{440-870}$ ~0.09) the spectral SSA noted still shows increasing SSA with wavelength. Chaudhry et al., (2007) reported insitu measurements of coarse mode particles over XiangHe during March-2005 that exhibits high absorption in visible wavelengths ($\omega_o$ ~0.70-0.94 at 450, 550 and 700 nm). Li et al., (2007) explained the variation in SSA for coarse particles during March-2005 over XiangHe is a result of synoptic fluctuation – passage of cold fronts that uplifted ground-level pollution to higher altitudes influencing the aerosol absorption. Similar low $AAE_{340-646}$ values for coarse mode ($\alpha_{440-870} < 0.2$, dust) are noted for few sites over the Sahelian region during DJF (burning season). It is likely that these coarse particles are influenced by black carbon amounts emitted from biomass burning.

### 6.3 Urban-Industrial aerosols

Urban aerosols dominantly constitute sulfates and other forms of nitrate particles. Additionally, industrial emissions and fossil fuel combustion produces various forms of carbon (organic and black carbon) that contribute to the overall optical properties of urban aerosols. Further, the aerosol size growth due to increase in relative humidity in the atmosphere and coagulation processes are known to alter the absorbing nature of aerosols. The typical spectral SSA of urban aerosols decreases with increase in wavelength from UV-Vis spectrum (Bergstrom, 1972). The seasonal average of spectral SSA for carbonaceous aerosols found over major aerosol environments are shown in Figure 14.

Aerosols observed over the Central United States (US) at the sites Sioux Falls and Bondville are primarily produced from industrial activities and vehicle emissions. The average aerosol SSA for these urban aerosols shows increase ($0.93-0.95\pm0.03$) in the wavelength range 340-388 nm followed by a decrease ($0.89\pm0.06$ at 646 nm) towards the visible wavelengths. Average $\tau_{440}$ and $\alpha_{440-870}$ are about 0.48 and 1.7, respectively, while the average $AAE_{340-646}$ noted as 1.1. Urban aerosols over the Mid-Eastern US are noted for the sites at GSFC, MD_Science_Center, and SERC. The average aerosol SSA noted for these sites increases from ($0.93\pm0.02$) 340 nm to ($0.95\pm0.03$) 388 nm and then decreases attaining a maximum absorption (~$0.87\pm0.06$) at 646 nm. The regional average SSA for the MAM and JJA months at 466 nm is $0.94\pm0.03$ and $0.92\pm0.03$, respectively. Seasonally, the average SSA noted for JJA follows typical 'urban' absorption curve, while the spectral SSA shows decrease in aerosol absorption in the visible wavelengths for MAM. Since there are no significant changes in aerosol sources during MAM and JJA over Mid-Eastern US, the seasonal variation in SSA noted for visible

wavelengths are likely stemming from weak absorption signal insufficient for a robust retrieval or partially due to poor sample size. Recall, for weakly absorbing aerosols the error in SSA retrievals at visible wavelengths are high due to identified factors from our analysis such as cloud contamination, surface reflectance, particle size, etc.

For the site in Mexico City, Mexico, urban aerosols are noted in our sample. In general, Mexico City is a densely populated urban location that is well known for its high pollution levels among the other megacities worldwide. In addition to the high concentration of aerosols from fossil fuel combustion throughout the year, Mexico City also experiences biomass-burning aerosols during the relatively dry months of March-May from local sources. The seasonal average aerosol SSA over Mexico City shows decrease in absorption in the UV ($0.90\pm0.03$ and $0.91\pm0.05$ at 340 and 388 nm) and increases towards the visible wavelengths during MAM. However, the spectral SSA noted for DJF ($0.88\pm0.04$ and $0.87\pm0.05$ at 340 and 388 nm) and SON ($0.88\pm0.04$ and $0.85\pm0.05$ at 340 and 388 nm) deviates from known pattern and exhibit high absorption in the UV spectral range with nearly constant or slight increase in the visible spectral range. The average SSA at 466 nm for DJF, MAM and SON months are $0.85\pm0.06$, $0.86\pm0.06$ and $0.84\pm0.06$, respectively with average $\alpha_{440-870}$ about 1.7 throughout the seasons. Although the seasonal average $AAE_{340-646}$ of aerosols over Mexico City ranges from 0.94-1.27, the $AAE_{354-388}$ is higher for (5.13) MAM than compared to (2.2) DJF and (0.6) SON months. It is likely that our aerosol samples obtained during MAM are influenced by biomass burning emissions from local neighbour sources exhibiting high $AAE_{354-388}$ values suggesting the absorption is driven by organic components (Barnard et al., 2008). While for the samples noted during SON and DJF the aerosol absorption is primarily driven by the black carbon components emitted from the heavy industrial and vehicular fleet over the region.

Our subset of samples noted urban aerosols at the site Sao Paulo, South America. Sao Paulo is the largest megacity in South America with population exceeding 21 million inhabitants. Heavy industrial and vehicular emissions are the dominant source of aerosols observed over Sao Paulo. In addition, aerosols from northern parts of the Amazon Basin advecting south or southeast over Sao Paulo is not uncommon during the peak burning season (August-September). Average $\tau_{440}$ and $\alpha_{440-870}$ noted for these aerosols are ~0.57 and 1.5, respectively during JJA and SON months. Spectral SSA noted here resembles typical urban absorption curve during SON. However, during JJA spectral SSA shows steep decrease from 340 nm to 388 nm ($0.87\pm0.03$ to $0.85\pm0.03$) and remains nearly invariant towards the visible spectral range from 466 to 646 nm ($0.85\pm0.05$ to $0.84\pm0.07$). The average SSA noted at 466 nm is $0.88\pm0.06$ ($0.92\pm0.06$) for JJA (SON) months. Our results indicate that the urban

aerosols at Sao Paulo are more absorbing during JJA than in SON, it is likely that mixing of carbonaceous and urban aerosol samples caused this feature.

For the urban aerosols observed over Cairo and Nes Ziona in the Middle East, the average SSA is noted to be 0.89±0.03 at both 340 nm and 646 nm. While urban aerosols and pollution prevail over Nes Ziona, emissions from crop residue burning (rice straws) over the Nile delta region during winter (DJF) and heavy pollution dictate the aerosol absorption noted over Cairo. Average $\tau_{440}$ and $\alpha_{440-870}$ noted for these aerosols are ~0.60 and 1.3,

respectively throughout the seasons, with mean $AAE_{340-646}$ ranging between 1.5-2.0. Unlike typical urban aerosol absorption curve, the spectral SSA noted for these aerosols do not exhibit steep decrease from 388 to 646 nm indicating mixture of dust and carbonaceous particles.

Over Europe, dominantly urban aerosols are observed at the sites Avignon, Carpentras, Ispra, IMS-METU-ERI,

Lille, Lecce_University, Minsk, Modena, Moldova, Moscow_MSU, Palaiseau, Rome, and Thessaloniki. Primarily industrial activities, and vehicular emissions are dominant sources of aerosols over Europe. In addition, fuel combustion for residential heating during winter and episodic dust transported during spring-summer over Iberian Peninsula and Mediterranean basin are known to influence the aerosol loading (Basart et al., 2009; Mallet et al., 2013). The regional average SSA increases from 340 nm (0.91±0.02) to 388 nm (0.93±0.03) and then

decreases reaching a minimum value (0.87±0.07) at 646 nm for most seasons. The observed aerosol absorption is similar for spring (MAM) and summer (JJA). However, there is an increase in absorption at wavelengths other than 646 nm for fall (SON) that reaches a maximum absorption 0.86±0.03 and 0.90±0.04 at 340 nm and 466 nm, respectively. These highly absorbing aerosols in our sample are noted over polluted urban sites at Ispra, Modena, and Rome from northern to central Italy are likely result of mixtures of pollution and carbon particles from wood

burning for domestic heating during fall through winter months. Our results are consistent with that showing decrease in aerosol SSA over Ispra, Northern Italy during fall through winter (Putaud et al., 2014). Average $\tau_{440}$ and $\alpha_{440-870}$ noted for these aerosols are ~0.50 and 1.5, respectively throughout the seasons, with mean $AAE_{340-646}$ ranging between 1.0-1.3.

Over Northeastern China, urban aerosols are noted for the sites Beijing, and XiangHe. Significant seasonality in aerosol SSA is noted with minimum (0.93±0.04 at 466 nm) and maximum (0.88±0.05 at 466 nm) absorption during JJA and DJF, respectively. Seasonal variability in aerosol absorption noted here is likely caused by the humidification and secondary aerosol processes. In addition to the high industrial and vehicular emissions

throughout the year, fuel combustion for residential heating purposes and agricultural waste burning during DJF adds additional aerosol burden over the Northeastern China. High aerosol absorption noted during DJF is likely a result of such emissions. Average $\tau_{440}$ and $\alpha_{440\text{-}870}$ noted for these aerosols are ~0.65 and 1.4, respectively throughout the seasons, with mean $AAE_{340\text{-}646}$ ranging between 1.2-1.7. For the urban aerosols noted at the sites Shirahama, and Osaka in Japan during MAM, the spectral aerosol SSA noted follows typical urban absorption curve with slight increase in SSA from (0.90±0.02) 340 nm to (0.92±0.02) 388 nm and then a steep decrease towards (0.87±0.07) 646 nm. Average $\tau_{440}$ and $\alpha_{440\text{-}870}$ noted for these aerosols are ~0.5 and 1.5, respectively with mean $AAE_{340\text{-}646}$ about 1.2.

Over Northern India, urban aerosols are noted for the sites at Jaipur, Kanpur, and Gandhi College. Major source of aerosols over the region includes industrial and vehicular emissions, combustion of biomass and fossil fuels, and seasonally occurring agricultural burning. The average aerosol SSA increases from (0.88±0.02) 340 nm to (0.90±0.04) 466 nm and decreases towards 646 nm (0.87±0.07) 646 nm. These aerosols are noted to exhibit relatively high absorption $\omega_o$ ~0.89±0.04 at 466 nm. Average $\tau_{440}$ and $\alpha_{440\text{-}870}$ noted for these aerosols are ~0.60 and 1.4, respectively throughout the seasons, with mean $AAE_{340\text{-}646}$ about 1.3. Throughout the seasons, influence of pollution aerosols is clearly evident in the observed aerosol absorption.

## 7 Discussion

Through extensive studies in the literature, it is known that optical properties of biomass burning aerosols depend on fuel/vegetation type, combustion processes, and available moisture content (e.g., Ward, 1992; Reid and Hobbs, 1998; Reid et al., 1998; Eck et al., 2001). Such studies reported varying properties of aerosols emitted from the two phases of vegetation burning: flaming and smoldering. While flaming phase rapidly oxidizes the available volatile hydrocarbons in the biomass, smoldering phase mostly requires a surface where slow diffuse oxygen converts the biomass through exothermic reaction. In general, burning of grasslands happens dominantly through flaming phase combustion that emits high amounts of black carbon, while smoldering phase combustion prevail the burning of woodlands/deciduous forest that emits less black carbon and more organic carbon. The observed aerosol absorption at the biomass burning sites (Figure 12) clearly makes this distinction. Over South America, in addition to the emissions from burning rainforest (nearby Alta Floresta and Ji Parana), Cerrado (wooded grasslands) type vegetation dominates at the Cuiaba site (Schafer et al., 2008). Biomass burning of tropical forests occurs through smoldering combustion exhibiting aerosol $\omega_o$ (0.93±0.03 at 466 nm). Compared to

South America, the aerosols over Southern Africa have distinct seasonality and high absorption ($\omega_o \sim 0.88\pm0.02$ at 466 nm). Eck et al (2013) demonstrated that this seasonality in aerosol absorption is likely a result of shift in fuel type and combustion process. At the beginning of dry season (starting June), the savanna grasses in the central region are prone to undergo a rapid burning through flaming process, while in the late dry season (ends November) the wooded lands located in southeastern parts begins to burn dominantly through smoldering phase. For the savanna with open grasslands in the northern Australia, biomass burning happens dominantly through flaming phase producing high amounts of soot, as also noted in our retrievals. Figure 15 shows the range of AAE obtained for carbonaceous aerosols at three wavelength pairs. Overall, the average slope of absorption in visible ($AAE_{466-646}$) and UV-Vis ($AAE_{340-646}$) for carbonaceous aerosols is found to be within 2. This is consistent with the studies that report AAE of biomass burning aerosols from several field campaigns in the range 1 to 3 (Kirchstetter et al., 2004; Schnaiter et al., 2005; Bergstrom et al., 2007; Clarke et al., 2007). However, the average $AAE_{354-388}$ obtained is high up to 4 for most regions. This is likely a result of higher organic matter in the regional biomass types and highlights the importance of UV spectral region in delineating such group of aerosols. Despite high errors in the estimated uncertainties of AAE computation over the narrow spectral range 354 to 388 nm, our results are consistent with studies show that spectral dependence of aerosol absorption in the UV-visible range can be high up to 6 for aerosols with organic compounds (Kirchstetter et al., 2004; Bergstrom et al., 2007 and references therein). Among the biomass burning regions, for the emissions where contribution of flaming phase combustion is high, the mean $AAE_{354-388}$ noted is relatively low–Northern Australia (2.1) and Sahel (1.6). Further, it is noted that carbonaceous aerosols observed over Northern India, Northeastern China, Sahel, and the Middle East has an average $\alpha_{440-870} \sim 1.4$ (i.e, at the lower end of the fine-mode range), while in South America, South Africa and Australia has average $\sim 1.8$. This indicates the role of aerosol mixing (for example mixing of dust in Sahel), aging, humidification (for example aerosols observed during summer over Northeastern China) and secondary processes in emanating the observed variability in aerosol absorption other than its composition alone.

Figure 16 shows the range of AAE obtained for dust aerosols at three wavelength pairs. Regional average of the UV-Vis spectral dependence ($AAE_{340-646}$) is found to be close to or greater than 3 for all regions, except for the Sahel and Northeastern China, where average value ranges 1.5 to 2.5. Although no distinct seasonal variation in spectral absorption of dust is noted, the variability in spectral dependence over the regions is quite evident. The regional average of SSA for dust aerosols from 340 nm to 646 nm are $0.87\pm0.02$ to $0.98\pm0.02$, $0.89\pm0.03$ to $0.96\pm0.02$, $0.89\pm0.02$ to $0.98\pm0.01$, $0.87\pm0.02$ to $0.97\pm0.02$, and $0.87\pm0.03$ to $0.95\pm0.03$ over Sahara, Sahel,

Middle East, Northern India, Northeastern China, respectively. While the average SSA ranges 0.87-0.89±0.03 at 340 nm, differences in retrieved SSA are evident with increasing wavelength. The regional average spectral SSA noted at 466 nm are 0.94±0.02, 0.93±0.02, 0.96±0.02, 0.95±0.02 and 0.93±0.02 for Sahara, Sahel, MiddleEast, Northern India, and Northeastern China, respectively. A recent study uses soil samples collected from over different arid regions worldwide to characterize the mineral composition of dust and estimate the spectral SSA using the measured scattering absorption coefficients through aethalometer operating at seven discrete wavelengths from 370–950 nm (Di Biagio et al., 2019). Their study reports high absorption (~0.70-0.75 at 370 nm) for dust samples obtained over Niger, Mali, Southern Namibia, and Australia due to the presence of higher amounts of iron oxides. While for the samples collected over Bodélé, Northern Namibia, Arizona the estimated absorption (~ 0.91-0.96 at 370 nm) and amounts of iron oxides are relatively low. In comparison, our retrievals indicate the dust aerosols noted at Sahel and northeastern China are highly absorbing, while those noted over MiddleEast, and Northern India are less absorbing and dust over Sahara shows intermediate absorption. These results can be explained by combination of varying mineral composition (iron oxide amounts) and mixing of dust with other sources along the transport pathway. However, the magnitude of SSA and AAE reported by Di Biagio et al., (2019) are lower than those retrieved here for all regions. The reason for the differences noted in SSA is unknown and needs further investigation. In contrary, our retrieved SSA for coarse mode particles agrees well with AERONET SSA (59% and 87% observations within ±0.05 envelope at 466 nm and 646 nm respectively). Overall the regional average of AAE for dust aerosols observed here is consistent with insitu measurements (Bergstrom et al., 2004, 2007; Müller et al., 2009; Petzold et al., 2009) that report values ranging 2.0 to 3.5. Observations at individual sites (Table S1) show that the spectral dependence of the observed dust for few sites is relatively high than those reported by insitu measurements. Considering our retrieval method where aerosol absorption is derived independently for each wavelength and have computed the dependence, our results agree reasonably well with the insitu measurements reported in the literature.

Figure 17 shows the regional average AAE obtained for urban aerosols at three-wavelength pairs. Urban aerosols in highly polluted environments such as over the Mexico City have near unity spectral dependence. While the passage of biomass burning emissions over such environment show unusual decrease in absorption at the UV region attributing to high $AAE_{354-388}$. This is consistent with studies that report relatively high AAE in UV region and near unity in visible region for the aerosol mixture consisting of organic matter and black carbon amounts (Barnard et al., 2008; Martins et al., 2009; Bergstrom et al., 2010; Jethva and Torres, 2011). The urban aerosols found in our sample over Northern Indian and northeastern China are highly absorbing exhibiting $AAE_{340-646}$ ~1.5

than the carbonaceous aerosols with $AAE_{340\text{-}646}$ ~2. These results suggest the combination of magnitude of aerosol absorption and its spectral dependence in UV, visible and UV-Visible spectrum could be used to partition mixture of aerosol types found in such environments. Overall, the regional average UV-Visible AAE for the urban aerosols is found to be near 2.

## 8 Summary

Ground-based measurements of direct and diffuse solar radiation under cloud-free conditions over worldwide sites are providing valuable insights into regional aerosol characteristics. Long-term measurements obtained from such network, such as from AERONET, are widely used to develop regional aerosol climatology and investigate seasonal/annual variability, in addition to providing validation data set for the satellite-based AOD retrievals. Satellite measurements of TOA radiances are able to provide global distribution of columnar aerosol amounts.

However, deriving aerosol optical properties from satellite measurements require constraints on particle sizes and optical properties. Reliable aerosol measurements from ground-networks and airborne/field campaigns are traditionally used to validate and improve the constraints in satellite aerosol retrievals.

In this work, we use AERONET measured extinction $\tau$ as constraint in a robust inversion technique that uses

satellite measured TOA radiances from OMI and MODIS to derive spectral aerosol absorption in the UV-Vis part of the spectrum. Other than cloud contamination of the TOA radiances, major sources of error in our retrieved SSA come from surface reflectance, and aerosol layer height. We use TOA radiance observations with minimal or no cloud-contamination reported by both OMI and MODIS products. Sensitivity tests show that our retrieved aerosol SSA has reliable accuracy up to ±0.04 from UV-Visible wavelengths for absorbing aerosols

(carbonaceous and dust) with $\tau_{440} > 0.4$. However, for less-absorbing aerosols the error in SSA retrieval reaches up to ±0.07. Using a subset of results where SSA is retrieved independently for 340, 354, 388, 466 and 646 nm wavelengths for the same day with observations $\tau_{440} > 0.4$, we examine the seasonal variability in aerosol SSA and derive spectral dependence (AAE) at three wavelength pairs in the UV-Vis spectrum.

Key observations noted from the spectral aerosol absorption data set derived here are highlighted below:

**Carbonaceous aerosols**

    a)  Among sites dominated by biomass burning aerosols, Mongu in Southern Africa has high absorption $\omega_o$ ~0.85±0.02 and 0.84±0.05 at 340 nm and 646 nm, respectively.

b) Strong seasonality in absorption of carbonaceous aerosols is evident in Southern Africa indicating the role of biomass types and combustion process. The average $\omega_o$ noted at 340 nm and 646 nm during JJA (SON) are $\sim$0.85$\pm$0.02 (0.88$\pm$0.02) and 0.84$\pm$0.05 (0.87$\pm$0.05), respectively.

c) Carbonaceous aerosols found over Northern Australia are as strongly absorbing ($\omega_o$ $\sim$0.87$\pm$0.03 and 0.86$\pm$0.06 at 340 nm and 646 nm) as smoke over Southern Africa but has nearly invariant $\omega_o$ from UV-Vis spectra.

d) Carbonaceous aerosols found over Alta_Floresta in the Amazon Basin have similar absorption ($\omega_o$ $\sim$0.89$\pm$0.02 and 0.91$\pm$0.06 at 340 nm and 646 nm) and $AAE_{340-646}$ (1.8) to those found over Missoula in Northwestern US.

e) Highly absorbing carbonaceous aerosols with weak spectral dependence are found in Cairo ($\omega_o$ $\sim$0.89$\pm$0.02 and 0.91$\pm$0.06 at 340 nm and 646 nm) and Ilorin ($\omega_o$ $\sim$0.86$\pm$0.02 and 0.87$\pm$0.03 at 340 nm and 646 nm) during winter (DJF) in the Middle East and Sahel respectively.

f) Carbonaceous aerosols found over Northern India ($\omega_o$ $\sim$0.91$\pm$0.02 and 0.92$\pm$0.04 at 340 nm and 646 nm), Northeastern China ($\omega_o$ $\sim$0.87-0.90$\pm$0.03 and 0.90-0.94$\pm$0.05 at 340 nm and 646 nm), Sahel ($\omega_o$ $\sim$0.86$\pm$0.02 and 0.87$\pm$0.03 at 340 nm and 646 nm), and Middle East ($\omega_o$ $\sim$0.89$\pm$0.03 and 0.92$\pm$0.05 at 340 nm and 646 nm) has low average $\alpha_{440-870}$ ($<$ 1.4) than over other prominent biomass burning regions, suggesting mixture of fine and coarse modes.

g) Distinct seasonality in spectral absorption of carbonaceous aerosols is noted for Northeastern China. The maximum (minimum) absorption is noted during DJF (JJA) exhibiting $\omega_o$ $\sim$0.87$\pm$0.02 and 0.90$\pm$0.05 at 340 nm and 646 nm ($\omega_o$ $\sim$0.91$\pm$0.02 and 0.94$\pm$0.04 at 340 nm and 646 nm), respectively.

**Dust aerosols**

a) For desert dust aerosols, the SSA is known to increase with wavelength from UV to Visible spectrum. No distinct seasonality in SSA is noted. The regional average of SSA for dust aerosols from 340 nm to 646 nm are 0.87$\pm$0.02 to 0.98$\pm$0.02, 0.89$\pm$0.03 to 0.96$\pm$0.02, 0.89$\pm$0.02 to 0.98$\pm$0.01, 0.87$\pm$0.02 to 0.97$\pm$0.02, and 0.87$\pm$0.03 to 0.95$\pm$0.03 over Sahara, Sahel, Middle East, Northern India, Northeastern China, respectively.

b) Among the dust dominated regions considered here, our retrievals indicate relatively high absorption ($\omega_o$ $\sim$0.93$\pm$0.03 at 466 nm) for aerosols noted at Sahel and northeastern China, while those noted over MiddleEast, and Northern India ($\omega_o$ $\sim$0.97$\pm$0.03 at 466 nm) are less absorbing and dust over Sahara ($\omega_o$ $\sim$0.95$\pm$0.02 at 466 nm) shows intermediate absorption. These results can be explained by combination of

varying mineral composition (iron oxide amounts) and mixing of dust with other sources along the transport pathway.

**Urban aerosols**

a)  Urban aerosols ($\omega_o$ ~0.87±0.04 and 0.84±0.08 at 340 nm and 646 nm) are highly absorbing and exhibit
distinct seasonality (higher absorption during JJA than SON) at Sao Paulo, South America. The urban aerosols noted here are likely mixtures of carbonaceous particles transported over the region and prevailing pollution from local sources.

b)  Polluted aerosols observed over Mexico City show high absorption in UV extending to visible spectrum during DJF ($\omega_o$ ~0.88±0.04 and 0.84±0.09 at 340 nm and 646 nm) and SON ($\omega_o$ ~0.88±0.04 and
0.80±0.08 at 340 nm and 646 nm) months.

c)  Polluted aerosols noted over the Cairo and Nes Ziona in the Middle East are highly absorbing and exhibit weak spectral dependence ($\omega_o$ ~0.87-0.89±0.03 and 0.87-0.90±0.07 at 340 nm and 646 nm).

d)  Polluted aerosols noted over the Jaipur, Kanpur and Gandhi College in the Northern India are highly absorbing and exhibit weak spectral dependence ($\omega_o$ ~0.88±0.03 and 0.87±0.07 at 340 nm and 646 nm).

As mentioned, the results presented here are limited to our subset of retrievals, where SSA is retrieved for all five wavelengths from UV-Visible range with $\tau_{440} > 0.4$. Since one of our objectives is to derive UV-Visible spectral dependence (AAE) of aerosols, and given the inherent sampling bias of the OMI, MODIS and AERONET colocations – the analysis method employed here is well justified. In other words, we made use of the best
available data synergy and derive unique aerosol absorption data set with no prior assumptions on wavelength dependence, which otherwise is assumed in the standard satellite-based aerosol retrieval algorithm. It should be noted that our results may be biased toward dense pollution/industrial, smoke, and dust events. In addition, the regional aerosol absorption derived here may not be representative of the entire region due to limited sampling and fewer sites used. However, these absorption models offer essential guidance for selecting spectral absorption
in satellite aerosol retrievals using UV (OMI), Vis (MODIS), and even spanning the UV-Vis spectrum, such as planned under the upcoming PACE mission. From our analysis of worldwide inland sites: (a) it is suggested that satellite aerosol retrieval techniques could employ regional dynamic absorption models to avoid potential bias in $\tau$ retrievals noted in earlier studies, and (b) the spectral dependence of aerosol absorption noted here for the UV (354-388 nm), visible (466-646 nm) and UV-Visible (340-646 nm) range for all aerosol types other than black

carbon varies considerably. Overall, the UV absorption data set well compliments and provides more information on the regional aerosol absorption than with the visible data set alone.

Given the lack of aerosol absorption information at near-UV wavelengths in the currently existing AERONET inversion products and limited availability of insitu measurements, the UV-Vis aerosol absorption data set developed here, perhaps for the first time, offers a valuable source of information useful for a variety of aerosol and trace gas studies. As mentioned earlier the newer models of AERONET sunphotometers include sky radiance measurements at 380 nm, and the derived SSA at this near UV wavelength is expected in the future upgrade of AERONET inversion product. The derived spectral dependency can be used with either subset of the results or all SSA retrievals to construct and investigate long-term trends in UV-Visible aerosol absorption. Further, the spectral aerosol SSA derived here could be used to parameterize absorption in models and better understand the radiative effects of aerosols. Our ongoing investigation utilizing the complete data set developed here will explore some of these applications in the future.

### Acknowledgments

NASA ROSES (ACMAP) – 2016 provided financial support for this work under the grant NNH16ZDA001N. The authors are grateful to Brent Holben and the entire AERONET team for their efforts on maintaining AERONET sites worldwide and providing quality assured data to the community. The authors thank all PIs and Co-PIs of the individual AERONET sites that were used in this work.

### Competing Interests

The authors declare no conflict of interests.

### Author Contributions

Omar Torres (OT) and Hiren Jethva (HJ) had conceptualized the research. Vinay Kayetha (VK) developed the data set, performed formal analysis, and wrote the manuscript with inputs from OT and HJ. All authors reviewed results, helped with the data interpretation, and edited the manuscript to make a final version.

**Data Availability**

The spectral aerosol absorption data set developed here will be made available upon request to the authors.

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

## Tables

**Table 1: Description of the ground and satellite data products used in this work.**

| Purpose | Instrument | Product | Level & Version | Parameter(s) |
|---|---|---|---|---|
| For SSA retrieval in this work | AERONET | AOD | L2, V3 | AOD and extinction Ångström exponent. |
| | | Inversion | L2, V2 | Particle size distributions, and real part of refractive index at 440 nm. |
| | OMI | OMLERWAVE, OMAERUV | L2, V1.8.9.1 | TOA reflectances (with QFs), Aerosol type, LER, Aerosol layer height obtained from CALIPSO. |
| | MODIS | MYD04 | L2, C006 | TOA reflectances (with QFs) provided by Deep-Blue algorithm. |
| | | MAIAC - MCD19A1 | L2, C006 | Surface reflectance at 466 and 646 nm. |
| Comparison | AERONET | Inversion | L2, V3 | SSA at 440 and 675 nm. |

**Table 2: Theoretical estimated uncertainties in the retrieval of aerosol SSA due to error in the input variables. Configuration of sensitivity tests are: SZA = 20º, VZA = 40º, RAA = 130º, and $\omega_0(388) = 0.9$.**

| | Input Uncertainty | Theoretical SSA Uncertainty ($\Delta\omega_0$) for $\tau_{440} = 0.4$ | | | | | |
|---|---|---|---|---|---|---|---|
| | | $\lambda = 340$ nm | | | $\lambda = 646$ nm | | |
| | | Carb. | Dust | Urban | Carb. | Dust | Urban |
| Extinction AOD | $\lambda < 400$ nm, $\Delta\tau = \pm0.02$ $\lambda > 400$ nm, $\Delta\tau = \pm0.01$ | 0.002 | 0.001 | 0.002 | 0.009 | 0.007 | 0.011 |
| Particle sizes | $\Delta VMR = \pm20\%$ | 0.018 | 0.003 | 0.014 | 0.044 | 0.0006 | 0.040 |
| Real part of RI. | $\Delta RRI = \pm0.04$ | 0.007 | 0.007 | 0.009 | 0.001 | 0.002 | 0.002 |
| Calibration of TOA measurements | OMI = $\pm1.8\%$ MODIS = $\pm1.9\%$ | 0.026 | 0.021 | 0.027 | 0.020 | 0.027 | 0.037 |
| Surface reflectance | $\Delta\rho_{surf} = \pm0.01$ | 0.006 | 0.011 | 0.006 | 0.032 | 0.022 | 0.050 |
| Aerosol layer hgt. | $\Delta ALH = \pm1$ km | 0.021 | 0.028 | 0.006 | 0.001 | 0.001 | 0.0006 |
| Presence of cloud | $\tau_{cloud} = 0.5$ | 0.016 | 0.020 | 0.017 | 0.042 | 0.041 | 0.056 |
| Surface pressure | $\pm12$ mb/hPa | 0.011 | 0.011 | 0.011 | 0.004 | 0.0004 | 0.006 |
| Variability in AOD | $\lambda < 400$ nm, $\Delta\tau = \pm0.2$ $\lambda > 400$ nm, $\Delta\tau = \pm0.1$ | 0.022 | 0.014 | 0.012 | 0.006 | 0.001 | 0.053 |
| $O_3$ absorption | $\pm50$ DU | 0.003 | 0.005 | 0.003 | 0.008 | 0.007 | 0.011 |
| $NO_2$ absorption | $\pm1$ DU | 0.016 | 0.023 | 0.014 | 0.001 | 0.0007 | 0.001 |
| **Max. combined theoretical uncertainty** | | **0.051** | **0.047** | **0.043** | **0.073** | **0.055** | **0.108** |

**Table 3: Theoretical uncertainties in the computation of Absorption Ångström Exponent.**

| | | $\Delta\omega_o$ | $\omega_o$ overestimation | | | $\omega_o$ underestimation | | |
|---|---|---|---|---|---|---|---|---|
| | | | $\Delta AAE_{354-388}$ | $\Delta AAE_{466-646}$ | $\Delta AAE_{340-646}$ | $\Delta AAE_{354-388}$ | $\Delta AAE_{466-646}$ | $\Delta AAE_{340-646}$ |
| Carbon. | $\tau_{440} = 0.4$ | 0.01 | -0.022 | -0.021 | -0.021 | 0.018 | 0.017 | 0.018 |
| | $EAE_{340-646} = 1.9$ | 0.02 | -0.051 | -0.046 | -0.048 | 0.034 | 0.031 | 0.032 |
| | $\omega_o(388) = 0.90$ | 0.03 | -0.087 | -0.078 | -0.081 | 0.046 | 0.044 | 0.045 |
| | $AAE_{340-646} = 1.7$ | 0.04 | -0.135 | -0.119 | -0.126 | 0.058 | 0.054 | 0.056 |
| Dust | $\tau_{440} = 0.4$ | 0.01 | 0.228 | 0.688 | 0.488 | -0.190 | -0.423 | -0.325 |
| | $EAE_{340-646} = 0.2$ | 0.02 | 0.506 | 2.067 | 1.369 | -0.351 | -0.711 | -0.562 |
| | $\omega_o(388) = 0.90$ | 0.03 | 0.854 | 8.769 | 5.024 | -0.489 | -0.922 | -0.744 |
| | $AAE_{340-646} = 2.5$ | 0.04 | 1.302 | - | - | -0.609 | -1.082 | -0.889 |
| Urban | $\tau_{440} = 0.4$ | 0.01 | -0.117 | -0.077 | -0.092 | 0.095 | 0.066 | 0.077 |
| | $EAE_{340-646} = 1.9$ | 0.02 | -0.265 | -0.166 | -0.204 | 0.173 | 0.124 | 0.144 |
| | $\omega_o(388) = 0.90$ | 0.03 | -0.458 | -0.272 | -0.343 | 0.239 | 0.175 | 0.201 |
| | $AAE_{340-646} = 0.9$ | 0.04 | -0.722 | -0.399 | -0.522 | 0.295 | 0.221 | 0.250 |

**Figures**

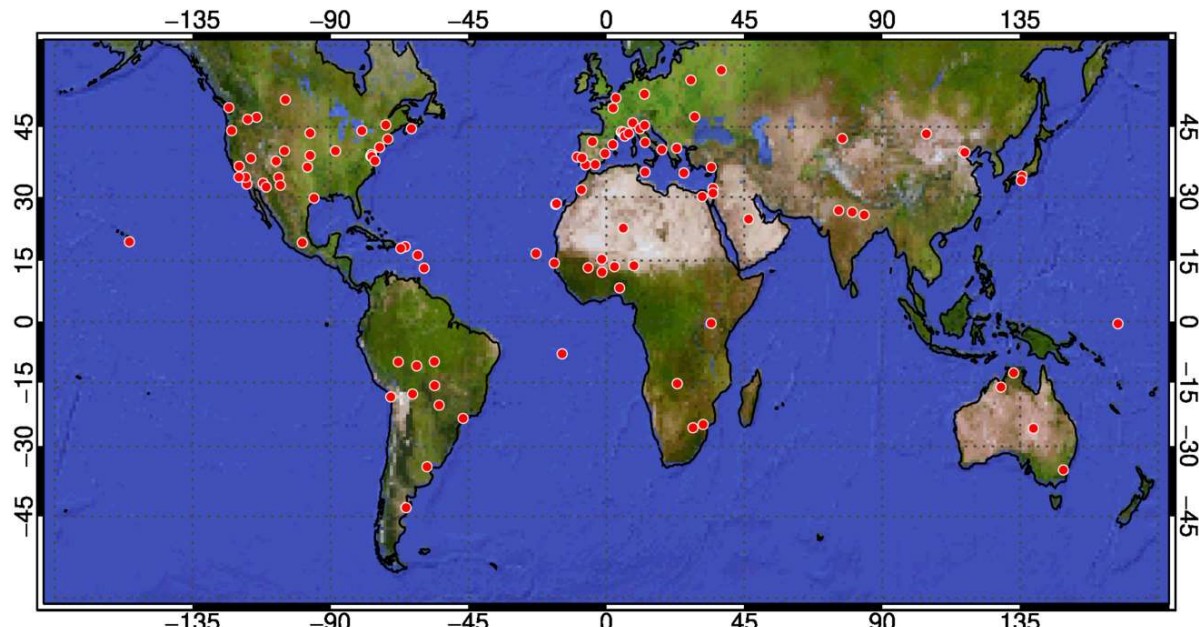

Figure 1: The geo-distribution of AERONET sites whose AOD data are used for the retrieval of spectral aerosol single scattering albedo in this work.

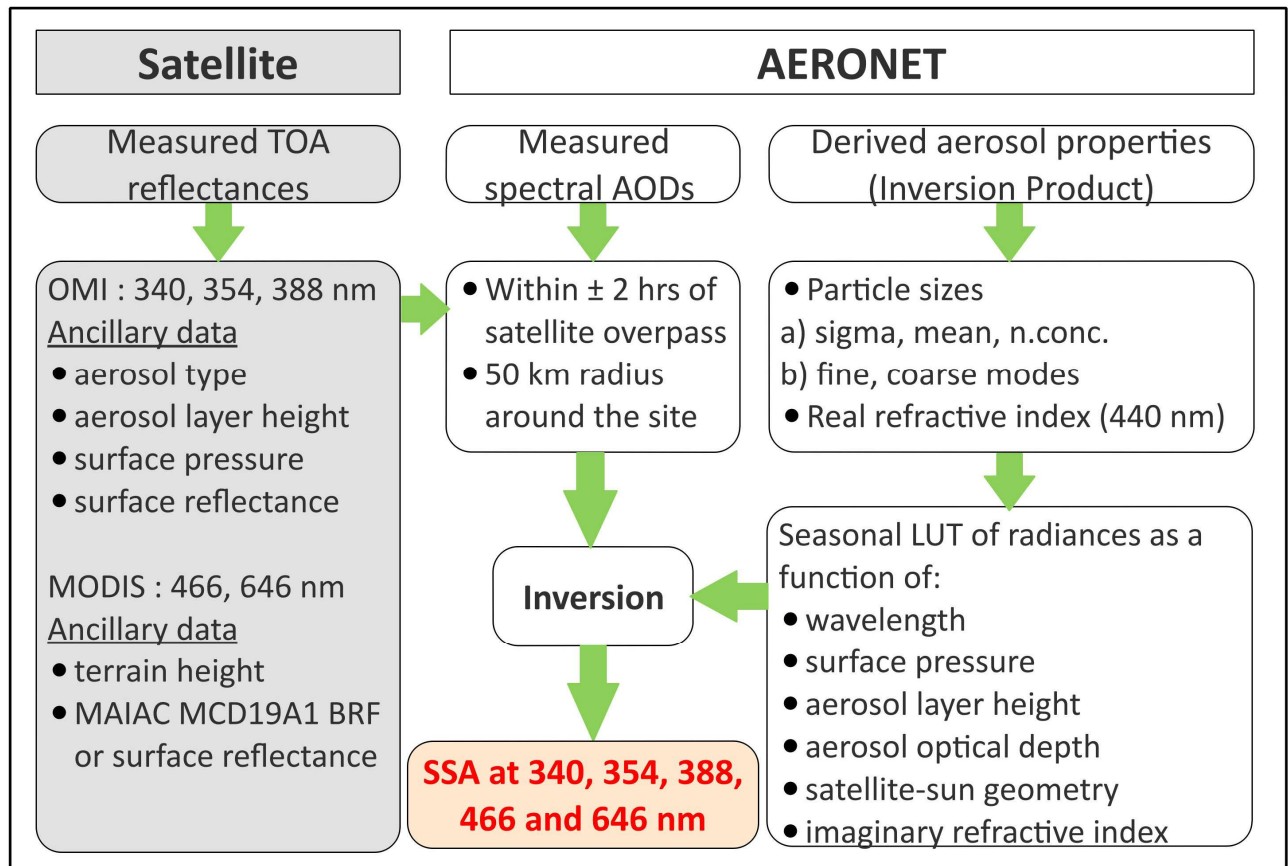

**Figure 2: Schematic flow chart of the methodology used to retrieve aerosol spectral single scattering**
**albedo.**

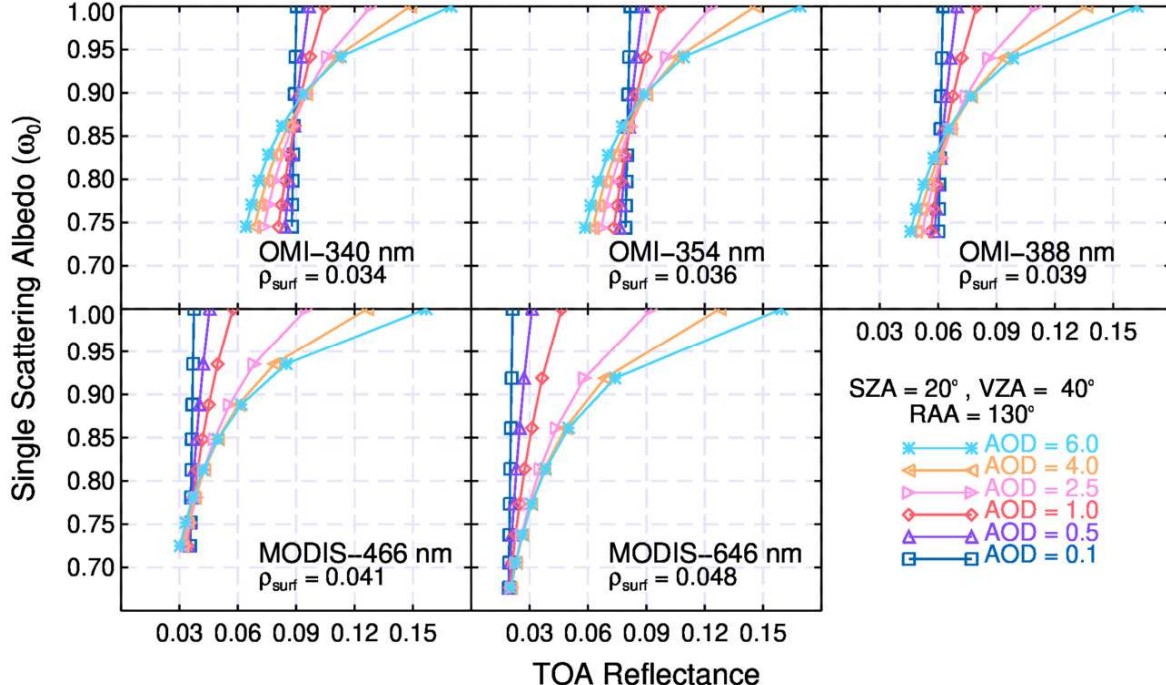

**Figure 3: Simulated TOA radiances for the aerosols over the GSFC site as a function of $\tau$ and SSA in the**
**UV-Visible range.**

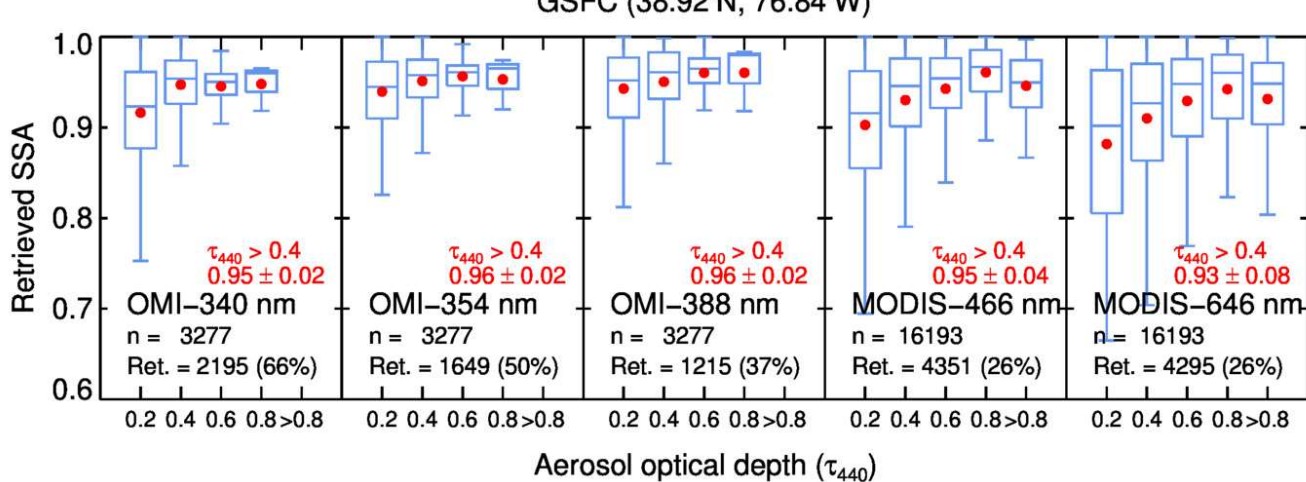

**Figure 4: Retrieved aerosol SSA over the GSFC site for the satellite observations period of 2005 – 2016.**
**Average and standard deviation of retrieved SSA for the observations with $\tau_{440} > 0.4$ is shown in red.**
**Abbreviations used: n is the number of satellite-AERONET collocated observations, and ret. is the number**
**of observations for which SSA is retrieved.**

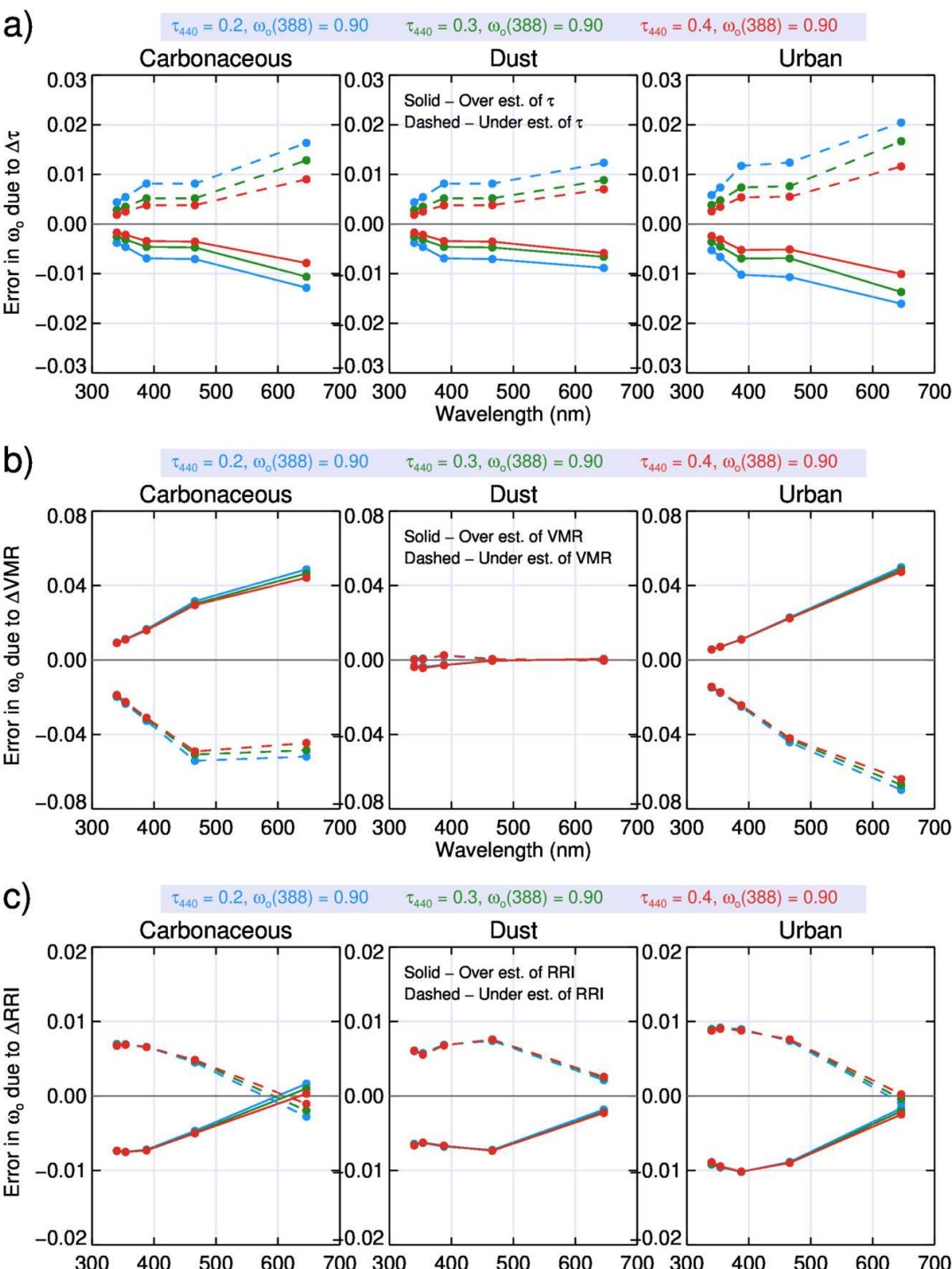

**Figure 5: Theoretical uncertainty in SSA retrievals due to changes in (a) ±0.02 (λ < 400 nm) and ±0.01 (λ > 400 nm) τ, (b) ±20% volume mean radius (VMR), and (c) ±0.04 real part of refractive index (RRI). Solid and dashed lines represent the over estimation and under estimation of the corresponding input variable.**

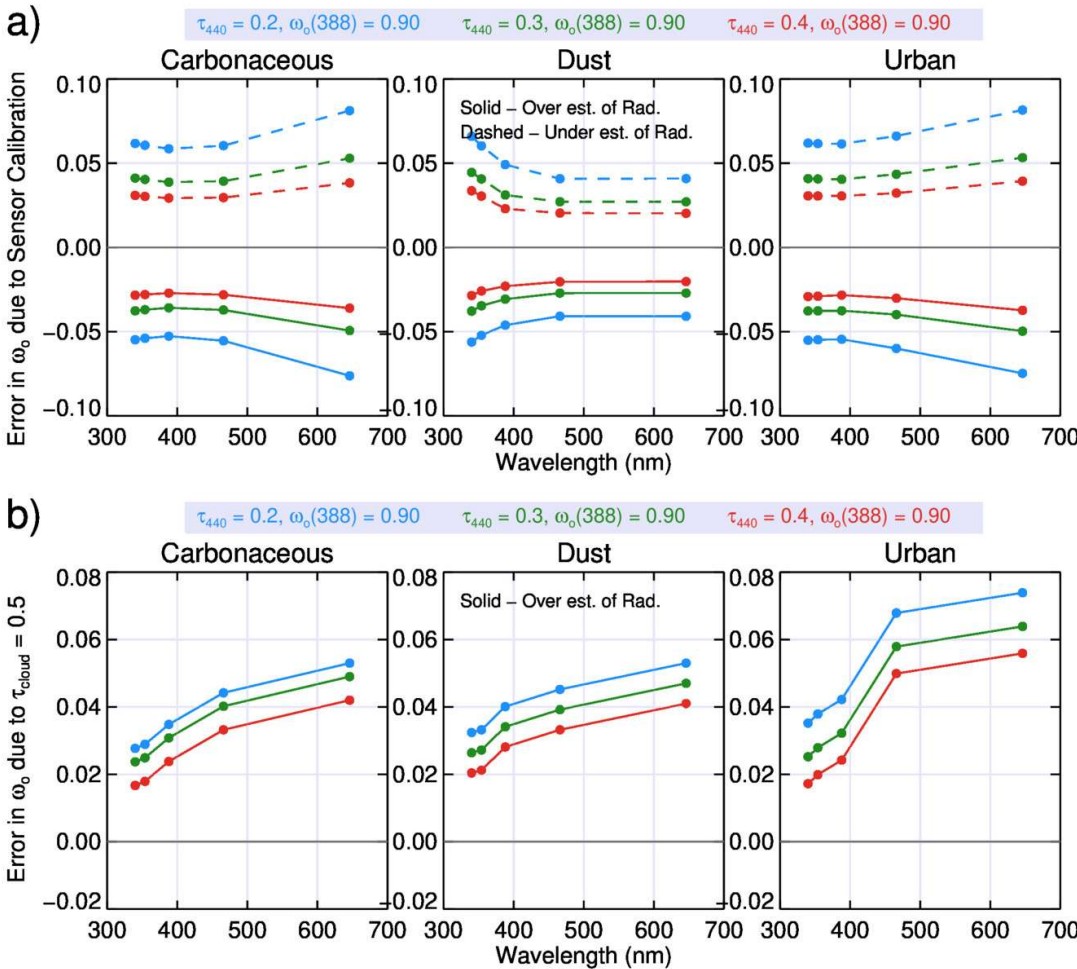

Figure 6: Theoretical uncertainty in SSA retrievals due to (a) ±1.8% (OMI) and ±1.9% (MODIS) sensor calibration, and (b) cloud contamination (τ_cloud = 0.5). Solid and dashed lines represent the over estimation and under estimation of the corresponding input variable.

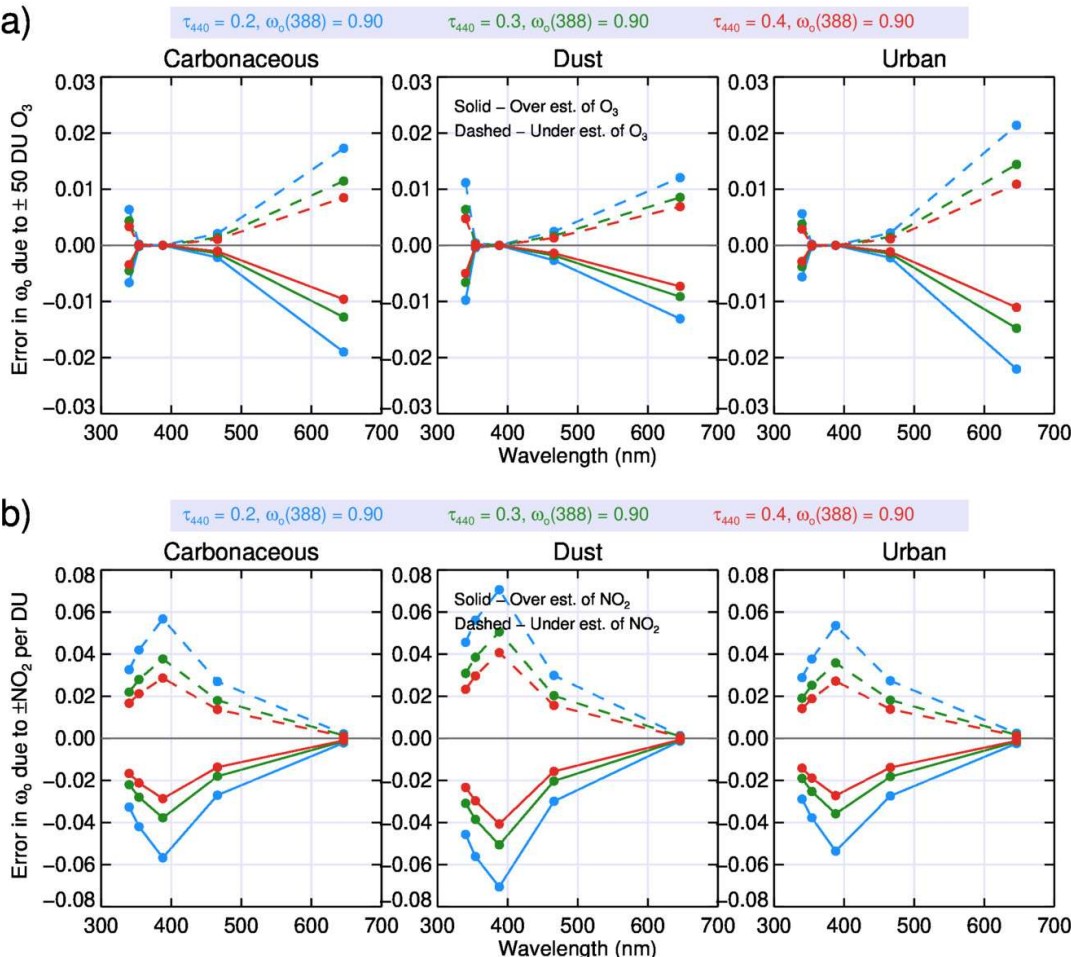

**Figure 7: Theoretical uncertainty in SSA retrievals due to (a) ±50 DU O₃, and (b) ±1 DU NO₂**
**concentrations. Solid and dashed lines represent the over estimation and under estimation of the**
**corresponding input variable.**

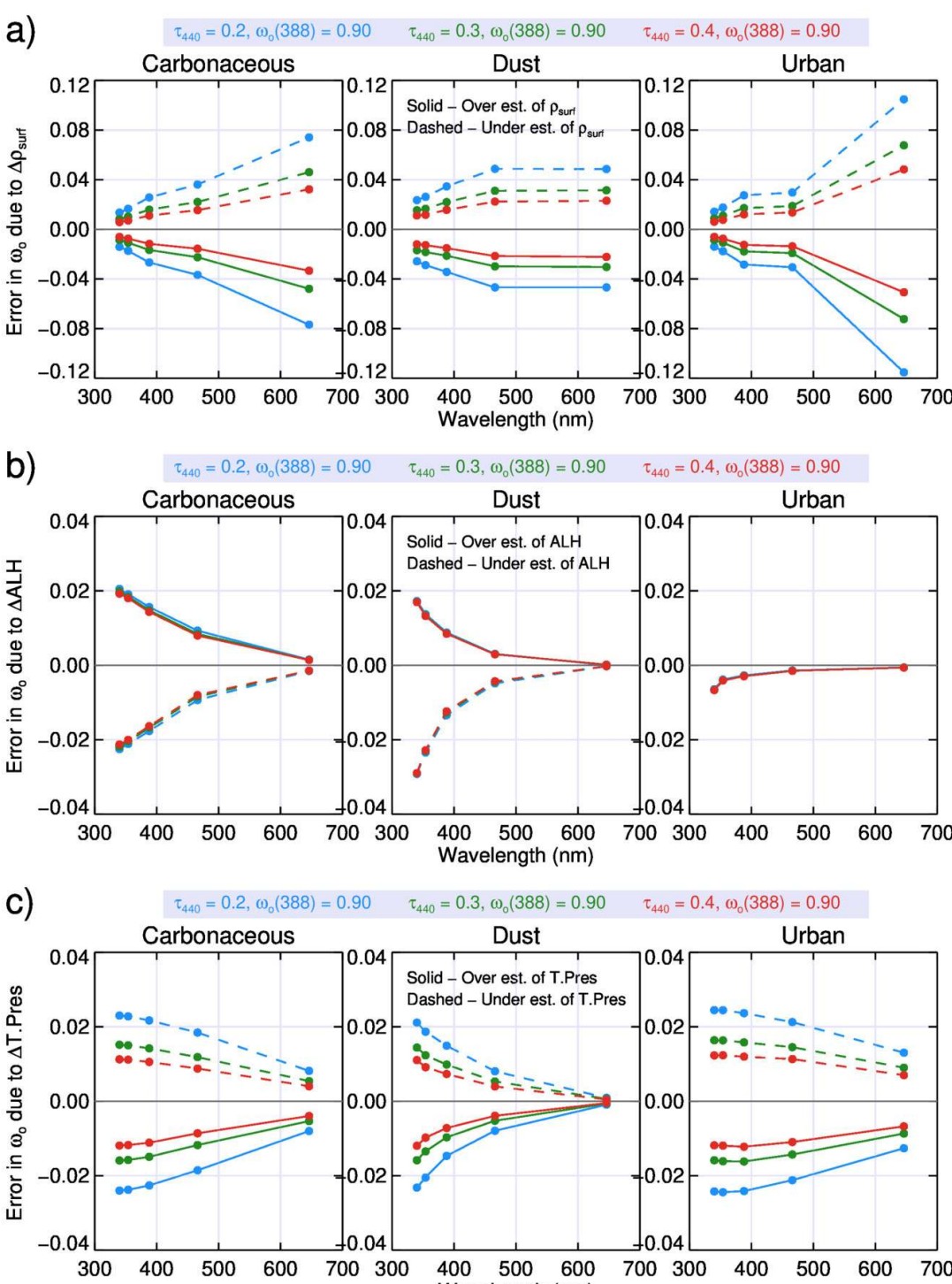

**Figure 8: Theoretical uncertainty in SSA retrievals due to changes in (a) ±0.01 surface reflectance, (b) ±1**
**km ALH, and (c) ±12 hPa surface pressure. Solid and dashed lines represent the over estimation and**
**under estimation of the corresponding input variable.**

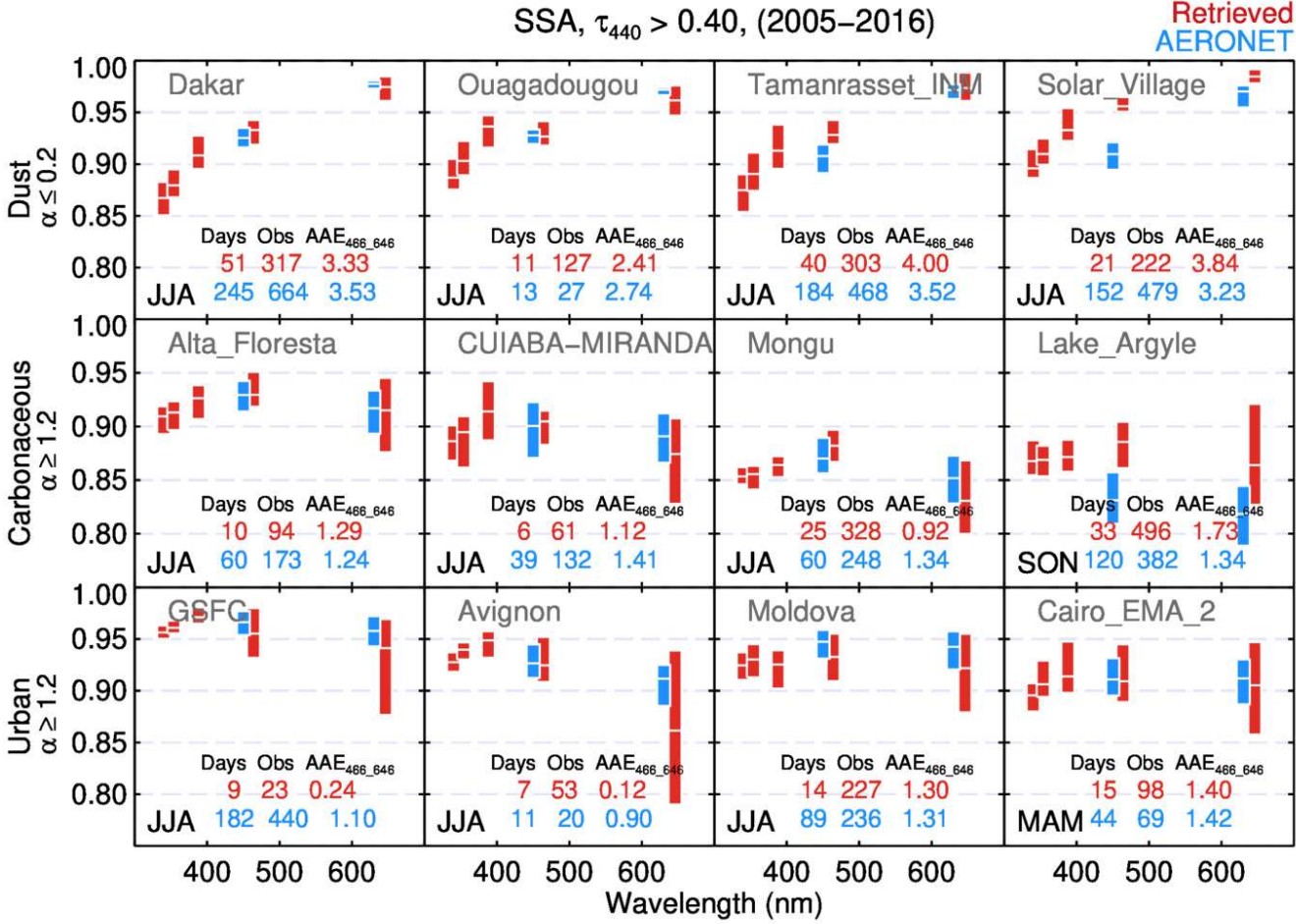

**Figure 9: Comparison of retrieved aerosol SSA with that of AERONET for selected sites. Box plots here represent lower and upper quartile of observations with mean values shown as white line.**

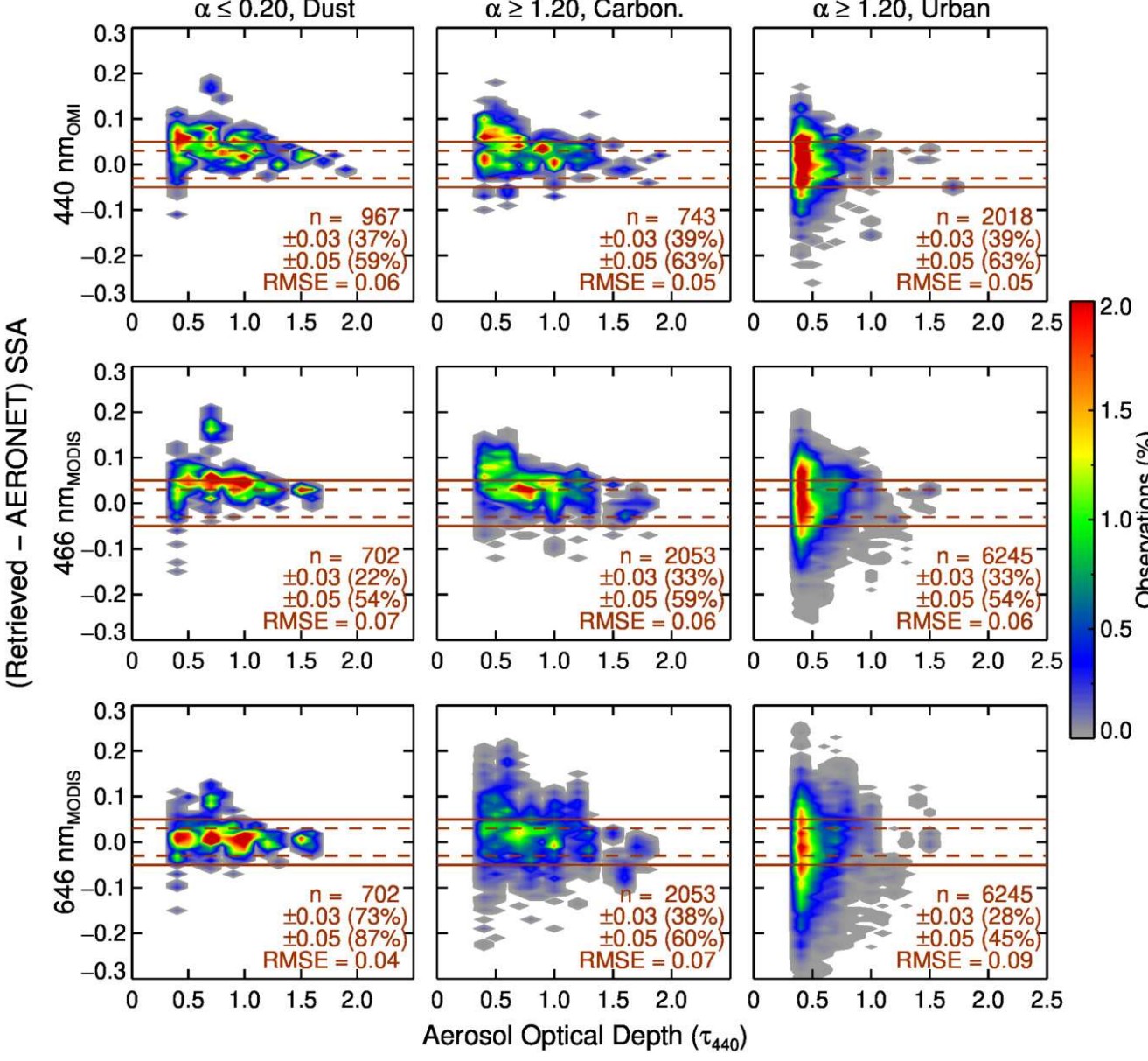

**Figure 10: Absolute difference in retrieved minus AERONET SSA versus AOD for all observations with**
$\tau_{440} > 0.4$ **for coarse-mode dust, fine-mode carbonaceous, and fine-mode urban aerosols at 440 nm, 466 nm,**
**and 646 nm.**

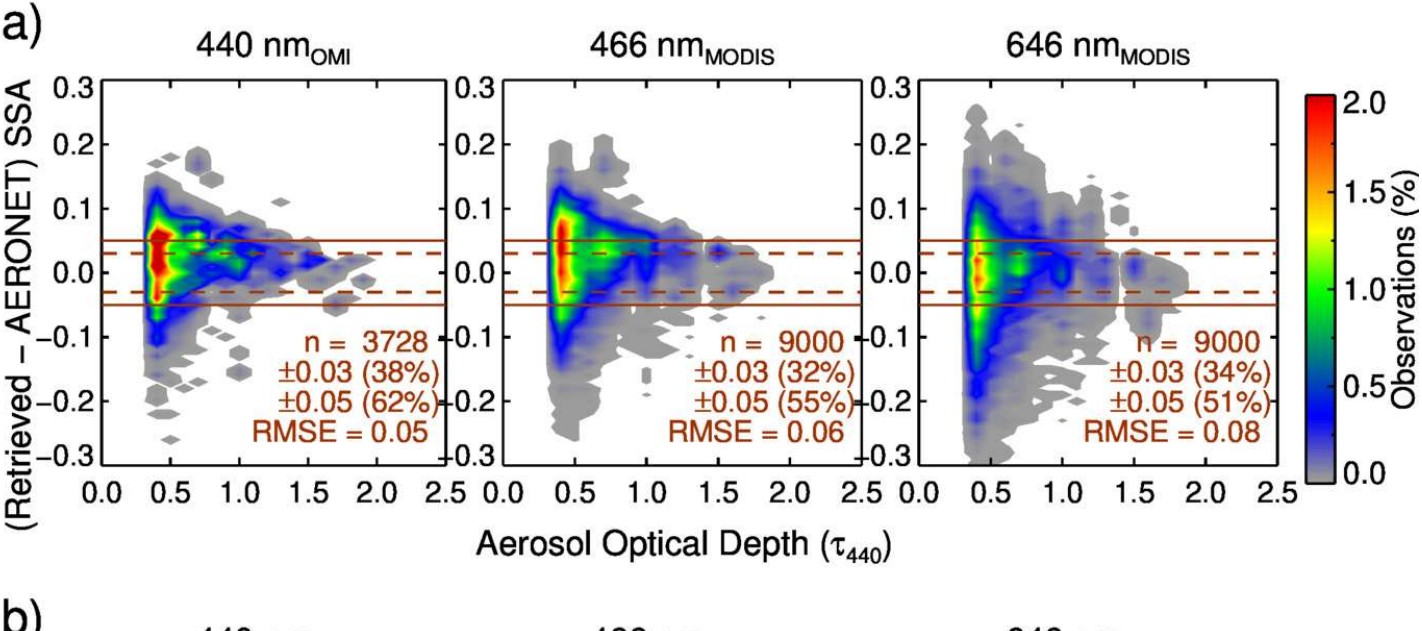

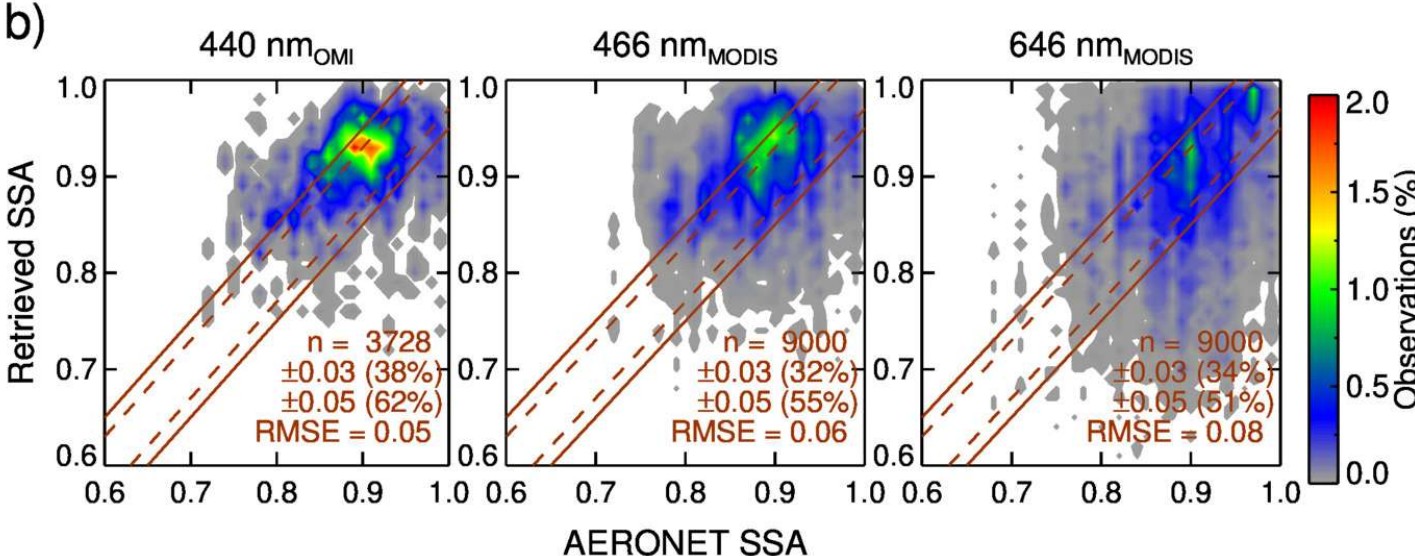

**Figure 11: (a) Absolute difference in retrieved minus AERONET SSA versus $\tau_{440}$, and (b) Retrieved SSA**
**versus AERONET SSA for observations with $\tau_{440} > 0.4$ for the combined aerosol types at 440 nm, 466 nm**
**and 646 nm.**

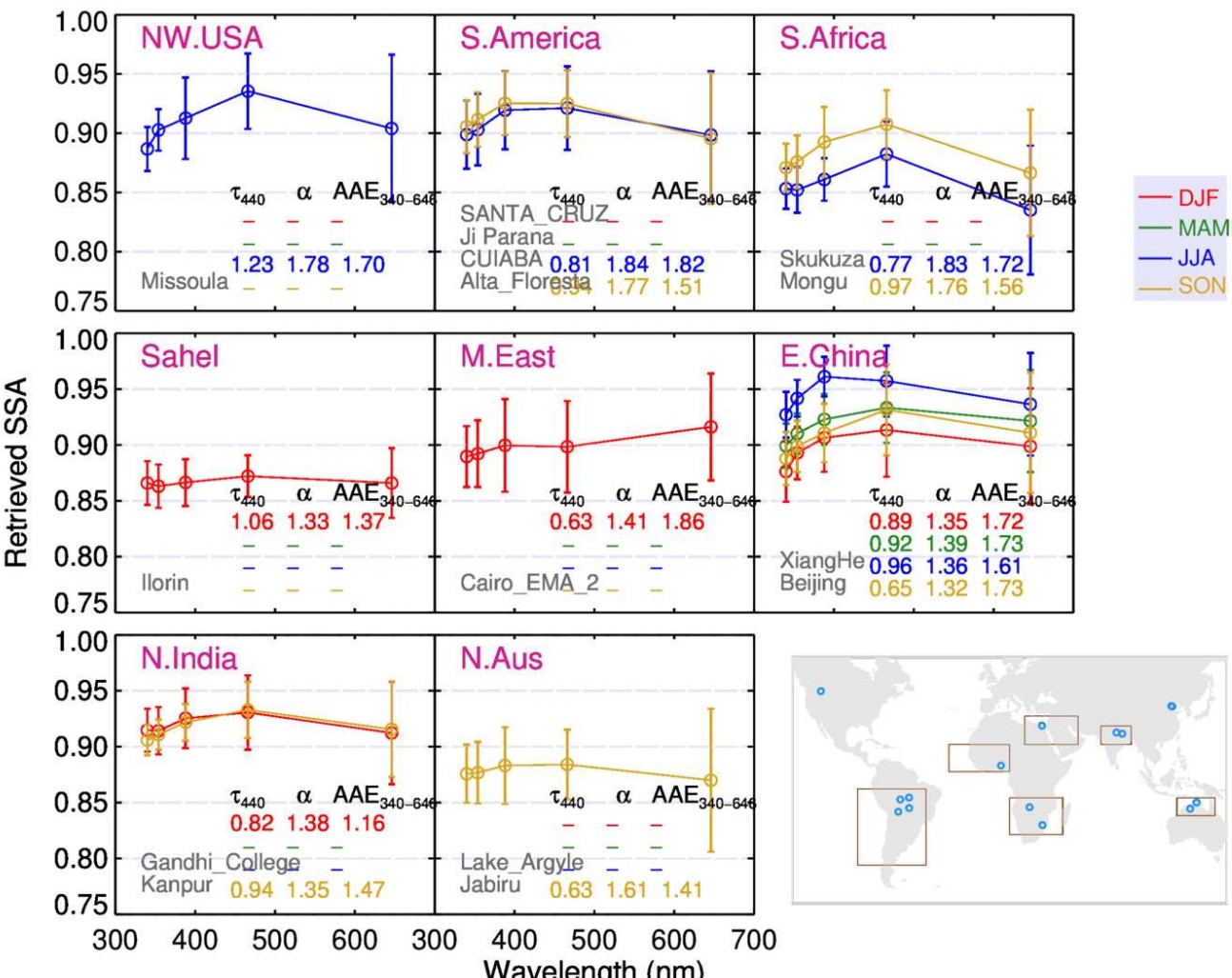

Figure 12: Seasonal average of spectral aerosol SSA derived for observations with $\tau_{440} > 0.4$ over regions dominated by fine-mode carbonaceous aerosols. The error bars represent the standard deviation of the observations.

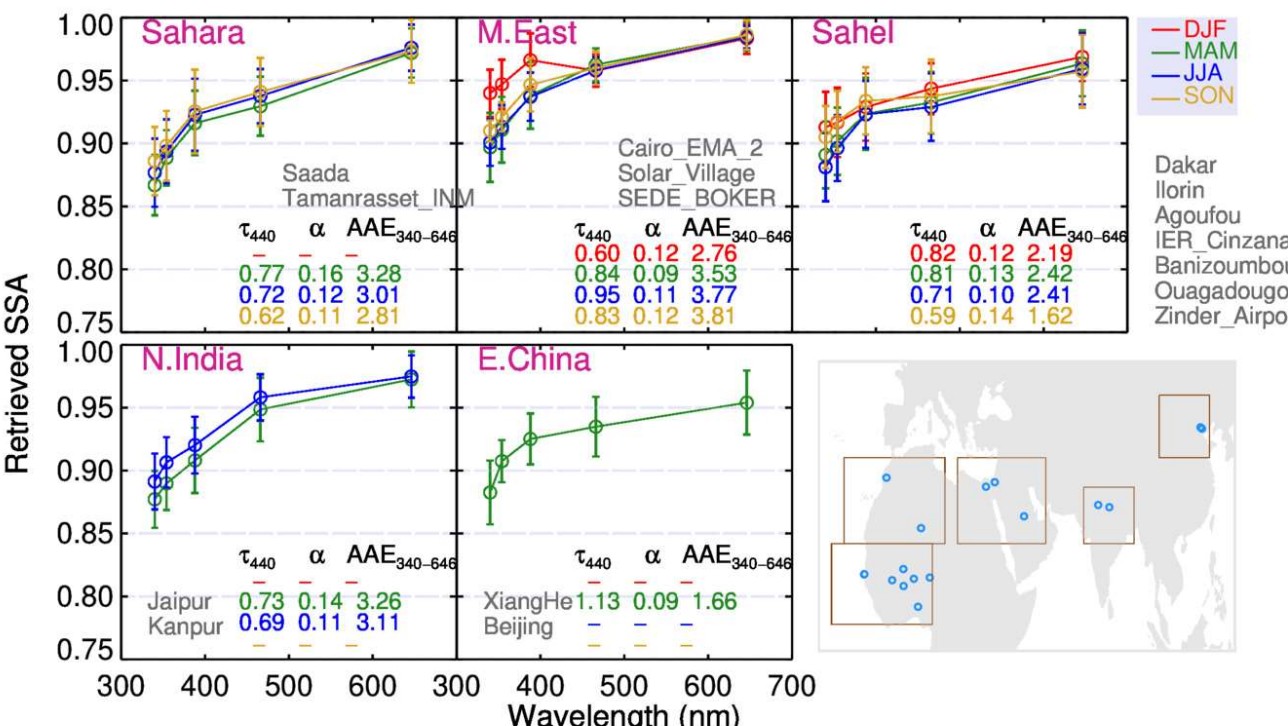

Figure 13: Seasonal average of spectral aerosol SSA derived for observations with $\tau_{440} > 0.4$ over regions dominated by coarse-mode dust aerosols. The error bars represent the standard deviation of the observations.

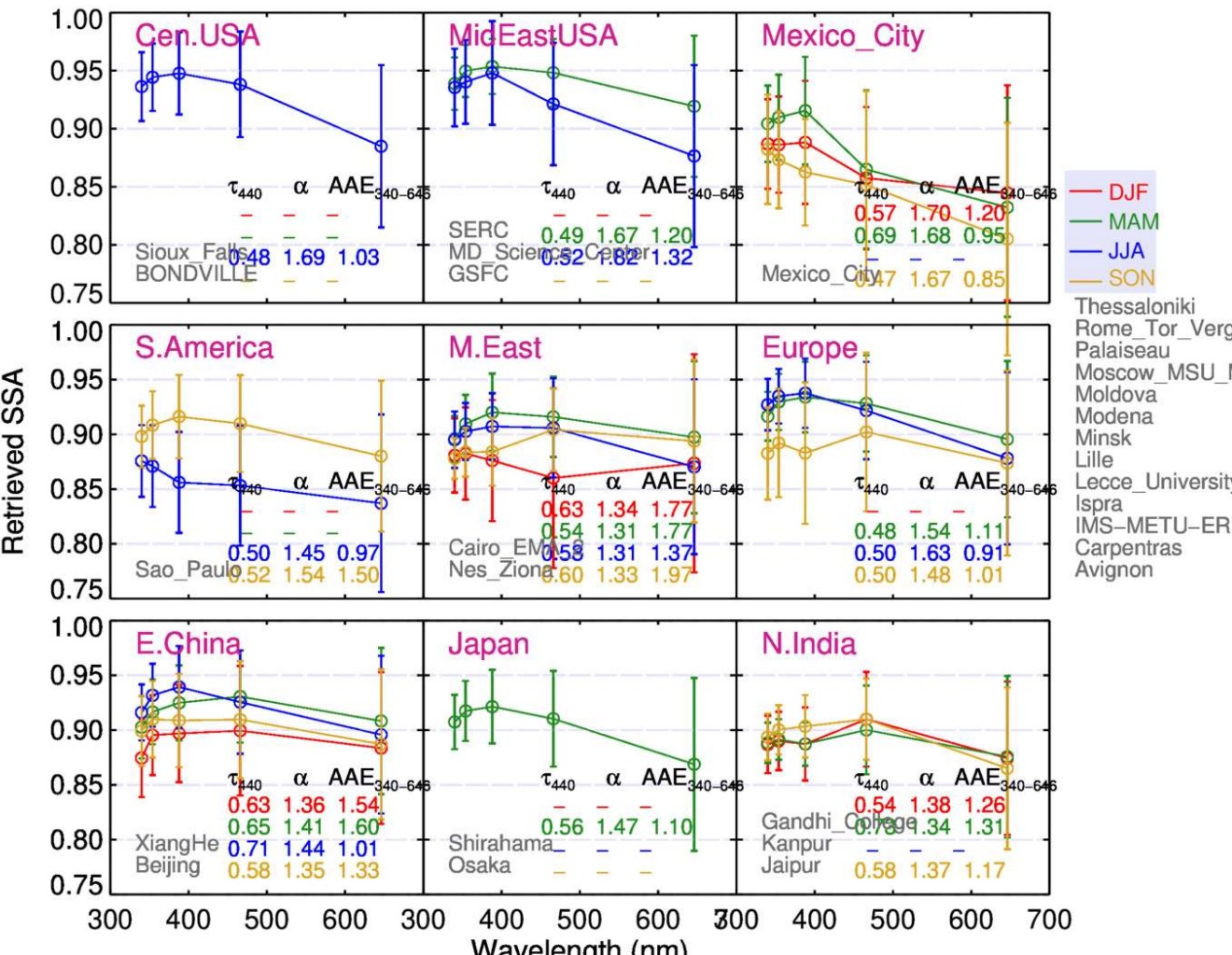

**Figure 14: Seasonal average of spectral aerosol SSA derived for observations with $\tau_{440} > 0.4$ over regions**
**dominated by fine-mode urban or mixture of aerosols. The error bars represent the standard deviation of**
**the observations.**

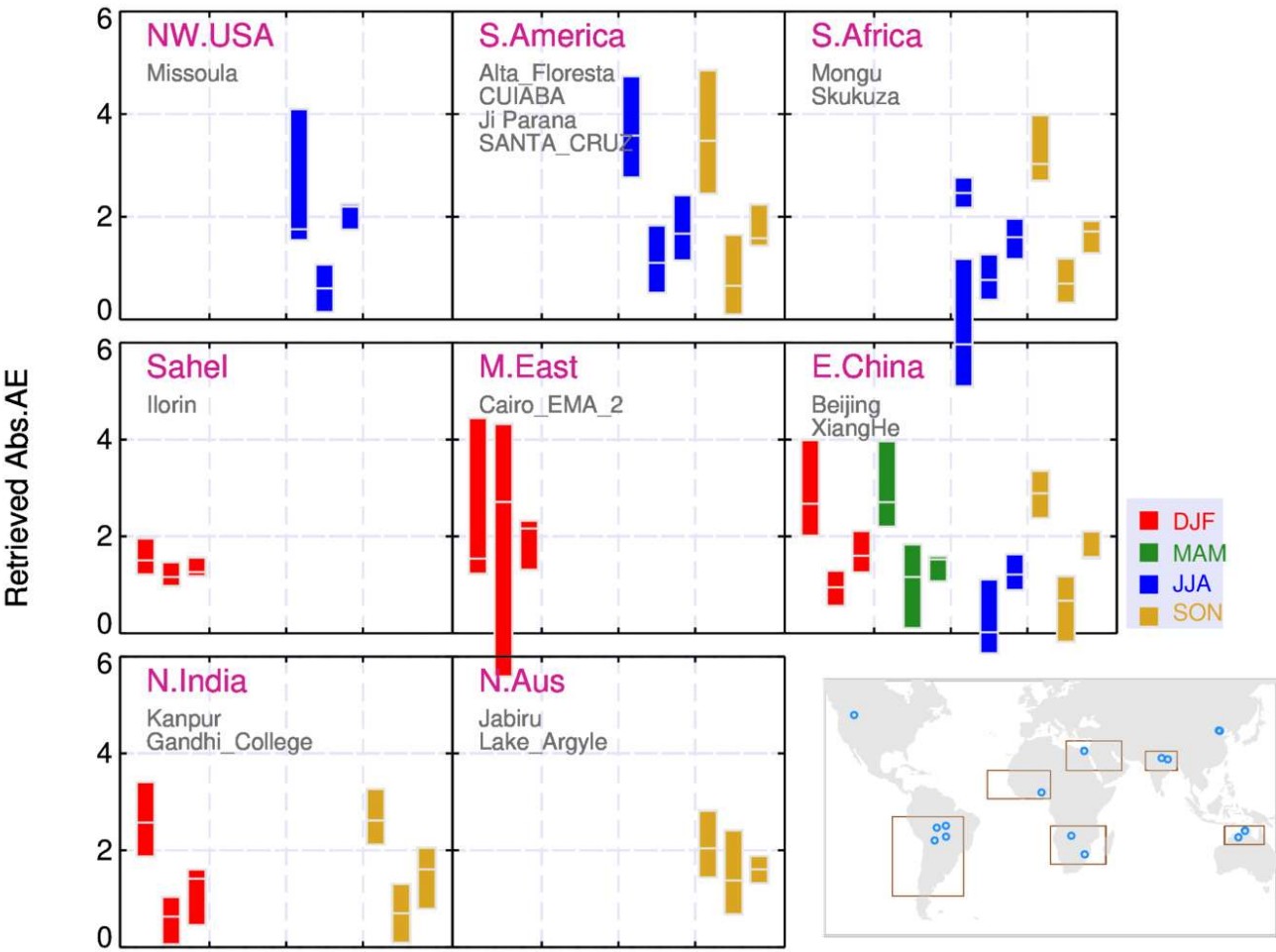

Figure 15: Absorption Angstrom exponent derived at three wavelength pairs for observations with $\tau_{440} >$ 0.4 for the fine-mode carbonaceous aerosols. Boxplot represents lower and upper quartile of observations with mean as white line.

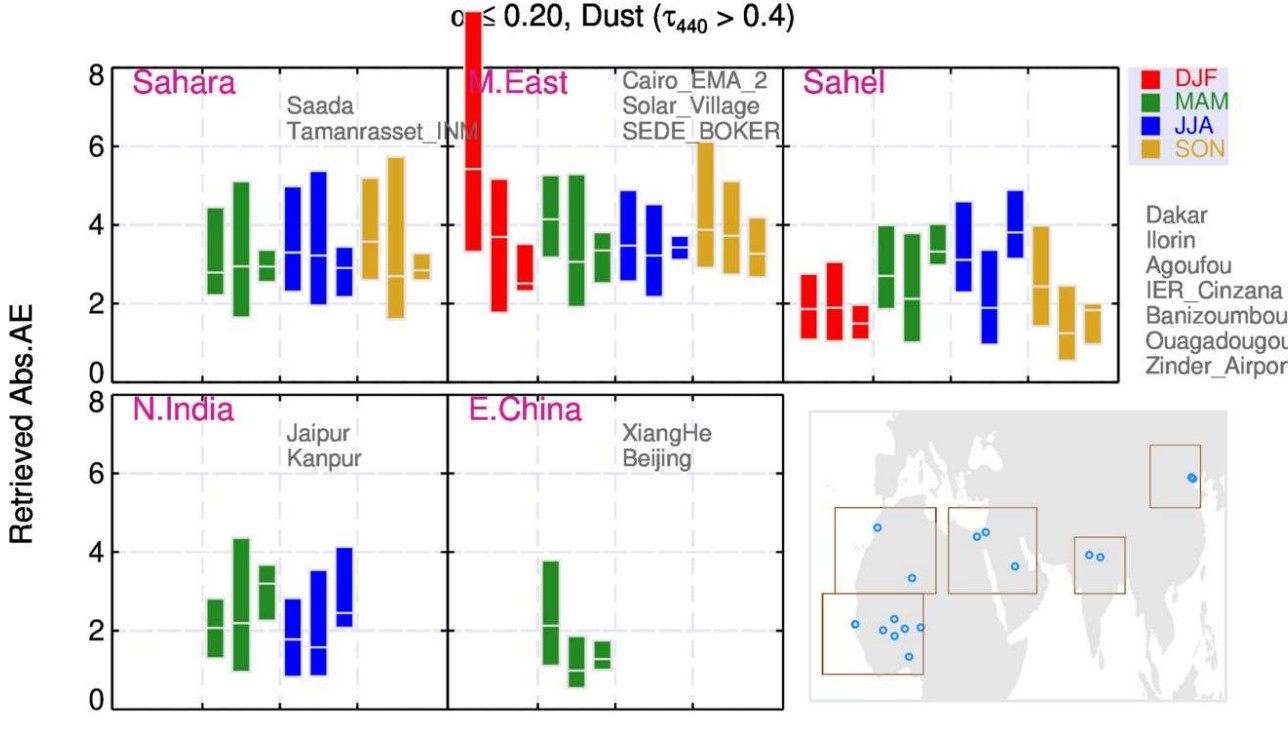

Wavelength Pairs (354,388),(466,646),(340,646)

**Figure 16: Absorption Angstrom exponent derived at three wavelength pairs for observations with $\tau_{440}$ >**
**0.4 for the coarse-mode dust aerosols. Boxplot represents lower and upper quartile of observations with**
**mean as white line.**

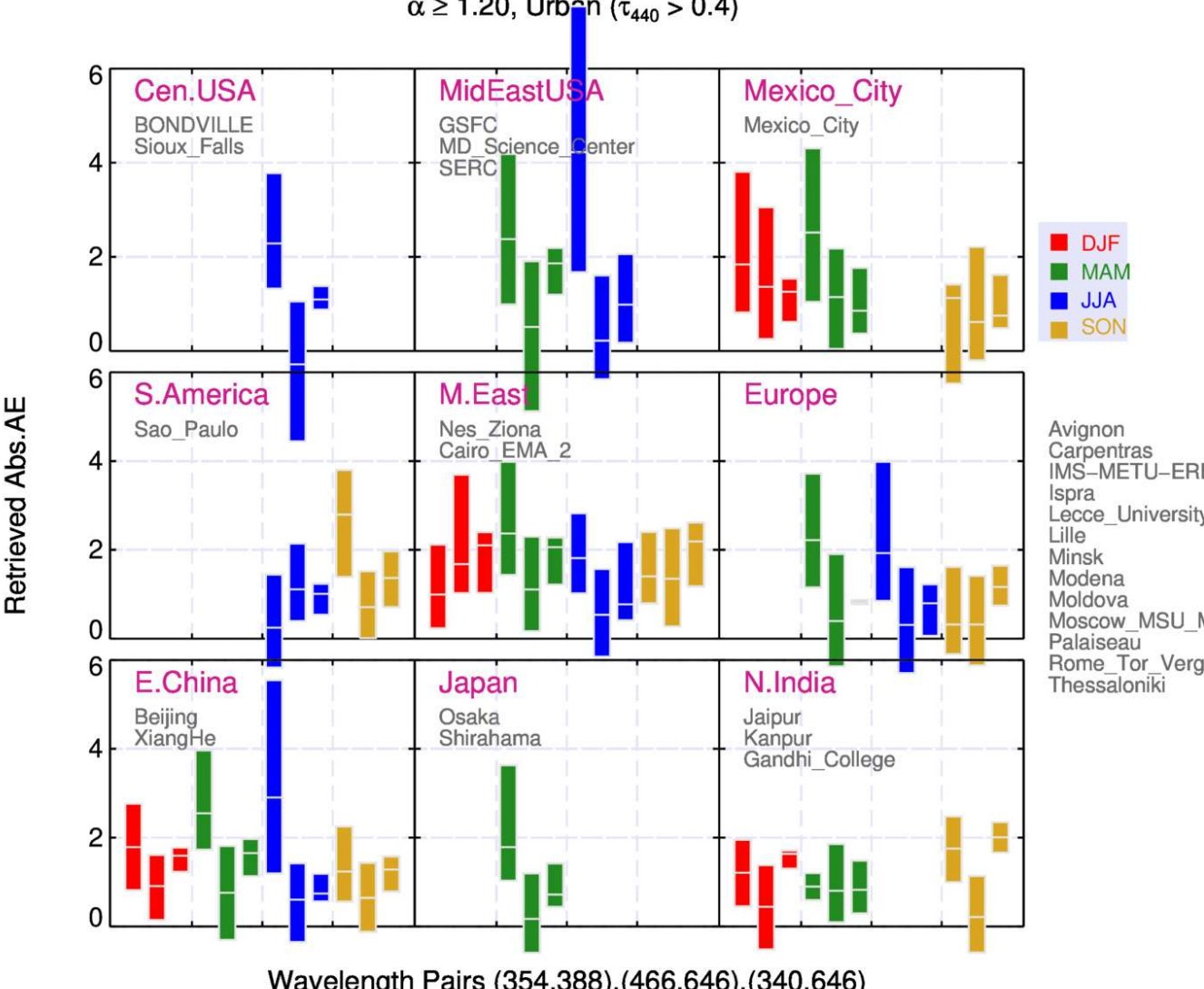

**Figure 17: Absorption Angstrom exponent derived at three wavelength pairs for observations with $\tau_{440} >$**
**0.4 for the fine-mode urban or mixture of aerosols. Boxplot represents lower and upper quartile of**
**observations with mean as white line.**