# Peer review of "Retrieval of UV-Visible aerosol absorption using AERONET and OMI-MODIS synergy: Spatial and temporal variability across major aerosol environments"

_Atmospheric Measurement Techniques, 2021_

## Author Comment (AC1)

RC – Referee comments are in Black
AR – Authors response are in Blue

**Reviewer – 1**

This is a paper that addresses a new approach to the remote sensing (RS) of aerosol absorption by satellite, and with comparison to the same ground-based RS retrievals that are a major input to the algorithm/technique. The use of all AERONET retrieved parameters except for imaginary refractive index is justified for the retrieval of SSA in the UV wavelengths, which is currently not an AERONET product (440 nm is the shortest wavelength in V3). The concept of the paper is somewhat new in scope and has potential to add significantly to the current knowledge on global aerosol absorption. However, there are numerous significant shortcomings to this paper, especially related to the estimated uncertainty of these satellite retrievals of aerosol single scattering albedo (SSA). The uncertainty of the satellite retrievals is somewhat inconsistent in the text and the values in Table 2. You need to clearly state that the uncertainty in SSA from these retrievals due to the complete ensemble of the known sources of error. These known sources of error are uncertainties in all of these factors: AOD, BRDF, aerosol layer height (ALH), size distribution, real refractive index and satellite sensor calibration uncertainty. Full uncertainty calculations and a complete discussion seems to be missing from this paper, and it is essential. Your assumed uncertainty in surface albedo of 0.01 is quite small and is somewhat hard to believe, except in the UV where the reflectivity is quite low for most surface types. References are missing to support this level of surface reflectance uncertainty for the visible wavelengths. Additionally surface albedo is not what is important here but it is BRDF that is a function of both view and solar zenith angles. Additionally, since you use a climatology of size distribution and refractive index from AERONET how much does that affect the retrievals at high and/or low AOD when there are departures from the presumed averages used in your climatological values? See comments below in 'Specific Comments' about the dynamic nature of size distributions as a function of AOD and relative humidity. Also, you make no mention of the magnitude of the uncertainty in satellite sensor calibration and yet this needs to be included in your calculation of overall uncertainty of the satellite retrievals. Your computed uncertainty of +-0.03 in the UV is as good as AERONET at 440 nm (see Sinyuk et al. 2020) and is somewhat hard to believe since for AERONET the effect of uncertainty in surface reflectance and ALH is minimal at AOD(440)>0.4, while the factor of ALH is quite a large of error in the OMI satellite algorithm retrievals in the UV. Table 2 does seem to include AOD uncertainty calculations as +- 0.02 but you fail to discuss this in the text. Also the header of Table 2 only mentions the effects of ALH, so this header needs to be re-written to include the other two factors.

Another problematic aspect of this paper is classification of some sites as biomass burning sites (ex. Arica, Chile) when that this clearly erroneous and other sites as urban aerosol types when they are clearly biomass burning sites (ex. Cuiaba, Brazil). I give details below (in Specific Comments) including several other sites plus some references in the peer-reviewed literature regarding the aerosol types at these AERONET sites.

Additionally, the referencing of published scientific literature is sometimes lacking and sometimes completely erroneous in this manuscript. I provide numerous examples below in my Specific Comments. The authors need to follow normal standards for correct referencing for a journal of high standards such as AMT, and in this manuscript these standards have not been met.

As a result, my opinion is that this manuscript requires extensive revisions and corrections before it may possibly be suitable for publication in AMT.

Specific Comments:

RC: Line 40: It would be nice to quantify here the SSA value corresponding to 50% less UV flux. Also, it seems that you meant 'decrease' instead of 'increase' in this sentence.
AR: Corrected as 'decrease'.

RC: Lines 51-53: It should also be mentioned that most in situ measurement techniques make several assumptions and have significant 'corrections' made to the data to overcome instrumental shortcomings or issues. Therefore the community in general does not consider these data to be a 'gold standard' for validation and there currently is no such gold standard data product available.
AR: We concur with reviewer's comment. The insitu measurements, either taken in lab or measured in the fields, often undergo corrections and subsequent releases. We have added a sentence mentioning instrumental assumptions and corrections.

RC: Lines 60-63: This statement is not entirely accurate. The Hybrid scan has been taken for several years now from newer model Cimel instruments utilized by AERONET. These Hybrid scan retrievals are robust down to 25 degrees solar zenith angle as compared to 50 degrees for the almucantar scans (see Sinyuk et al. 2020). Therefore the high SZA limitation no longer applies. Also, these same instruments have taken 380 nm sky radiance data which is now being analyzed by the AERONET group for a future SSA retrieval at this wavelength. Although this data product is not yet available it should be mentioned that the data exits at many sites for a 380 nm retrieval of SSA, which is an ongoing research topic within the AERONET group. Additionally the recently published paper of Sinyuk et al. (2020) provides uncertainty values of SSA for all AOD levels (figures and tables) so that a data user can assess the level of uncertainty of these retrievals at the 4 retrieval wavelengths for a complete range of AOD for four different aerosol types.
AR: Revised description of the AERONET products. We have now clearly mentioned about the availability of 380 nm sky radiance data and the expected 380 nm SSA in the future upgrades of the AERONET inversion product.

RC: Lines 76-77: SSA is sometimes assumed constant for a region in some MODIS retrievals, but not one constant value globally. You should reference the relevant papers on this regionally varying assumed SSA by Remer et al. 2005 and Levy et al. 2007.
AR: Added a sentence mentioning the use of regionally varying constant SSA in MODIS-DT aerosol products.

RC: Line 90: 'Sahara belt' is an odd terminology. Maybe you mean the Saharan and/or Sahelian regions?
AR: Corrected as 'Saharan' region.

RC: Line 102: This is confusing, as MODIS does not measure at a local noon overpass time. Do you mean 'local time' here instead of 'local noon'?
AR: Corrected as 'local time'.

RC: Line 120: You need to reference Giles et al. (2019) here for the AERONET Version 3 Level 2 AOD data that are cloud screened and QA checked.
AR: Included appropriate reference for AERONET Version 3 Level 2 AOD product.

RC: Line 121: You say nine nominal wavelengths and then you only give 8 wavelengths, omitting 1640 nm.
AR: Corrected for 9 wavelengths.

RC: Line 122: You say the AERONET measurement interval is 15 minutes, but for many instruments the current interval is 5 minutes and has been for a few years. Please convey this in the text.
AR: Corrected for the AERONET measurement time interval.

RC: Line 122: You need to say how you interpolated the spectral AOD measured by AERONET to the OMI measurement wavelengths. A second order polynomial fit of AOD versus wavelength in logarithmic coordinates is much more accurate than using the Angstrom or linear fit (Eck et al., 1999).
AR: Added description on how the AERONET $\tau$ is converted to satellite wavelengths in the revised manuscript (section 3.3.1).

We derive the $\tau$ at the satellite wavelengths using the closest available measurement (within $\pm 2$ hours centered at the satellite overpass time) and $\alpha$ through the power-law approximation (Ångström, 1929) as shown in the below equation. For OMI wavelengths, the AERONET 340 and 380 nm measurements are readily available while $\tau_{354}$ is obtained with $\lambda_{Ref} = 380$ nm and $\alpha_{\lambda\_Ref} = \alpha_{340-440}$. For the few sites with older models of AERONET photometer that does not have direct sun measurements at 340, and 380 nm (for example Banizoumbou, Avignon, etc.), we use $\lambda_{Ref} = 440$ nm and $\alpha_{\lambda\_Ref} = \alpha_{440-675}$. Similarly, for MODIS wavelengths the $\tau$ at 466 and 646 nm are obtained using $\lambda_{Ref} = 440$ nm and $\alpha_{\lambda\_Ref} = \alpha_{440-675}$.

$$\tau_\lambda = \tau_{\lambda_{Ref}} \left(\frac{\lambda}{\lambda_{Ref}}\right)^{-\alpha_{\lambda\_\lambda Ref}}$$

RC: Line 123: Dubovik and King (2000) is the retrieval algorithm paper and not did analyze the AOD measurement accuracy. Therefore, this is a totally inappropriate reference here. The AERONET paper that provides an analysis and estimate of AOD measurement uncertainty is Eck et al. (1999).
AR: Eck et al. (1999) citation for AERONET AOD uncertainty is now included.

RC: Line 128: The Dubovik et al. 1998 paper is a totally inappropriate reference here. Please replace this with Dubovik et al. (2006).
AR: Dubovik et al. (2006) reference added.

RC: Line 129-130: This is a completely obsolete reference for the uncertainty of SSA from AERONET. For Version 3 retrievals there are spectral estimates of SSA uncertainty in Sinyuk et al. (2020) for each of the 4 wavelengths that are also provided as a function of AOD.
AR: Corrected for appropriate reference.

RC: Lines 130-132: This is confusing since below and in Table 1 you say that you use L2 retrievals, yet in this sentence the mention of L1.5 data implies that possibly some of this data may also have been used in this paper. Please clarify this issue in the text.
AR: Corrected for clarity. AERONET Level 1.5 data products are not used in this work.

RC: Lines 135-137: Note that you have an error in Table 1 where you say that for AERONET retrievals Version 2 is used, while in the text you say Version 3 is analyzed.
AR: This is not an error. We have now mentioned it clearly in the text as well. AERONET version 3 level 2 almucantar inversion products were released in late 2017, much latter than we started working on this project. Therefore, for constructing climatology of aerosol particle sizes AERONET version 2 level 2 data is used. Other than that, elsewhere (i.e., AOD inputs for the retrievals and SSA comparison) in our work version 3 level 2 products are used.

RC: Lines 164-167: In section 2.3 MODIS, it seems that at least one reference for MODIS data is warranted in the text.
AR: Appropriate reference is now included.

RC: Line 177: Did you create a size distribution climatology that is a function of AOD? It is well known that fine mode size increases significantly as AOD increases at many sites due to aerosol aging and/or hygroscopic growth. This has been shown in many AERONET papers, see Eck et al. 2012 where the fine radius increases by over 50% over the range of AOD at the GSFC site. Additionally, Eck et al. 2010 presents examples where the fine mode radius doubles over a wide range of fine mode fraction.
AR: We used the entire range of AOD to create seasonal climatology of size distribution. The revised manuscript includes sensitivity tests on our SSA retrieval by perturbing the particle sizes (volume mean radius, VMR) for all aerosol types by 20%. We used 20% perturbation based on examination of the seasonal climatology of particle sizes as a function of AOD and most frequently occurring AOD bin size for over several sites.

RC: Line 211: This sentence is very confusing, please clarify in the text here. Do you average all AERONET measurements of AOD within the +-2 hour interval? Also average all retrievals of size distribution and refractive index within the +-2 hour interval? What specifically do you mean by 'keep it intact'?
AR: Rewritten the sentence for clarity.
- We look for valid AERONET AOD measurements within ±2 hours of satellite overpass and assign the AOD closest in time to all ground pixels. No average of AOD measurements is performed in this work.
- To develop LUT of radiances, we use the entire record of AERONET particle sizes available in the inversion product without any temporal constraint.

RC: Line 221-222: How accurate is this aerosol typing? How many types are there to choose from? Please give a short summary of the typing procedure and it's accuracy or reliability.
AR: In the revised manuscript, aerosol type information is described in section 3.3.2.
"We use a combination of Extinction Ångström Exponent ($\alpha_{440\_675}$) derived from AERONET and near UV Aerosol Index (UVAI) from OMAERUV product to categorize the observed aerosols into three basic types – dust, carbonaceous, and urban/industrial. Initially, our algorithm

uses $\alpha_{440\text{-}675}$ to identify the aerosols as coarse ($\alpha_{440\text{-}675} \leq 0.2$) and fine ($\alpha_{440\text{-}675} \geq 1.2$) mode dominated particles. Threshold $\alpha_{440\text{-}675}$ of 0.2 chosen for coarse mode particles unambiguously identifies dust aerosols. However, the sample of fine mode particles consists of both absorbing carbonaceous and weakly absorbing urban type aerosols. The near-UV aerosol index is an excellent indicator to identify the presence of absorbing aerosols. Threshold UVAI value adopted from OMAERUV algorithm is used to separate carbonaceous (UVAI $\geq$ 0.8) and urban (UVAI < 0.8) aerosols, respectively."

RC: Lines 224-226: Please provide a short description of the accuracy of this aerosol height climatology in the text, as this is a critical factor in the overall uncertainty of the satellite retrievals of SSA in the UV wavelengths.
AR: A brief description of monthly aerosol layer height climatology derived from the joint OMI-CALIOP data set is provided in the section 3.3.2. The expected uncertainty in the ALH from the prescribed OMI-CALIOP data set is within ±1 km (Torres et al., 2013).

RC: Line 237: What is the wavelength of the AOD in the x-axis of Figure 4? This should be added to the figure.
AR: Revised the figure to add $\tau_{440}$ in the x-axis.

RC: Lines 239-241: It should be noted that these are significantly different SSA values than those retrieved by AERONET in the visible wavelengths for the GSFC site. See Giles et al. (2012) that gives SSA at 440 nm = 0.96 and SSA at 675 nm = 0.95. These are VERY large differences (~0.06 to 0.10), especially when it is considered that the SSA parameter range is only ~0.80 to 0.99 for >99% of AERONET retrievals. However you show a large trend as a function of AOD and the lowest AOD bin may be dominating these averages. Why give equal weight to low AOD retrievals that you are aware have very large uncertainty and likely large biases? It is good to see that the Table 3 values for SSA are for AOD(440)>0.4 and that these are much closer to the AERONET retrieval values of the Level 2 database.
AR: The average SSA values reported here is for the entire range of $\tau$. However, we have now replaced it with average SSA for observations with $\tau_{440} > 0.4$ and mentioned it in the main text.

RC: Lines 252-253: This sentence is somewhat confusing and suggests that you may possibly have used different surface reflectances from the MAIAC product that you documented above.
AR: We used MAIAC MCD19A1 spectral BRF or surface reflectance product. Brief description of the product along with the reported uncertainty estimates are provided in the section 3.3.3. "The prescribed measurement-based uncertainty in the MCD19A1 product ranges from 0.002 – 0.003 for visible wavelengths (Lyapustin et al., 2018)".

RC: Line 260: This site name has been mis-spelled, it is Mauna Loa, the Langley calibration site for AERONET. Please note that the AOD is also VERY low at these 3 sites, well below the threshold for accurate retrievals of SSA. Additionally, the Mauna Loa site is on a high mountain (3400 meter elevation for the site) so the AOD from AERONET would not correspond to the aerosol signal of the total atmospheric column as measured by OMI or MODIS. You should have filtered out all sites on mountains for this reason before attempting this analysis.
AR: Revised manuscript now excludes the AERONET sites with very low AODs.

RC: Line 270: Angstrom must always be capitalized as it is the name of a scientist.
AR: Corrected as 'Ångström'.

RC: Line 275: Why not reference the Angstrom 1929 paper, as it is the origin of this parameter.
AR: Included the reference for the Ångström exponent.

RC: Line 282-284: The selection of AE<0.2 for coarse aerosol cases is somewhat extreme as there are relatively few cases of desert dust with AE<0.2 in the AERONET database. Utilizing AE<0.4, which is still coarse mode dominated, would have resulted in many more dust cases to analyze. There are many papers based on in situ measurements that have identified fine mode dust thus the AE of airborne desert dust does not often equal zero or have a negative value which would be the case if there were only coarse mode particles. The 0.2<AE<1.2 bin for mixed mode cases encompasses a very wide range of fine mode fraction of AOD, ~30% to 70% at 500 nm (see Eck et al. 2010).
AR: We agree, the choice of AE ≤ 0.2 for dust aerosols might be extreme. However, allowing the AE up to 0.6 as dust aerosols will only (a) increase in the number of dust samples, and (b) likely introduce 'mixtures' of aerosols in that category – especially in the Sahel region. Therefore, we chose to use the AE ≤ 0.2 to derive unbiased spectral signature of coarse mode dominated dust aerosols.

RC: Lines 287-294: The factors you have identified here (a-d: (a) aerosol extinction measurements, (b) estimation of particle size distribution, (c) real part of refractive index, (d) calibration of satellite measured TOA radiances) however are significant sources of uncertainty in the satellite retrieval of SSA. How much do these factors affect the uncertainty of your SSA retrievals with this algorithm?
AR: The revised manuscript now includes sensitivity tests for the estimated error in SSA retrieval due to all input variables used in the algorithm.

RC: Lines 309-311: How did you arrive at an uncertainty estimate of 0.01 for surface reflectance? This is a very small uncertainty in my opinion since surface reflectance varies seasonally (vegetation phenology), and also as a function of view and solar zenith angle within a day. Additionally, how did you arrive at +-1 km for uncertainty in Aerosol Layer Height?
AR: For the UV wavelengths, we use a surface albedo data set developed from long-term measurements using the minimum Lambertian Equivalent Reflectance (LER) – directly adopted from the operational OMAERUV product. The prescribed uncertainty for this surface albedo data set is expected to be within ±0.01 (Torres et al., 2018). For surface characterization in visible wavelengths, we use MAIAC MCD19A1 BRF product. The prescribed measurement-based uncertainty in the MCD19A1 product ranges from 0.002–0.003 for visible wavelengths (Lyapustin et al., 2018). However, in our sensitivity tests we used a consistent ±0.01 perturbation in surface reflectance for all wavelengths.

To obtain an estimate of ALH required for the retrieval of SSA for both carbonaceous and dust aerosols, we use joint OMI-CALIOP data set. The joint OMI-CALIOP data set uses coincident observations and aerosol index to identify absorbing aerosols and obtain corresponding CALIOP derived layer height from backscattering profiles at 1064 nm. The prescribed uncertainty in the

derived layer height primarily stemming from limited sampling of CALIOP overpasses is expected to be within ±1 km (Torres et al., 2013).

RC: Lines 312-313: Why is the SSA satellite retrieval uncertainty expected to increase with increasing wavelength, just because AOD is less at larger wavelengths or some other reason? If it is related to AOD only then the dust cases would not show a decrease in SSA uncertainty for the longer wavelengths. Please clarify in the text.

AR: Revised the error analysis to include uncertainties due to all input variables in the algorithm. "Among the nine input variables used in our algorithm, the $\Delta\omega_o$ at 340 nm arises mostly from (in descending order) the uncertainties in calibration of TOA radiances, sub-pixel cloud contamination, ALH, particles sizes and so on. While for the visible wavelength at 646 nm, the $\Delta\omega_o$ arises mostly from (in descending order) cloud contamination, surface reflectance, calibration of TOA radiances, particle sizes and so on. These sensitivity tests clearly indicate that $\Delta\omega_o$ is (a) spectrally dependent due to multiple variables, (b) decreases with increasing $\tau$, and (c) varies with absorbing nature of aerosols".

RC: Lines 316-318: "Our analysis shows that for small $\tau440$ (~0.2), the error in retrieved SSA is much higher (> +-0.05) for visible wavelengths, while that in the near-UV region reaches up to +-0.03." Again, is this because the AOD is higher in the UV wavelengths? I am surprised that the ALH effect, which is much greater in the UV, does not counter the AOD wavelength dependence. If it is AOD wavelength dependence that is the main factor then this statement is not true for cases with low values of the Angstrom Exponent.

AR: Please see the above response.

RC: Lines 326-328: The way you have written this section it appears to me that you have only considered the factors of uncertainties in ALH and surface reflectance combined in your estimates of SSA uncertainty. It seems that you have not considered the effects of 0.01 uncertainty in AOD as measured by AERONET and this does not even factor in the spatial variance of AOD over the OMI pixel size. Satellite sensor calibration is also not mentioned in your written description of these estimates of SSA uncertainty. What value of satellite sensor calibration uncertainty did you use? Did you include all of these sources of uncertainty in your calculations but just failed to document them in the text of this paper, or vice versa?

AR: Revised the error analysis to include uncertainties due to all input variables in the algorithm.

RC: Line 344: The aerosols at the Arica site are definitely not biomass burning aerosols as you suggest here. They are dominated by sulfate emissions from copper smelters and therefore non-absorbing. See Eck et al (2012) for a discussion of fine mode size and SSA for this site as follows: "...typical of most retrievals at Arica, where the average SSA is 0.98 for all wavelengths, from nearly 400 retrievals from 1998 to 2000, where AOD (440 nm) >0.4. This is consistent with the principal aerosol sources in the Arica region, as the SO2 emissions from copper smelting create sulfate particles that are non-absorbing"

AR: The AERONET site 'Arica' mentioned in the section 5.1.1 is a typo/mistake. For the regional average SSA presented in figures 6, 7, 8, and 9 the sites names are provided in the upper right corner with abbreviations D (dust), M (mixed), C (carbonaceous), and U (urban) indicating the aerosols type that were averaged to produce the regional spectral SSA. In the revised manuscript these figures are re-produced, and the sites names are mentioned more clearly.

RC: Line 345: Note that Sao Paulo is a major urbanized region (one of the largest on Earth) and therefore the primary aerosol type is urban-industrial, not biomass burning.

AR: Yes, we agree. Please see the above response, in the figure 6 Sau Paulo is denoted with 'U' indicating urban aerosols.

RC: Lines 357-359: Why do you even include a discussion of urban aerosols in this section since this section is titled 5.1 Biomass Burning? Have you averaged the Arica retrieval results into the urban or into the biomass-burning category in this confusing paragraph? Note that the Arica site is neither in an urban region nor is it a biomass-burning site. In my opinion the Arica, Sao Paulo and CEILAP-BA sites should be dropped from this subsection, or you should rename the title of section 5.1.

AR: The section 5 is re-written to avoid confusion.

RC: Lines 360-362: Please note that the higher absorption at the CUIABA sites are due to biomass burning of Cerrado vegetation (similar to wooded savanna) which exhibits more flaming phase combustion therefore more BC than the predominately smoldering combustion at the tropical rain forest sites (see Schafer et al 2008). Your lumping the CUIABA site aerosol type into the urban category of Sao Paulo is erroneous.

AR: In our results we observed both smoke and urban aerosols for the Cuiaba site (please see figure 6 where Cuiaba is denoted with 'C' and 'U'). As mentioned, carbonaceous and urban aerosol types are distinguished using a threshold near-UV AI – synonymous to the aerosol categories in OMAERUV algorithm. Based on extensive tests on the OMI signal strength on all surface types it is determined that a minimum UVAI of 0.8 is required to identify absorbing aerosols (Torres et al., 2007, 2013). Although UVAI is an excellent indicator to identify absorbing aerosols, the large OMI pixel size and sub-pixel contamination of signal strength might at times underestimate the UVAI placing the observed aerosols in urban category. In the revised discussion of carbonaceous aerosols over South America, we have included appropriate references for the high absorption of smoke at Cuiaba.

RC: There are many errors in Table 3 for South America in my opinion. You list the Rio Branco site as urban when it is located in a rural region dominated by biomass burning emissions. The Arica site is missing from Table 3 so why even include it in the averaging in your analysis in this section? You also list a high percentage of retrievals for all of the biomass burning sites in South America as urban which is erroneous. Why do you classify some smoke cases as carbonaceous and some as urban in Table 3? This apparent mis-classification needs to be discussed in the text.

AR: We revised the section 5 (sub-titles) and analysis to include only sites with well know aerosol sources from the literature. However, we chose to keep the results reported in Table 3 (Table S1 in revised version) intact. Please refer to the above response.

It should be noted that our technique to retrieve SSA for carbonaceous and urban aerosols differ only in the use of ALH and uses the same site-specific LUT radiances. It is well known and shown in our sensitivity tests that the effect of ALH is pronounced in UV and gradually reduces towards the visible spectrum. Nonetheless, the results reported in Table 3 provide useful information on aerosol absorption over these sites.

RC: Line 365: Again, it is an odd choice to mix this Pretoria site (urban) in with two rural sites that are dominated by biomass burning aerosols, in a section titled Biomass Burning.
AR: Revised the section 5 (sub-titles) and analysis to include only sites with well know aerosol sources from the literature.

RC: Lines 365-366: There are very few natural forest fires in this southern Africa region. Most biomass burning emissions are from savanna burning initiated by farmers or livestock grazers with minor contribution of crop residue burning. Please refer to Eck et al. (2001 & 2003) for discussion of biomass fuel types in southern Africa.
AR: Added appropriate references for biomass fuel types in southern Africa.

RC: Line 374-375: It seems very doubtful that these relatively high AOD cases in southern Africa are urban aerosol dominated although there may be some mixtures. This urban versus carbonaceous type classification is very confusing and needs to be addressed before these regions are analyzed/introduced in these sections.
AR: Please see above responses regarding the aerosol types of categories based on EAE and UVAI used in this work.

RC: Lines 377-378: Please give the spectral range of AAE here for these values and also note that for the visible to NIR spectral region the AAE of urban or biomass burning types is never as high as 2.2 in the published literature.
AR: In the revised manuscript only carbonaceous aerosols (section 6.1 – revised version) over Southern Africa are reported for Mongu, and Skukuza. Average $AAE_{340-646}$ for these carbonaceous aerosols are 1.7 and 1.6 during JJA and SON. However, as per the aerosol type identification scheme employed here there are few aerosol observations categorized as urban aerosols (Table S1).

For carbonaceous and urban aerosols where black carbon is the sole absorber the AAE values are close to 1 at UV to NIR spectral range. However, aerosols with organic compounds the AAE values varies widely up to 6 (Bergstrom et al., 2007; Kirchstetter et al., 2004 and the references therein) in the UV spectral range, and greater than 1 in the UV-Vis spectra. In addition, it should be noted that AAE values in the literature are derived from linear-fit of power-law expression for any spectral range, while the AAE determined in this work are computed based on the AAOD values at the particular wavelength pairs – these two approaches also lead to some differences.
The mean AAE values reported in this work for carbonaceous (Figure 14) and urban aerosols (Figure 16-in revised version) are consistent with those in the literature.

RC: Lines 384-385: In northern Australia it is not just savanna but also woodland and forests.
AR: Added this information.

RC: Lines 396-397: Please be clear here, are these values AAE. The text should be easier to read without constantly referring to the Figure.
AR: Yes, it is UV-Vis spectral dependence of aerosol absorption ($AAE_{340-646}$).

RC: Line 397: Note that neither of these 2 sites (Jabiru and Lake_Argyle) are strongly influenced by urban aerosols therefore this classification seems erroneous.

AR: The revised text is now limited to carbonaceous aerosols observed over the Jabiru and Lake Argyle sites.

RC: Line 402: Note that the Saada site is adjacent to the city of Marrakesh (~1 million pop.) therefore influenced by urban emissions while the Tamanrasset site is in a rural region in the middle of the Sahara.

AR: Although the site Saada is located adjacent to the city, dominantly coarse-mode dust aerosols are observed during JJA and SON months.

RC: Lines 422-425: This is a completely erroneous reading/interpretation of Eck et al. (2003). There have never been smoke particles in Africa with AE ranging from 0.2 to 0.5 from AERONET measured AOD spectra. You need to eliminate this false statement. Smoke/dust mixtures may have a wide range of AE but these mixtures cannot be called 'smoke particles'. Also note that Eck et al. (2003) reports no AOD measurements or retrievals of smoke properties from West Africa. This paper is focused on Southern Africa biomass burning aerosols, a completely different region. The inaccurate and misleading use of references in this paper is somewhat disturbing. From Eck et al. (2001), from page 3442 of the ZIBBEE paper: "Liousse et al. [1995] found AE (computed for similar wavelengths: 450, 650, and 850 nm AOD) for savanna burning smoke to range from 0.84 for aged smoke to 1.42 for fresh smoke at Lamto, Ivory Coast. However, in that West African location it is possible that these relatively low AE values may be influenced by the presence of Sahelian/Saharan coarse mode dust as a second aerosol type."

AR: We mentioned in the previous line that these aerosols ($0.2 < \alpha_{440-870} < 1.2$) are mixture of dust and carbonaceous aerosols. In the next line, it is a typo/mistake and should have been 'mixture' of aerosols. In the revised manuscript aerosols with mixed mode ($0.2 < \alpha_{440-870} < 1.2$) particles are eliminated.

RC: Lines 429-433: See Eck et al. (2010) for data and discussion of the highly absorbing smoke cases at the Ilorin site, from grassland and treed savanna biomass burning aerosols in the winter season (DJF). Include some comparison in the text.

AR: Added appropriate information with reference.

RC: Line 438: This section should be renamed to "Middle East/North Africa/Arabian Peninsula' since only 1 site out of the 4 sites you have listed is actually geographically located on the Arabian Peninsula (per Wikipedia).

AR: Renamed as 'Middle East'.

RC: Line 459: Classifying the aerosol at Missoula and Rimrock sites in the urban/industrial category (section 5.3 here) is somewhat absurd. These are far from having significant sources of urban/industrial pollution and just about all of the cases with moderate to high AOD (>0.4 at 440 nm) could readily be attributed to biomass burning sources.

AR: Please see above responses regarding the aerosol types of categories based on EAE and UVAI used in this work.

RC: Line 473: No, there is no similarity between the aerosol sources of the sites you analyze in your section 5.3.1 Western North America with the aerosol sources / types in Eastern North

America. As mentioned above 2 of the 3 sites you list for western NA are dominated by biomass burning aerosol sources.

AR: Removed this sentence.

RC: Line 477-478: You should mention that 4 of the 5 sites in this subsection are in a relatively small area, the central mid-Atlantic US (3 in central MD including GSFC), while one site is in the mid-West US (Bondville, Illinois).

AR: The revised analysis now includes regions: central US (Sioux Falls, BONDVILLE), Mideast US (SERC, GSFC, MD_Science_Center) for urban aerosols.

RC: Line 482-484: This statement is not clear, please elaborate here to better explain what you are referring to.

AR: Revised this sentence, to clearly mention that 'for weakly-absorbing aerosols the error in SSA retrievals is high due the identified factors from our analysis such as cloud contamination, surface reflectance, particle sizes and so on'.

RC: Line 487: There are 17 sites that are averaged together here for Europe, much more than for any other region. Some discussion regarding differences between sites is warranted. Also it seems probable that the somewhat noisy looking wavelength dependence of SSA for the SON season is likely due to a small sample size. The number of days of observations should be included in these plots, Figures 6-9, and when the sample size is small and therefore statistically weak it should be noted.

AR: Added some description of the aerosols over the European sites. Since these figures are seasonal averages of several sites over a region it is difficult to indicate the sample size of each site. The readers are advised to refer Table S1 for the sample size.

RC: Line 492-493: However this organic carbon absorption that you suggest is inconsistent with the AAE values in the same plot, as these AAE values which are close to 1 are typical values associated with black carbon absorption. Please discuss this apparent discrepancy and explain why you interpret such an AAE value as being associated with organic carbon absorption. Also include references to support your interpretation.

AR: We agree, AAE values close to 1 are associated with black carbon absorption. Revised sentences(s) consistent with the obtained results.

RC: Line 504: This section is very odd, in that you only include one site and this site is one of the largest urbanized regions in the world (Mexico City) that also happens to be located in Central America. Mid-Atlantic North America, which is your title of this subsection, is Maryland, Virginia etc. Mexico City is equally distant from the Pacific Ocean and Caribbean Sea therefore using mid-Atlantic to describe its location is beyond strange. It seems that the authors are either extremely careless with geographical labeling (or very careless in writing) or need to become more familiar with commonly used regional/geographical names.

AR: Replaced the subtitle Mid-Atlantic North America with Mexico City, Central America.

RC: Line 509-510: "It is clearly evident that such absorption curve and seasonal variation is a result of prevailing mixture of aerosols." This is a confusing sentence. Please elaborate what you

mean here by a mixture of aerosols since elsewhere you define an AE range as mixtures due to fine/coarse particle size mixtures.

AR: In the revised manuscript, the typical spectral SSA expected for the dust, smoke and urban aerosols are clearly described with references before any results are presented. The spectral SSA noted for urban aerosols over the Mexico City deviates from the typical behavior indicating mixture of aerosol components from high vehicular emissions and other sources.

RC: Line 512-514: Please discuss the uncertainty in your computed values of AAE somewhere in this paper. The AAE parameter is highly susceptible to small errors in AOD and SSA and the uncertainty in both AAE and AE increases as the wavelength range decreases due to the resulting small differences in AOD that approach the uncertainty level in the AOD itself. The uncertainty in the AAE for 354 to 388 nm is therefore very high for this reason of relatively small differences in AOD between these two close wavelengths (only 34 nm apart).

AR: Yes, AAE is highly susceptible to small errors in AOD and SSA. We added a section to discuss the uncertainties in the computed AAE by perturbing the SSA at 0.01 intervals.

RC: Line 518: This reference makes no sense as there is no mention of Mexico City in this particular paper (Eck et al. 1998). The use of inappropriate or plain wrong references in this paper is disconcerting at best. The authors need to follow normal standards for correct referencing for a journal of high standards such as AMT.

AR: Removed this reference.

RC: Line 521-522: Please note in the text that for cases of AOD(440)>0.4 as shown in this figure the two sites in China (Beijing and Xianghe, only ~60 km apart) completely dominate the statistics of these 4 site averages. The AOD in these China sites are very high while the AOD levels for the two sites in Japan are much lower such that there are relatively few cases in Japan that exceed the AOD threshold of 0.4. Averaging multiple sites in these types of plots can sometimes be justified but in this case it is very misleading.

AR: The revised analysis now includes regions: northeastern China (Beijing, XiangHe), Japan (Shirahama, Osaka).

RC: Line 522: If you want to label this Figure 9b plot 'Eastern China' then leave the 2 sites in Japan out of the data averaging. In fact, you write this entire section as though the Japanese sites are not included so why did you average these data in with the Chinese sites data at all?

AR: Please see the above response for northeastern China and Japan sites.

RC: Line 525-526: Note that the variation of AAE as a function of fine mode fraction (FMF) for both the Xianghe and Beijing sites are shown in Eck et al. (2010). For low FMF which is equivalent to your dust category, the AAE for both sites was ~2.5 which is consistent with most AAE values in for dust aerosol in the published literature. However your dust value of AAE is much lower than that and therefore you should explain why it is anomalously low for this aerosol type.

AR: From Eck et al., (2010) - Figure 17 c & d, the AAE for coarse particles noted over XiangHe and Beijing are 2.37 and 2.73, respectively. In this work, the AAE noted for dust (EAE ≤ 0.2) over northeastern China is 1.56 for 340-646 spectral range.

It is likely that these coarse particles are influenced by black carbon components over such highly polluted environments. However, due to large particle size (average $\alpha_{440\text{-}870}$ ~0.09) the spectral SSA noted still shows increasing SSA with wavelength. Among the regional dust observations presented in this work, northeastern China exhibit high absorption in visible wavelengths ($\omega_o$ ~0.93 and 0.95 at 466 and 646 nm). Chaudhary et al (2007) reported insitu measurements of coarse mode particles over XiangHe during March-2005 that exhibit high absorption in visible wavelengths ($\omega_o$ ~0.70-0.94 at 450, 550 and 700 nm). Li et al., (2007) explained the variation in SSA for coarse particles during March-2005 over XiangHe is a result of synoptic fluctuation – passage of cold fronts that uplifted ground-level pollution to higher altitudes influencing the aerosol absorption.

Similar low $AAE_{340\text{-}646}$ values for coarse mode ($\alpha_{440\text{-}870} \leq 0.2$, dust) are noted for few sites over the Sahelian region during DJF (burning season). It is likely that these coarse particles over Sahel are influenced by black carbon amounts emitted from biomass burning.

RC: Line 534-535: "The spectral behavior of urban aerosols is similar to carbonaceous aerosols with decrease in magnitude of average SSA, AOD, and UV-Vis AAE." This sentence seems to be somewhat incomplete, as what decrease in these 3 parameters are you referring to? A spectral decrease? Please clarify what you mean here.
AR: Removed this sentence.

RC: Lines 559-561: I have not been convinced that your separation of urban versus carbonaceous aerosol types is robust or has much relation to other classifications in the published literature. There is a lack of discussion of the accuracy of the separation of these two fine mode aerosol types.
AR: The aerosol classification scheme employed in this work uses combined Extinction Angstrom Exponent (EAE) derived from AERONET and near UV aerosol Index (UVAI) derived from OMAERUV product. The EAE and UVAI threshold values for the three aerosol types are: Dust – EAE $\leq$ 0.2, Carbonaceous – EAE $\geq$ 1.2 & UVAI $\geq$ 0.8, and Urban – EAE $\geq$ 1.2 & UVAI < 0.8, respectively. It should be noted that although UVAI is an excellent indicator to identify the presence of absorbing aerosols, due to large OMI footprint and sub-pixel contamination of signal strength might at times underestimate UVAI and categorize the observed aerosols in urban category.

RC: Lines 578-581: However, you inexplicably have averaged the rainforest burning dominated sites with the cerrado vegetation burning dominated site (CUIABA). Note that cerrado vegetation is much like wooded savanna in southern Africa and that is why the SSA values for Cuiaba are lower than for the other South American biomass burning sites, see Schafer et al. 2008. You also have made a distinction between carbonaceous and urban types for these SA biomass burning sites which does not make any sense except for the Sao Paulo site (which has very few retrievals with AOD(440)>0.4). In short Figure 6 is quite misleading since you have mixed apples and oranges so to speak and then average the whole ensemble of sites.
AR: Yes, the regional average spectral SSA for carbonaceous aerosols over South America is derived from observations at Alta Floresta, Cuiaba, Ji Parana and Santa Cruz. Here our purpose is to derive regional average SSA that could be used for guidance in selection of absorption models for satellite aerosol retrievals.

In the revised manuscript, the figures showing regional average SSA are clearly displayed with the sites names to avoid confusion.

RC: Line 585: This would make more sense to add 'the savanna grasses' to 'in the central region' here.
AR: Added 'the savanna grasses'.

RC: Line 588-589: Biomass burning is always a mixture of flaming and smoldering combustion, but when the fraction of flaming combustion increases then the black carbon production increases and the SSA decreases. Please correct this sentence since you imply that this Australian vegetation burns entirely in the flaming phase, which is false.
AR: Corrected as 'biomass burning happens dominantly through flaming phase……'.

RC: Line 595: Again, the uncertainty in AAE for such a narrow wavelength range is much greater than for the 358 to 646 nm range. You should (as a computational exercise) vary the AOD by +0.02 at 354 nm and by -0.02 at 388 nm and see how different the AAE is for this expected AOD uncertainty alone. Add to this the uncertainty in SSA and it may not be very clear if the AAE is really that different in the UV alone from the UV-Visible wavelength range values, or just within the uncertainty error bars.
AR: Uncertainty in AAE for the three wavelength pairs is now included in the revised manuscript.

RC: Lines 597-598: "flaming combustion prevails" is too strong here as there is just a higher fraction of flaming combustion, while smoldering still produces more than half of the smoke aerosol.
AR: Replaced 'flaming combustion prevails' with 'where the contribution of flaming phase combustion is high'.

RC: Line 599: Please note that for the Ilorin site, Eck et al. (2010) found AAE to vary from 1.37 for fine mode dominated to 2.1 for dust dominated as a function of fine mode fraction. Please put your results in that context of variation as a function of Angstrom Exponent or FMF.
AR: We have now mentioned it clearly that the fine mode ($\alpha_{440-870} \geq 1.2$) carbonaceous aerosols found over Ilorin has average $AAE_{340-646}$ 1.38.

RC: Line 603-605: Mixing with coarse mode aerosol is only a part of the explanation of lower AE in these sites. Another factor that you should mention is that the fine mode particle size is larger at these sites due to aging processes of coagulation, condensation, hygroscopic growth and cloud processing.
AR: In the revised manuscript we limited the discussion of spectral SSA in the main text to only coarse-mode dust, fine-mode carbonaceous, and fine-mode urban aerosols. We eliminated the discussion of intermediate-mode particles ($0.2 < \alpha_{440-870} < 1.2$) in the main text and reported these results only in the Table S1 – supplemental material for informational purposes.

RC: Lines 610-612: I would suggest that you change 'often found' to 'sometimes found' since this varies greatly depending on the continent, region and season.
AR: Done.

RC: Lines 612-614: The reference of Torres at al. (2002) for this phenomenon (change in desert dust SSA during transport over the Atlantic from north Africa) is not very robust since the very large TOMS pixel size is very susceptible to partial cloud contamination which would be much more of an issue over the Atlantic Ocean than over the desert source regions and would thus yield higher SSA over the ocean. Additionally, the aerosol height utilized in these retrievals may have introduced significant uncertainty that may also differ for these two regions.

AR: Removed this sentence.

RC: Lines 615-618: Your suggested explanation seems very unlikely to be true. See in Eck et al. (2010) the section on the Ilorin site where the fine mode (biomass burning) aerosols are highly absorbing in the Sahel/Sudanian zones since this is primarily grassland and savanna burning with a relatively high contribution from flaming combustion (more BC produced). The SSA decreased at this site as more fine mode smoke was mixed with the dust. Additionally, some desert dust aerosol sources that advects into the Sahel region are relatively weakly absorbing. The Bodele Depression, which is perhaps the largest single dust source, is an example of weakly absorbing mineral dust since some of the material is diatomaceous sediment, which does not contain iron oxides. See Eck et al. (2010) and Di Biagio et al. (2019) for more information on the SSA of this specific source and other dust sources.

AR: Re-written the sentence for clarity and added appropriate references.

Lines 618-619: These dust SSA values from your retrievals should be compared to the values in the literature such as Di Biagio et al. (2019).

AR: Added appropriate references and comparison of dust SSA.

RC: Line 620: Please include in the Figure 11 caption an explanation of the 5.2 and 6.2 with the X inside the circle symbol.

AR: The circle and the X symbol was used to denote break in the bar chart when the AAE exceeds the maximum value in the y-axis (AAE). The revised version now replaces bar charts showing regional AAE with box-plots showing lower and upper quartile range of observations.

RC: Lines 625-628: It should be noted that this is such a wide range of AAE for dust (1.5 to 3.5) that it could be argued that even with very large retrieval uncertainties you still can fall within this range. Please defend the value in these dust AAE retrievals including estimates of the uncertainties.

AR: The range of mean $AAE_{340-646}$ values reported in the summary section correspond to all regions where dust aerosols ($\alpha_{440-870} \leq 0.2$) are observed in our sample. Excluding northeastern China and Sahel, the mean $AAE_{340-646}$ noted for remaining regions fall within the range 2.2-3.7 (Figure 12 and 15 – revised version). Considering the uncertainties in our AAE estimates these values agree well with those in the literature.

RC: Lines 630-632: Same comment as immediately above: It should be noted that this is such a wide range of AAE for dust in the literature that it could be argued that even with very large retrieval uncertainties you still can fall within this range.

AR: Revised manuscript includes discussion on the uncertainties in the computed AAE.

RC: Line 638: Please include in the Figure 12 caption an explanation of the 5.1 with the X inside the circle symbol.
AR: Revised manuscript now shows the same data as box plot with central 25 and 75-percentile range of observations.

RC: Line 665: Please provide the number of days of data for each site in Figure 13 (for both the AERONET and satellite retrievals) so that the relative statistical robustness may be evaluated.
AR: Revised figure now shows the number days and observations of the data.

RC: Line 666-667: Please note in the text that the uncertainty in the AERONET retrieved SSA is ~0.03 at AOD(440)~0.4 but that this uncertainty decreases for higher AOD levels (see Sinyuk et al. 2020: Fig 22 and Tables 14-17 ).
AR: Added a sentence to mention uncertainty in AERONET SSA decreases with increasing AOD.

RC: Line 667-668: This must be an error, or else you need to change the value of 0.05. Why only include the effect of surface reflectance in your estimate of retrieval uncertainty of SSA from satellite when there are several other significant sources of uncertainty such as AOD, aerosol layer height, aerosol size distribution, refractive index and satellite sensor calibration.
AR: Re-written the sentence to include an ensemble of error in our retrieved SSA due to all input variables.

RC: Line 671-673: However, please note in the text that the overlapping error bars comprise a wide range of +-0.08 which is a large fraction of the expected parameter space for aerosol single scattering albedo (~0.8 to 0.99).
AR: Removed the sentence that says… 'the errors in retrieved and AERONET SSA are consistent as shown with the overlapping error bars'.

RC: Lines 676-678: Please be aware that surface reflectance is a second order source of uncertainty for AERONET since the instrument is upward scanning, and the primary sources of uncertainty for AERONET retrievals of SSA are sky radiance calibration (which is independent of the direct sun cal), solar flux and AOD. For downward viewing satellite retrievals however the surface reflectance is a major source of uncertainty and therefore the way this sentence is written might possibly be quite misleading.
AR: Re-written the sentence(s) to include appropriate information from the complete error analysis.

RC: Lines 679-681: "For absorbing aerosols, the SSA differences observed here is likely a result of different surface reflectances data employed by the two data sets." No your statement here is not true at all. See my comment immediately above.
AR: Please see the above response.

RC: Lines 691-692: It should be noted that the uncertainty in SSA for AERONET is somewhat higher at 675 nm than at 440 nm for fine mode aerosol cases (for a given value of 440 nm AOD; see Sinyuk et al. (2020); Figure 22). This is due to the lower AOD at 675 nm therefore lower

absorption signal in the data. A similar increase in satellite uncertainty at the longer wavelength is inevitable for the same reason.

AR: Yes, we agree. As the absorption signal diminishes at higher wavelengths, surface reflectance contributes much of the upwelling radiances measured by the sensors. Therefore, high uncertainties of satellite retrieved SSA at longer wavelengths is inevitable particularly for weakly absorbing urban aerosols. While with AERONET the uncertainty in SSA primarily due to lower AODs at higher wavelengths.
Re-written the sentence for more clarity in the main text.

RC: Lines 703-705: It should be noted here in the text that the uncertainty in AERONET retrieved SSA decreases as AOD increases (Sunyuk et al. (2020) and that the same is true for the satellite retrievals since at high AOD the aerosol signal overwhelms the sources of uncertainty such as surface BRDF, calibration and AOD.
AR: Please see the above response. Re-written the sentence for more clarity in the main text.

RC: Lines 710-712: This statement is only partially true and needs to be revised. For dust aerosol at low Angstrom Exponent (AE) the AERONET retrieval imposes very weak constraint on the spectral variation of the imaginary refractive index since the AOD is high at all wavelengths and the absorption signal is therefore sufficient at all 4 retrieval wavelengths (440, 675, 9870 and 1020 nm). The sky radiances are fit well at all wavelengths for dust cases and therefore the retrieval is robust at all wavelengths. For fine mode dominated aerosol at high AE values however AERONET version 3 imposes a constraint on the spectral variation of imaginary refractive index. This constraint for high AE retrievals is based on the fact that black carbon exhibits minimal wavelength dependence in imaginary refractive index, plus the fact that for large AE the AOD at the longer wavelengths is quite low and therefore the aerosol absorption signal is insufficient for a robust retrieval at the long wavelengths.
AR: Revised the sentence(s) for more clarity.

RC: Lines 745-746: This statement is misleading as it suggests that the SSA is spectrally flat for the entire wavelength range. However in Table 3 the values for Mongu show significantly higher SSA at 466 nm than at 646 nm.
AR: In the summary of results, we are mentioning the shortest (340 nm) and longest (646 nm) wavelength SSA retrieved in this work. We have re-phrased the sentences for more clarity.

RC: Line 750-752: Again you are exhibiting a tendency to ignore the fact that in your Table 3 the SSA retrievals at 466 nm are significantly higher than at 646 nm for both the Alta Floresta and Missoula sites. Please explain the lack of consistency in your retrieval data versus your interpretations in the text of the paper.
AR: Please see the above response.

RC: Line 758-759: You call the Arabian Peninsula a biomass burning region? Please back up this interpretation with published references and some substantial analysis. Also regions such as eastern China and Northern India do have biomass burning in specific seasons however they are more strongly dominated by other emission sources for more months of the year and therefore are not considered biomass burning regions in the literature (except for specific ~1 month periods).

AR: Re-written the sentence to clearly mention the sites and region (Middle East) we are referring.

RC: Lines 761-762: Why did you only mention 466 nm SSA for this region in this Conclusions section? This section has been inconsistently written.
AR: Corrected for consistency in this section.

RC: Line 774-776: It is somewhat absurd to categorize the CUIABA site as representative of the urban aerosol type. The cases where the AOD at 440 nm exceed 0.4 at Cuiaba are almost always dominated by biomass burning smoke from cerrado vegetation plus some long-range transport of smoke from rain forest burning to the north (see Schafer et al., 2008)
AR: As per the aerosol categorization scheme employed using EAE and UVAI, we noted both carbonaceous and urban aerosols for the site at Cuiaba (see Table S1). We agree that Cuiaba is representative of biomass burning emissions. In the revised discussion (section 6.1) of biomass burning aerosols, Cuiaba is included.

RC: Line 779-781: Please provide some references to back up this interpretation, including discussion of the fuel types in most of Europe you are alluding to.
AR: Removed this sentence. Added appropriate aerosol sources for the sites in Europe.

RC: Line 795: This does not make a lot of sense and needs to be supported with additional references and analysis. The uncertainty of the retrievals at low AOD levels is inherently greater therefore this does not have much basis in rigorous analysis.
AR: Removed this sentence and eliminated the results of SSA at AODs lower than 0.4 in the manuscript.

RC: Lines 806-807: "Given the lack of aerosol absorption information at near-UV wavelengths in the existing AERONET record…" This is not a completely accurate statement. There is currently no AERONET product available on absorption at 380 nm but the sky radiances measurements have been made for years at 380 nm from many instruments in the global network. A retrieval that includes the 380 nm imaginary refractive index and SSA is currently under development by the AERONET project. This statement needs to be revised and expanded to reflect this additional information.
AR: We have now clearly mentioned about the availability of 380 nm sky radiance measurements and expected 380 nm SSA in the future upgrades of AERONET inversion product.

**References**

Ångström, A.: On the atmospheric transmission of sun radiation and on dust in the air, Geogr. Ann., 11, 156–166, doi:10.2307/519399, 1929.

Bergstrom, R. W., Pilewskie, P., Russell, P. B., Redemann, J., Bond, T. C., Quinn, P. K. and Sierau, B.: Spectral absorption properties of atmospheric aerosols, Atmos. Chem. Phys., 7(23), 5937–5943, doi:10.5194/acp-7-5937-2007, 2007.

Chaudhry, Z., Martins, J. V., Li, Z., Tsay, S. C., Chen, H., Wang, P., Wen, T., Li, C. and Dickerson, R. R.: In situ measurements of aerosol mass concentration and radiative properties in

Xianghe, southeast of Beijing, J. Geophys. Res. Atmos., 112(23), doi:10.1029/2007JD009055, 2007.

Dubovik, O., Sinyuk, A., Lapyonok, T., Holben, B. N., Mishchenko, M., Yang, P., Eck, T. F., Volten, H., Muñoz, O., Veihelmann, B., van der Zande, W. J., Leon, J. F., Sorokin, M. and Slutsker, I.: Application of spheroid models to account for aerosol particle nonsphericity in remote sensing of desert dust, J. Geophys. Res. Atmos., 111(D11208), 1–34, doi:10.1029/2005JD006619, 2006.

Eck, T. F., Holben, B. N., Reid, J. S., Dubovik, O., Smirnov, A., O'Neill, N. T., Slutsker, I. and Kinne, S.: Wavelength dependence of the optical depth of biomass burning, urban, and desert dust aerosols, J. Geophys. Res. Atmos., 104(D24), 31333–31349, doi:10.1029/1999JD900923, 1999.

Eck, T. F., Holben, B. N., Sinyuk, A., Pinker, R. T., Goloub, P., Chen, H., Chatenet, B., Li, Z., Singh, R. P., Tripathi, S. N., Reid, J. S., Giles, D. M., Dubovik, O., O'Neill, N. T., Smirnov, A., Wang, P. and Xia, X.: Climatological aspects of the optical properties of fine/coarse mode aerosol mixtures, J. Geophys. Res. Atmos., 115(D19205), 1–20, doi:10.1029/2010JD014002, 2010.

Kirchstetter, T. W., Novakov, T. and Hobbs, P. V.: Evidence that the spectral dependence of light absorption by aerosols is affected by organic carbon, J. Geophys. Res. D Atmos., 109(D21208), 1–12, doi:10.1029/2004JD004999, 2004.

Li, C., Marufu, L. T., Dickerson, R. R., Li, Z., Wen, T., Wang, Y., Wang, P., Chen, H. and Stehr, J. W.: In situ measurements of trace gases and aerosol optical properties at a rural site in northern China during East Asian Study of Tropospheric Aerosols: An International Regional Experiment 2005, J. Geophys. Res., 112(22S04), 1–16, doi:10.1029/2006JD007592, 2007.

Lyapustin, A., Wang, Y., Korkin, S. and Huang, D.: MODIS Collection 6 MAIAC algorithm, Atmos. Meas. Tech., 11(10), 5741–5765, doi:10.5194/amt-11-5741-2018, 2018.

Torres, O., Tanskanen, A., Veihelmann, B., Ahn, C., Braak, R., Bhartia, P. K., Veefkind, P. and Levelt, P.: Aerosols and surface UV products form Ozone Monitoring Instrument observations: An overview, J. Geophys. Res. Atmos., 112(D24), 1–14, doi:10.1029/2007JD008809, 2007.

Torres, O., Ahn, C. and Chen, Z.: Improvements to the OMI near-UV aerosol algorithm using A-train CALIOP and AIRS observations, Atmos. Meas. Tech., 6(11), 3257–3270, doi:10.5194/amt-6-3257-2013, 2013.

Torres, O., Bhartia, P. K., Jethva, H. and Ahn, C.: Impact of the Ozone Monitoring Instrument row anomaly on the long-term record of aerosol products, Atmos. Meas. Tech., 11(5), 2701–2715, doi:10.5194/amt-11-2701-2018, 2018.

---

## Author Comment (AC2)

RC – Referee comments are in Black
AR – Authors response are in Blue

**Reviewer – 2**

RC: This manuscript presents a method for deriving aerosol absorption from a combination of satellite and ground-based remote sensing measurements. The single scattering albedo is derived in five wavelengths in the UV-Visible range. The method can be applied over locations with co-located satellite and ground-based measurements. The method adds UV information on top of the existing data from AERONET. The method is applied to a data record over more than 100 sites globally for the period 2005 – 2016. Results are discussed per region.

Overall, I think this is interesting work, which has a lot of potential. However, the way that it is presented can be much improved. I recommend splitting this manuscript in two parts, where part I describes the method, sensitivity study, case studies and validation, and the part II describes global and regional results. Part I would fit for AMT, whereas part II would better fit in ACP or a similar journal. Part I should answer questions like, what is the added value of this method with respect to the standard AERONET retrievals of SSA? How can the retrieved spectral behavior of the SSA be explained by the expected refractive index? That is why I encourage the authors to withdraw the current manuscript and resubmit it in two improved parts. Because of this recommendation I will not provide detailed textual comments on the current manuscript, but rather indicate where the work needs further improvements.

We sincerely thank the reviewer(s) for providing detailed comments and suggestions wherever applicable that helped make substantial improvement of the manuscript. Major changes made in the revised manuscript are summarized as bullet points here.
- Performed sensitivity tests for all input variables used in our retrieval algorithm, including trace gaseous absorption (in the visible wavelengths) to provide an ensemble of theoretical uncertainty in the retrieved SSA.
- Aerosol type categorization from the combined the used of EAE derived from AERONET and near UV Aerosol Index (UVAI) from OMAERUV product.
- Clarification on the surface reflectance products used in the work and their uncertainty.
- Rewritten the entire section 5 (section 6 in revised version) with focus on the dominant aerosol type(s) prevailing over the worldwide regions.
- Replaced the line/bar plots with box plots wherever applicable to show the range of observations.
- Provided figures showing climatological seasonal aerosol size distributions used for developing LUTs.
- Corrected for appropriate references and other specific comments.

RC: Section 3: Before section 3.1 a text needs to be added that introduces the methods physical background. E.g. what determines the radiation measured at ground-based and satellite level. A diagram would be useful for this.
AR: Added description of the physical basis (section 1) of the satellite remote sensing of aerosols.

RC: Section 3.1: The selected wavelengths in the visible are affected by NO2 and ozone absorption. However, these absorptions are apparently not part of the LUT design. These should be included, or the authors should justify why these absorptions can be neglected. In the end, the method uses reflectances from the satellite instruments and derives the SSA using the LUT. Therefore, the SSA is the depended parameter for the method. For this reason, the axes of Figure 3 should be switched (x-axis reflectance, y-axis SSA). This will immediately visualize the problem for low AOD, where the method will be very noisy.

AR: Revised manuscript now includes estimated error incurred due to unaccounted trace gases in the LUT radiances. Revised the figure 3 as suggested.

RC: Section 3.3: It is unclear where the surface pressure information is coming from.

AR: Included surface pressure information in section 3.3.3. For OMI the surface pressure is directly adopted from OMAERUV product, while for MODIS the terrain height available in the aerosol product is used to convert it to surface pressure as shown in the below equation.

$$P(z) = P_o \exp\left(-\frac{z}{H}\right)$$

Where, $P_o$ is the sea level pressure 1013.25 hPa, and H is scale height of the atmosphere 7.5 km.

RC: Section 4: The sensitivity analysis is incomplete. The approach to only perform a sensitivity analysis for parameters, which are controlled in the retrieval, is clearly not acceptable and also not true, because the real part of the refractive index and other aerosol model parameters are also selected as part of the algorithm. So, all identified parameters should be included in the sensitivity analysis, along with the surface pressure, signal to noise of the instruments, the (tropospheric) ozone column and the NO2 column. This should be presented in graphical way to convince the reader that the method is sound.

AR: Revised manuscript now includes sensitivity tests quantifying the error in SSA due to perturbation in all input variables.

RC: While the analysis of the GSFC method is good, I am left with a number of questions. First of all, the SSA should significantly lower values at 646 nm compared to the other wavelengths. Is this realistic. Is this in line with our knowledge about the refractive index in the visible? Is it possible that this is related to ozone absorption in the visible?

AR: The average SSA values reported here is for the entire range of $\tau$. However, we have now replaced it with average SSA for observations with $\tau_{440} > 0.4$ and changed the x-axis to show AOD400. "The mean aerosol SSA retrieved at the GSFC site for observations with $\tau_{440} > 0.4$ at 340, 354, 388, 466 and 646 nm are 0.94, 0.95, 0.95, 0.94 and 0.93, respectively. These results agrees well with the values reported for GSFC site using AERONET products at 440 and 675 nm as 0.96 and 0.95, respectively (Giles et al., 2012)."

RC: I propose that on top of the GSFC analysis, the authors present 3-6 cases studies of single retrievals over different sites, where for a given day for which both the AERONET SSA and the combined SSA retrievals are available. These cases should cover both good and bad comparisons and discuss the reasons for these results. This gives an opportunity to demonstrate the added value of the satellite method.

AR: We thank the reviewer for encouraging to present more case studies of retrievals over different sites. However, we presented at least 12 sites (4 sites for each representative aerosol types) including GSFC in comparison of SSA section. These sites cover both good and not so

good comparisons showing mean SSA differences up to 0.05. We replaced the line plot to show the central lower and upper quartile range of observations (figure 8 in the revised version). Given the already lengthy manuscript, we decided to use only one site (GSFC) in the section 3.3 to demonstrate our results.

RC: Section 7: Logically, the next section would be the validation (currently section 7). The authors have chosen to compare only the 466 and 646 nm SSA to AERONET. Also comparison between 388 nm retrievals and 440 nm AERONET shall be included. Although the spectral distance is larger than for 466 nm (MODIS), it is the best way of also including the OMI retrievals in the validation. The validation data should also be split by AOD bin. In this way I hope that some correlation can be demonstrated for the medium and high AOD values. Alternatively, times series of data sets over sites with large variability in SSA could be presented to convince the reader that the retrievals add value wrt to the results from AERONET. The current Figures 14 and 15 (top plots) are not very convincing, however the representation of Figure 15 (bottom) is much better.
AR: We have moved the validation section to top, immediately after the error analysis. SSA comparison figures now include data presented as difference in SSA as a function of $\tau_{440}$ for individual aerosol types (carbonaceous, dust, and urban).

RC: Section 5: In section 5 regional results are presented. There are different ways of computing the average SSA. If I understand it correctly, the unweighted average SSA is presented. Alternatively, given the dependence of the accuracy on the AOD, the AOD could be used as weights. This would be equivalent to computing the mean SSA as (1-mean(AAOD)/mean(AOD). Also, mean and standard deviations can be significantly affected by outliers, whereas the median and percentiles are more robust statistics. How would the presented results be affected if other methods of computing statistics are used?
AR: Yes, we presented the unweighted average SSA for all observations with $\tau_{440} > 0.4$ in the regional results (section 6 in revised version). However, examination of the median SSA values (not presented in the manuscript) does not show any significant differences than the unweighted mean values reported here.

RC: The results are presented per region. However, given the poor spatial sampling the statistics will not be representative for the whole region. It is therefore questionable if this analysis is useful at all. For example, there is a huge difference between aerosols on the Californian coast and those of continental Canada, however they are in the region. The same is true Mediterranean sites and sites in Northern Europe as well as other regions. The authors should rethink how these data can be best presented, beyond the current split in regions. I suggest starting with some global maps where the data is plotted per season and aerosol type. Maybe as a circle of which a quarter is used per season or something similar.
AR: We agree that given the poor spatial sampling the regional averages presented here may not be representative for the entire regions. We mentioned it in the main text clearly. The revised manuscript uses appropriate region names than previously mentioned to avoid ambiguity.

RC: Section 5 is very hard to read, as it mixes observations with speculations. Also results from large regions are discussed in terms of very local phenomena. Here a clear choice should be made by authors to either discuss the global distribution of the SSA, or to dive into the details of

one or more regions. Now the scope is somewhere in between and that doesn't work for me. Furthermore, it should be clearly identified when the other claim that the data prove something, or when they speculate about possible explanations.

AR: We have rewritten the entire section 5 for more clarity.

RC: In many cases in Figure 6/7/8/9 a significantly lower SSA at 648 nm is reported as compared to the other wavelengths (e.g. 6a/b, 8 a/b/c, 9 b/c). Do the authors have an explanation of this, in terms of the spectral behavior of the refractive index? Is this reported in other studies? I am not convinced that this is not caused by measurement errors.

AR: It is well known that for other than dust, spectral SSA of aerosols will decrease with increasing wavelength from UV to Visible spectral range. For the results presented in regional analysis the curvature of spectral SSA is consistent with the known behavior. We included comparisons of SSA and derived AAE with insitu measurements and AERONET analysis over the corresponding sites available in the literature. Given the high uncertainties of retrieved SSA in visible wavelengths, the retrieved SSA and AAE compare very well demonstrating the effectiveness of our technique.

RC: Section 6: I am not convinced by the analysis based on the mean AAE. Overall, satellite retrievals of the AE are difficult and the AAE is much more difficult. Also note that AAE is a combination of the spectral behavior of the AOD and that of (1-SSA). Before concluding jumping to conclusions on the AAE, the authors should first provide that there is any value in these mean AAE results.

AR: We agree, satellite-based retrieval of AE and AAE are hard to quantify. In the revised manuscript, we provide sensitivity test by varying the retrieved SSA by the estimated uncertainty. We clearly mentioned in the text that AAE values provided here are for qualitative or informational purposes, as small changes in SSA would incur wide range of AAE.

RC: Tables 3. I propose remove this table to the supplemental material and make it available as complete data set (e.g. HDF5 file(s) or excel files(s)) for all the sites, containing both individual retrievals and statistics.

AR: Table 3 is now moved to supplemental section. We are still in the process of exploring the complete retrieved data set and will look for the opportunity to make it publicly available in a suitable format.

RC: Figure 1. Move this to supplemental material.

AR: We chose to keep this figure in the main text to show all the sites used in our SSA retrieval procedure.

RC: Figure 10-12, replace the bars by violin plots or box-whisker plots.

AR: Revised manuscript has these figures now shown as boxplots with lower and upper quartile range of observations.

RC: Figure 13, replace the plots by box and whisker plots using the spread of the points instead of the assumed uncertainty.

AR: Replaced this figure with boxplots with lower and upper quartile range of observations.

**References**

Giles, D. M., Holben, B. N., Eck, T. F., Sinyuk, A., Smirnov, A., Slutsker, I., Dickerson, R. R., Thompson, A. M. and Schafer, J. S.: An analysis of AERONET aerosol absorption properties and classifications representative of aerosol source regions, J. Geophys. Res. Atmos., 117(D17203), 1–16, doi:10.1029/2012JD018127, 2012.

---

## Author Comment (AC3)

RC – Referee comments are in Black
AR – Authors responses are in Blue

**Reviewer – 3**

RC: This work attempts to use the OMI-MODIS synergetic data and the measured and retrieved products from the AERONET measurements from different Earth regions. This work used UV-Vis spectral band measurements and retrieved the SSA separately for each bands using a LUT approach. Seasonal variability of the SSA in different regions worldwide is studied using the AERONET products and retrieved SSA and AAE from this work. In general, the attempt to quantify the aerosol absorption using the retrieved SSA in UV-Vis multi-year multi-dataset is interesting and useful for the scientific community. However, the methodology and the manuscript need to be improved before considering for publication.

We sincerely thank the reviewer(s) for providing detailed comments and suggestions wherever applicable that helped make substantial improvement of the manuscript. Major changes made in the revised manuscript are summarized as bullet points here.

- Performed sensitivity tests for all input variables used in our retrieval algorithm, including trace gaseous absorption (in the visible wavelengths) to provide an ensemble of theoretical uncertainty in the retrieved SSA.
- Aerosol type categorization from the combined the used of EAE derived from AERONET and near UV Aerosol Index (UVAI) from OMAERUV product.
- Clarification on the surface reflectance products used in the work and their uncertainty.
- Rewritten the entire section 5 (section 6 in revised version) with focus on the dominant aerosol type(s) prevailing over the worldwide regions.
- Replaced the line/bar plots with box plots wherever applicable to show the range of observations.
- Provided figures showing climatological seasonal aerosol size distributions used for developing LUTs.
- Corrected for appropriate references and other specific comments.

RC: AERONET measurement-based climatology is used for representing the particle size distribution in the retrievals. However, no discussion based on it was found in the manuscript. It would be ideal to provide the details of the PSD used in the retrievals. You could create similar plots like the figure 6,7,8, and 9; instead of SSA, you could plot the mean PSD distribution with the error bar as the SD (This can go to the supplemental material).
AR: The revised manuscript now includes figures showing the seasonal climatology of aerosol PSD as supplementary material.

RC: Authors should discuss the criterion for classifying AERONET stations into biomass burning, dust, Urban/Industrial, and mixed aerosol in detail. E.g., the Sao Paulo station classified as biomass burning is wrong since it is a megalopolis.
AR: We used aerosol typing information based on EAE derived from AERONET and UVAI obtained from OMAERUV product. Sao Paulo is not classified as biomass burning site. In the figures 6, 7, 8, and 9 we included the site names on the right upper corner with abbreviations D (dust), M (mixed), C (carbonaceous), U (urban) indicating the aerosol types that were averaged

to derive the regional spectral SSA. For the site at Sao Paulo, the only annotation used is 'U' indicating Urban type aerosols were observed over the site. In the revised manuscript, figures are now replaced, and the sites used in averaging are mentioned more clearly.

RC: Section 5 is not very clear. Please consider rewriting it to avoid ambiguity.
AR: In the revised manuscript, section 5 is re-written, and the results and discussion is limited to specific aerosol types.

RC: Consider moving Table 3 to supplemental material.
AR: Moved the table presenting seasonal SSA and AAE for all the sites to (now Table S1) supplemental material.

RC: A discussion on the correction of atmospheric gas absorption before the aerosols retrievals is needed.
AR: The RTM used in the work accounts for $H_2O$ and $O_3$ absorption. To account for other unaccounted trace gases, we have now included sensitivity tests for the estimated optical depth of all trace gases that has absorption lines in the visible spectrum.

RC: L37: What models are authors discussing here? GCM? Global Earth System models? Please specify it.
AR: General circulation models (GCM).

RC: L55: Did you mean by SSA retrievals?. Because it is clear that there is a long term measurement of AOD at UV bands from AERONET stations.
AR: Yes, we mean lack of SSA retrievals at UV wavelengths in the existing AERONET inversion product.

RC: L71-73: Specifically mention these retrieval algorithms with citation. Are you specifically talking about MODIS operational algorithms?
AR: Added citations for the appropriate retrieval algorithms.

RC: L164: Provide geometry information. Like SZA, VZA, RAA used for simulating TOA observation for the GSFC site.
AR: Each site-specific LUT is developed for several fixed nodes of SZA, VZA, RAA. The figure 3 illustrates the simulated TOA radiances for selected geometry at SZA=20º, VZA=40º, and RAA=130º.

RC: L165-167: This is calculated for a particular satellite-sun geometry, and it can vary considerably in the analysis used in this study. How can you generalize this sensitivity of SSA and AOD to other satellite-sun geometries?
AR: Our purpose here is to demonstrate the well-known theoretical concept of LUT design and 2D-retreival domain of SSA, AOD typical of any satellite retrievals. Depending on the scattering angle, the magnitude of sensitivity for SSA and AOD might vary. In our retrievals, we use the associated satellite-sun geometry for each satellite-AERONET collocated observation.

RC: L177-180: Can you calculate the uncertainty in SSA due to the assumption made here?. It will be important, especially since retrieved SSA differs for regions with different spectral signatures and magnitude.

AR: As mentioned, we used a unified model to create non-spherical LUT of radiances for dust aerosols. This approach is similar to any typical satellite aerosol retrieval where a few models are assumed to retrieve global dust aerosols. In our work, we have not performed any sensitivity test on site- or region-specific dust aerosol particle sizes. This is a computationally extensive work and to avoid a much lengthy manuscript than what we already have we decided to explore this in our future studies.

RC: L184-186: Did you use the closest observation as collocated data?

AR: Re-written the sentence for clarity. We used the AERONET measurement closest in temporal domain to the satellite overpass.

RC: L203: Why the inversion is done independently? You could use the multiple wavelength information in minimization. What is the advantage of doing inversion independently over different spectral bands?. In the discussion section, you are using only the data points with retrievals for all bands. It makes sense to use all the information together to do a retrieval. Please specify the rationale behind not using this method.

AR: Since the UV and visible bands used in this work comes from OMI and MODIS sensors with different spatial and radiometric characteristics, we chose to derive the aerosol absorption independently. While the multi-spectral fit could be another approach to retrieve spectral SSA simultaneously, the minimization scheme often results in error in spectral fit at different wavelengths. Such technique seems to be more suitable for a multi-spectral and/or hyperspectral instruments carrying UV and VIS wavelengths. Nonetheless, an overall good agreement of the derived spectral aerosol absorption (AAE) in our technique with insitu data reported in the literature for different aerosol types demonstrating the effectiveness of the adopted approach.

RC: L212-214: This is not correct for the blue band. The Rayleigh signal will mainly dominate the signal in this band when the aerosol loading is low.

AR: This sentence is re-written to say that Rayleigh scattering gradually diminishes with increasing wavelength proportional to $\lambda^{-4}$.

RC: L270-271: How did you come up with these numbers? Provide references for this. It would be best to use the uncertainty specified by the surface albedo product you are using in the retrieval.

AR: For UV wavelengths, we use a surface albedo data set developed from long-term measurements using the minimum Lambertian Equivalent Reflectance (LER) – directly adopted from the operational OMAERUV product. The prescribed uncertainty for this surface albedo data set is expected to be within ±0.01 (Torres et al., 2018). For surface characterization in visible wavelengths, we use MAIAC MCD19A1 BRF product. The prescribed measurement-based uncertainty in the MCD19A1 product ranges from 0.002–0.003 for visible wavelengths (Lyapustin et al., 2018). However, in our sensitivity tests we used a consistent ±0.01 perturbation in surface reflectance for all wavelengths.

To obtain an estimate of ALH required for the retrieval of SSA for both carbonaceous and dust aerosols, we use joint OMI-CALIOP data set. The joint OMI-CALIOP data set uses coincident observations and aerosol index to identify absorbing aerosols and obtain corresponding CALIOP derived layer height. The prescribed uncertainty in the derived layer height is expected to be within ±1 km (Torres et al., 2013). The uncertainty in aerosol layer height mainly rises from very limited sampling of CALIOP overpasses (16-day repeat cycle).

RC: L282: This achievable accuracy depends on the accuracy of surface reflectance products used in this study. It should be calculated based on the accuracy mentioned for the surface product used and should differ based on surface type, and it will become dominant in the longer wavelengths.
AR: In the revised manuscript, all sensitivity are performed for the most relevant satellite-sun geometry and representative aerosol types observed at GSFC, Mongu, and Tamanrasset sites. To estimate the achievable accuracy in our SSA retrievals, we used a consistent ±0.01 uncertainty in surface reflectance through UV and visible wavelengths. However, it should be noted that prescribed theoretical measurement-based uncertainty in the MAIAC MCD19A1 BRF product is much less, up to 0.002–0.003 at visible wavelengths.

RC: L325-327: I can't see a plot for the DJF season. Is this a typo?
AR: Yes, it is a typo. We corrected this as JJA.

RC: L366-367: This can be verified using the plots of PSD from AERONET retrievals.
AR: We presented the figures showing PSDs as supplementary material. It is evident from the aerosol PSD at Ilorin that both fine-mode and coarse-mode particles are noted during DJF.

RC: L369-370: Is this just the author's opinion? What are the pieces of evidence for this?
AR: Included appropriate reference showing similar result.

RC: L395-396: For the case of JJA, the SSA is increasing with the wavelength for the region 340-388 nm. It is contradicting to the sentence in these lines.
AR: Re-written the sentence for clarity.

RC: L405: Typical urban spectrum based on what work? Cite the literature!
AR: In section 6 (revised version), we discussed the spectral behavior of each representative aerosol type with references before presenting any results.

RC: L422-423: This has to be verified using PSD.
AR: The average aerosol PSD obtained for the European sites show both fine-mode and coarse-mode particles with nearly similar volume concentrations. The $\alpha_{440-870}$ for these aerosols fall in the intermediate-mode category as defined in our work as 0.2 – 1.2.

RC: L518-521: The SSA difference for those two regions can be due to the error from surface reflectance estimation.
AR: Re-written this sentence with references that also shows differences in SSA obtained for dust aerosols over Sahara and Sahel.

RC: L521-522: This is just another hypothesis; there is no proof here.

AR: We have re-written this sentence with references that also shows differences in SSA obtained for dust aerosols over Sahara and Sahel.

RC: L555: What kind of interpolation?

AR: AERONET SSA at 440 and 675 nm are used in linear interpolation to derive SSA for MODIS wavelengths at 466 and 646 nm.

RC: L560: In figure 13, why there is no STD for the UV wavelengths?

AR: There is no STD for UV wavelengths, since we are showing differences in SSA with AERONET at visible wavelengths. In the revised manuscript, we replaced this line plot with box plot to show the lower and upper quartile range of observations.

RC: L561-563: Show the SSA as a box plot with error bars. It will give us an idea of the spread of the SSA values for the averaging AERONET station.

AR: Revised figure now shows the data as box plot with lower and upper quartile range of observations.

RC: L576: Move the annotations to the lower right corner of the plots in Figure 14.

AR: Moved the placement of annotations for visual clarity.

RC: L601: You could cite Dubovik et al., 2006 to describe the smoothness parameter imposed in the retrieval.

AR: Included the appropriate reference for this sentence.

RC: L601-603: Another difference is the use of multi-angular- multispectral information in the AERONET retrieval, Whereas the work presented here used the PSD and real refractive index from those retrieval and basically, the imaginary part of the refractive index is varied to retrieve the SSA.

AR: Yes, thanks for pointing out this difference for us.

RC: L616-618: Define the range of parameters used for this sensitivity study.

AR: Re-written this sentence to include complete set of input variables used in the sensitivity tests and provided an ensemble of error estimated in the retrieved SSA.

RC: L661-662: These two AERONET stations is in the biomass burning aerosols category. Then why you have it here in Urban?

AR: In the figure 6a, we annotated the sites with D, M, C, and U indicating the aerosol types. In our sample for CUIABA both carbonaceous and urban aerosols were observed, while for Sau Paulo only urban aerosols were found. As mentioned earlier, the aerosol-typing scheme employed here uses combined EAE derived from AERONET and near UV Aerosol Index (UVAI) derived from OMAERUV product. The revised manuscript provides new figures clearly indicating the sites used the regional average of spectral SSA.

RC: L687: Specify that no SSA and IRI retrievals available at the moment.

AR: We have included clear statements now mentioning the wavelengths in the existing inversion product and the possibility of AERONET SSA and IRI at 380 nm in the future.

Technical corrections
RC: L40-41: Is it a 50% decrease?
AR: Corrected as 'decrease'.

RC: L59: Citation required for this statement.
AR: Included citation.

RC: L72: 'observations in the visible assume'. It should be 'visible spectrum' instead of just 'visible'.
AR: Corrected as 'visible spectrum'.

RC: L124: In Table 1 it is mentioned that version 2 is used. Which one is correct?
AR: For developing LUT radiances we used version 2 level 2 aerosol particle sizes. Other than that elsewhere (i.e., AOD inputs for the retrievals and SSA comparison) in our work version 3 level 2 AERONET products are used.

RC: L198-199: Did you mean an exponential distribution?. Because the peak is on the surface.
AR: Yes, exponential distribution with a peak on the surface.

RC: L290: Is it $\tau440 \geq 0.2$?
AR: Yes, it should be $\tau_{440} \geq 0.2$. However, the revised manuscript and supplementary materials now includes only observations with $\tau_{440} \geq 0.4$.

RC: L305: The values given for SSA are mean; specify it explicitly with the SD as uncertainty.
AR: We have now provided the mean SSA and SD in the main text discussions.

RC: L441: In figure 9b, instead of 'northeastern china', it is mentioned as 'eastern china.'
AR: Corrected the region as 'northeastern China'.

**References**
Lyapustin, A., Wang, Y., Korkin, S. and Huang, D.: MODIS Collection 6 MAIAC algorithm, Atmos. Meas. Tech., 11(10), 5741–5765, doi:10.5194/amt-11-5741-2018, 2018.

Torres, O., Ahn, C. and Chen, Z.: Improvements to the OMI near-UV aerosol algorithm using A-train CALIOP and AIRS observations, Atmos. Meas. Tech., 6(11), 3257–3270, doi:10.5194/amt-6-3257-2013, 2013.

Torres, O., Bhartia, P. K., Jethva, H. and Ahn, C.: Impact of the Ozone Monitoring Instrument row anomaly on the long-term record of aerosol products, Atmos. Meas. Tech., 11(5), 2701–2715, doi:10.5194/amt-11-2701-2018, 2018.

---

## Author Response (AR2)

RC – Referee comments are in Black
AR – Authors response(s) are in Blue

We sincerely thank the reviewer(s) for providing detailed comments and suggestions wherever applicable that helped make substantial improvement of the manuscript. Major changes made in the second-round review of the manuscript are summarized below.

- Included sensitivity tests of $O_3$ and $NO_2$ gas absorption through UV-Visible spectrum to estimate the error incurred in the retrieval of aerosol SSA.
- Applied corrections for the aerosol SSA to the entire data set due to $NO_2$ gas absorption to obtain the corrected aerosol SSA (Revised all figures and tables).
- Included the estimates of errors in the retrieved SSA due to uncertainty in variability of AOD from satellite observations over the 50-km radius at the AERONET sites.
- Clarification on the aerosol-typing scheme employed in this work. Renamed subtitle 'Biomass burning aerosols' as 'Carbonaceous aerosols' throughout the manuscript.
- Revised sentences wherever applicable in response to other specific comments.

**Review Report – 1**

**General Comments**

The revised version of this manuscript is much improved from the original, with a more complete section on the estimated uncertainty in the SSA retrievals from the new satellite-AERONET algorithm and better and more complete references to the published literature. However, the uncertainty in AOD is underestimated in the calculations of SSA uncertainty since the AERONET point source uncertainty is assumed for the entire 50 km radius area of the input satellite data. Therefore, the authors have assumed exactly homogeneous AOD over a 100 km diameter circle on earth, which is physically unrealistic. The authors should address this issue in a second revision. More details on this issue are given below in my specific comments. Also, a related problem is the computation of the uncertainty in AAE in this revised manuscript. Again, the uncertainty in AOD is assumed to be zero and the authors have only accounted for the effects of uncertainty in SSA on the computation of AAE uncertainty. This is particularly important for the AAE(354-388 nm) since a small error in spectral AOD can cause a large error in AAE for such a narrow wavelength interval. More details are given below in specific comments. This aspect of uncertainty in AAE also needs to be addressed and discussed in a 2nd revision of the manuscript.

Other issues that the authors should address in a revised manuscript are given below in 'Specific Comments":

**Specific Comments:**

Line 144: Please add this after 'almucantar plane': (plus hybrid scans to lower solar zenith angles)
AR: Done.

Line 201: Why not use OMI measurements of ozone or a realistic latitude dependent

climatology of ozone? Did you show that ozone amount does not matter in the retrieval? Did you use NO2 measurements from OMI, or what NO2 amount did you assume in the RTM?

AR: As mentioned in section 3.1, the RT model used in the current work accounts for ozone absorption and assumes a constant ozone concentration of 275 Dobson Unit (DU) for all sites. We have now included errors in retrieved SSA due to ±50 DU ozone amounts based on its variability (not shown here) for all the sites considered in this work. To determine the variability in ozone amounts we use the data provided in AERONET (AOD) product, which is derived from long-term (~25 years) monthly average climatology of the total column ozone retrievals from TOMS data gridded at 1.00 x 1.25 deg spatial resolution.

The RT model employed in this work does not account for $NO_2$ gas absorption. However, we estimate the optical depth of $NO_2$ and applied correction for our retrievals to obtain the corrected aerosol single scattering albedo. The revised manuscript now includes the corrected aerosol SSA after accounting for $NO_2$ gas absorption corrections. Details on the effect of $NO_2$ gas absorption on retrieval of aerosol SSA and correction applied are provided in section 3.3.4.

Line 229-231: This spatial and temporal averaging would certainly increase the difference in AOD between the AERONET point measurements and the AOD that exists in the 50 km radius plus 2-hour difference. Additionally, the delta in AERONET AOD versus the actual AOD in the satellite pixels will increase as a function of increasing AOD since AOD in general becomes less homogeneous in space and time as AOD increases.

AR: The revised manuscript now includes (section 4.1.5) estimate of error incurred in the derived aerosol SSA due to the variability of AOD within the 50 km radius of the point measurement (from AERONET).

Lines 242-243: These Angstrom Exponents are not computed from only two wavelengths as suggested by the authors. These are computed from 3 to 4 wavelengths of AOD with linear fit in logarithmic coordinates. The first three are 3-wavelength values (i.e. 380-500 uses 380, 440 and 500 nm AOD data) while the 440-870 AE uses the 440, 500, 675 and 870 nm AOD data to compute the Angstrom Exponent.

AR: Replaced 'wavelength pairs' as 'wavelength ranges (340-440, 380-550, 440-675, 440-875, etc.)'.

Lines 259-260: It should be noted that urban aerosols have a wide range of absorption; this is not exclusively a weakly absorbing category (see Dubovik et al. 2002 and Giles et al. 2014). Likewise, biomass-burning (or carbonaceous) aerosols exhibit a very wide range of absorption (see Giles et al 2014 and Eck et al. (2003 GRL)), depending largely on the relative contributions of the two phases of combustion (flaming and smoldering). There is extensive overlap in absorption between the two categories of urban and biomass burning. This needs to be discussed in this manuscript as the labels of urban and biomass burning (sometimes called 'carbonaceous' in your paper) often does not make sense in the way your classification system works.

AR: We have now clearly mentioned that fine mode particles with ($\alpha_{440-870} \geq 1.2$) consists of both carbonaceous and urban types of aerosols and exhibit wide range of absorption.

Lines 269-272: Therefore, this is climatology of aerosol layer height and should be clearly stated as such here. Climatological versus actual ALH can vary significantly for any given observation. Was this variation from climatology quantified in the uncertainty of ALH? It would also be expected to vary regionally, as some regions have greater variance in ALH both seasonally and day-to-day.

AR: Yes, we have now clearly mentioned that ALH used here is a global monthly climatology (1 x 1 deg) derived from 30-month long record of OMI-CALIOP collocated data set. Considering the limited samples of CALIOP over 1 x 1 deg gird (16-day overpass cycle) and the day-to-day variation of ALH the uncertainty in the derived monthly (NOT seasonal) ALH climatology is estimated to be within ±1 km.

Lines 286-287: Six-hour surface pressure from NCEP/NCAR reanalysis at 2.5-degree lat-long spatial resolution is interpolated to each AERONET site location and altitude and is provided with the AERONET files of AOD. It would have been much more accurate to have used those values of surface pressure rather than compute it from station altitude.

AR: As described in the section 3.3.3, we use surface pressure provided in the OMAERUV and MODIS aerosol products. These products use high-resolution digital elevation models (≤ 90 m) to compute surface pressure and are provided as ancillary data in the respective products. While OMAERUV product directly provides surface pressure for each ground pixel, MODIS aerosol product provides surface elevation, which are converted to pressure at standard atmospheric conditions.

Thanks for the suggestion on NCEP/NCAR reanalysis six-hour surface pressure data set; we will keep this in our thoughts for the future upgrade of our retrievals.

Line 360: Why stop at AOD(440)=0.4? It would be useful to show estimates at higher AODs also. The uncertainty in SSA retrieval will decrease significantly at higher AOD levels.

AR: Yes, we agree it would be useful to show estimates of higher AODs as well. However, it is well known that retrieval uncertainty decreases with increasing AOD, and our purpose is to determine the minimum AOD where SSA uncertainty is still reasonable within ±0.03-0.05 in the UV-Visible spectrum.

Line 371-373: This is the uncertainty in measured AOD at the AERONET site. However, you have used input satellite data over a 50 km radius and +-2-hour interval from the AERONET site location. Therefore, the variability in AOD over space and time certainty exceeds the point measurement uncertainty at an AERONET site by about a factor of ~50% to 100%. I therefore believe that you have underestimated the uncertainty in your SSA retrievals due your assumption of AOD uncertainty that is not representative of a 100 km diameter satellite average AOD.

AR: The revised manuscript now includes error estimates in our SSA retrievals due to variability in AOD over 50 km radius around the site and ±2 hours of the satellite overpass time (section 4.1.5). To estimate the error in our retrieved SSA due to this assumption we initially estimate the variability in AOD derived from OMAERUV and MODIS-DB AOD products, and ±2 hours of

AERONET AOD from the satellite overpass times. Based on the variability of AOD (not shown here) for the pixels within ±2 hours and 50 km radius of all sites considered we use a perturbation of ±0.2 for λ < 400 nm and ±0.1 for λ > 400 nm to determine the error in our SSA retrievals.

Lines 394-395: It is well known that fine mode particle size increases as AOD increases in many regions due to aging processes of coagulation and condensation (see Dubovik et al. 2002; Eck et al., 2010; Eck et al., 2012). Therefore, your errors due to the use of climatological size distribution averages will be biased as a function of AOD.

AR: We used seasonal climatology of particle sizes to develop LUT radiances. To estimate the error incurred in our SSA retrieval we used perturbation of ΔVMR = 20%. The value of ΔVMR was chosen based on the examination (not shown here) of particle sizes over all sites as a function of AOD and includes the effects of aging processes of coagulation and condensation.

Lines 414-415: Are all pixels assumed to be cloud contaminated or a fixed percentage of pixels assumed to be cloud contaminated in these calculations?

AR: For the sensitivity tests on cloud contamination, we developed LUT for each aerosol type assuming a cloud layer of 0.5 optical thickness in the RT simulations. Therefore, all the pixels are assumed cloud contaminated.

Lines 422-423: I cannot agree with your statement of minimal absorption in the UV from trace gases since NO2 absorption peaks at 380 and 440 nm. Also, NO2 column abundance varies tremendously across the globe and also seasonally. In winter in East Asia (China and South Korea) the NO2 amounts are very high and result in significant absorption in the UV and 440 nm. It appears as though you are basically computing SSA due to aerosols plus NO2 in eastern China and Korea thus overestimating aerosol absorption in these regions. AERONET utilizes a global monthly climatology of NO2 at 0.25-degree resolution derived from OMI data in order to correct the AOD and sky radiances for NO2 absorption effects. This bias in your SSA retrievals, which are maximum in China and South Korea, need to be discussed in the text.

AR: The revised manuscript now includes correction applied for the SSA retrievals to account for $NO_2$ gas absorption (section 3.3.4). We use $NO_2$ concentration provided in the AERONET AOD product (determined from monthly climatology of the total column $NO_2$ retrievals from OMI measurements gridded at 0.25 x 0.25 deg spatial resolution) and absorption coefficients from Vanadele et al 1998 to determine $\tau_{NO2}$. The obtained spectral $\tau_{NO2}$ is used to estimate the actual aerosol SSA (shown in the below equation) as demonstrated by Krotkov et al 2005.

$$\omega_a = \omega(no\ NO_2\ corr).\left[1 + \frac{\tau_{NO}}{\tau_a}\right]$$

Where, $\omega_a$ is the true aerosol SSA,

$\omega$ is the aerosol SSA unaccounted for $NO_2$ absorption,

$\tau_{NO2}$ is the optical depth of columnar $NO_2$ amounts, and

$\tau_a$ is the aerosol optical depth after correcting for Rayleigh, and trace gases including $NO_2$.

Lines 456-457: However, it seems like you ignore this significant component of atmospheric variation in pressure due to meteorology in your calculation of uncertainty.

AR: Figure 1b in Colarco et al 2017 reports the differences in OMAERUV (static) and MERRA-2 (6-hourly) surface pressure. This study suggests the differences in surface pressure employed in the two data sets are mostly found over mountainous terrain (up to ±15 hPa) and oceans (> ±15 hPa). Therefore, the assumed uncertainty of ±100 m terrain height (±12 hPa) in our sensitivity test in the retrieval of aerosol SSA accounts well for these effects.

Lines 477-479: It seems you have neglected a significant source of uncertainty in your computations of AAE uncertainty. The uncertainty in AOD is also a significant factor especially when the wavelengths are close together such as for AAE(354-388 nm). Therefore, your uncertainty estimates can be considered minimum values since it has been assumed that spectral AOD have zero error in your computations.

AR: This is not true. The uncertainty assumed in SSA here corresponds to the overall error in the retrieval of SSA. We have now mentioned it clearly.

Table 3 presents uncertainties in the computation of AAE for ±0.01 intervals of $\Delta\omega_o$. In the main text (section 4.2), only ΔAAE values associated with $\Delta\omega_o$ = ±0.04 are reported which corresponds to the combined error from all variables involved in the retrieval of $\omega_o$ at UV wavelengths as determined in the previous section. From our sensitivity test it is noted that a perturbation of $\Delta\omega_o$ = ±0.04 yields an error in AAE associated with the 354-388 wavelength pair within ±0.13, ±1.3 and ±0.7 for carbonaceous, dust and urban aerosols respectively.

Lines 496-498: It can be expected that this is the more typical situation since the SSA retrievals are independent at each wavelength. Therefore, I would expect the AAE uncertainty to be very large, much higher than your previous analysis that assumes the same bias in SSA in both wavelengths.

AR: For the case where $\Delta\omega_o$ is perturbed only at one of the tail ends of the wavelength pairs the uncertainty in AAE is much higher for 354-388 ranges. This is clearly demonstrated in our results. A small perturbation of $\Delta\omega_o$ = ±0.01 at one tail end of the pair yields an error in AAE up to ±1.2 at which is equivalent to a perturbation of $\Delta\omega_o$ = ±0.04 at both tail ends of the 354-388 wavelength pair.

Line 508: What uncertainty in AAE do you assume here?

AR: For the conversion of our retrieved SSA at 388 nm to 440 nm (matching AERONET wavelength), we use AAE determined from 388-366 wavelength pair. The uncertainty in $AAE_{388-466}$ (not shown here) is relatively less than those found at $AAE_{354-388}$.

Lines 526-527: Any ideas on why the difference is so large for the Lake Argyle site? Small sample size or surface reflectance uncertainty?

AR: Although additional investigation is required, we believe the large difference in SSA is attributed to the surface reflectance uncertainty. The same sample size at other sites produced much better agreement as evident from the figure 9 (revised version).

Lines 529-531: However, you should also note in the text of this manuscript that the surface reflectance is a relatively small source of error in the AERONET retrievals with upward viewing sky radiance measurements.

AR: We have now mentioned clearly that for AERONET SSA surface reflectance is relatively small source of error.

Line 540: The Figure 9 y-axis labels need to be clarified and/or changed, since it is not possible to know what is being plotted without reading the text first. The current y-axis labels just give a wavelength and a satellite name, so it is impossible to interpret by itself.

AR: Revised y-axis title for clarity.

Lines 545-547: This could be partially explained by more sensitivity to the variability in particle size for fine mode aerosols coupled with significant departures on some days from the climatological values used in the retrievals. For dust there is much less sensitivity to particle size.

AR: Thanks for pointing this out. We added a sentence to mention this point.

Line 565: However, the surface reflectance is only a significant source of error in AERONET for low AOD magnitude, ~ <0.2 at 440 nm.

AR: These sentences briefly describe the fundamental difference in AERONET and our SSA retrieval techniques irrespective of the degree of uncertainties of the variables involved.

Line 567: Add this after almucantar plane: (or hybrid scan).

AR: Done.

Line 569 & 571: replace 'weak' with 'relatively strong' in both lines.

AR: Done.

Line 578: I suggest that you keep consistency in your labeling/categorizing of the aerosol type that you often call 'carbonaceous'. Immediately below in section 6.1 you call this type 'biomass burning'. It would be clearer to the reader if you consistently used the term 'biomass burning' throughout the manuscript.

AR: The section title is renamed as 'Carbonaceous aerosols'.

Lines 596-597: September is also a month of significant biomass burning smoke in Missoula, while June typically has a very minor amount of smoke. Please correct this statement.

AR: We have now mentioned fires are common 'June through September'.

Lines 606-607: Should include country for each site name i.e. Brazil and Bolivia in this case.

AR: Done.

Lines 611-612: Are these UV values of SSA for the JJA or SON months? Please clarify this

sentence.
AR: The regional average SSA reported for the JJA, and SON months correspond to 466 nm.

Line 619: Please include the country names: Zambia and South Africa.
AR: Done.

Line 634: It is well known that there is always some dust present in the Sahel and Sudanian zones in the dry season. This is the reason for the relatively low AE of 1.3 since these are mixtures of fine and coarse mode particles. The presence of dust is the reason for the relatively flat spectral SSA at Ilorin. If these were all fine mode biomass burning particles with much black carbon then the SSA would decrease with increasing wavelength.
AR: Revised the sentence to mention these are mixtures of fine and coarse mode particles.

Lines 646-648: Cairo is a very large city, metropolitan area population of 21 million, with many emissions from industry and traffic. It is well known for very high levels of pollution from industry and vehicles. It is not possible to isolate the properties of the aerosol from agricultural burning alone. Mixture of biomass burning plus urban aerosols is inevitable. Please convey this in the text. It is quite odd to even include this site in the Biomass Burning section of this paper.
AR: The section title is renamed as 'Carbonaceous aerosols'. We have now clearly mentioned at the beginning of section 6 that our results do not represent a robust characterization of aerosol types, neither we intend to tag any site as 'biomass burning', 'dust', 'urban' or 'mixed' category. The aerosol typing scheme employed in this work based on UVAI and particle sizes are only to guide our algorithm to include ALH in the SSA retrieval procedure.

We also mentioned in the paragraph that Cairo is one of the megacities with high pollution levels throughout the year and emissions from agricultural waste burning adds additional aerosol burden during September through December.

Line 653: Here is another example of the confusion that your 'carbonaceous' aerosol classification causes. These are dominantly urban/industrial aerosols at Beijing and XiangHe, not predominantly biomass burning aerosols. Your inclusion of this site under the section "6.1 Biomass Burning" is wrong and therefore very misleading.
AR: It is well known that aerosol sources at Beijing and XiangHe are dominantly urban/industrial throughout the year and biomass burning emissions are noted only during a season. Therefore, mixtures of biomass and urban/industrial aerosols are inevitable. We have now mentioned it clearly. The section title is renamed as 'Carbonaceous aerosols'.

Line 655: Do you mean these are AAE values here, if so then clearly state it.
AR: Added 'AAE$_{340-646}$'.

Lines 662-663: Please also mention in the text the other significant aerosol sources in the Indo-Gangetic Plain region in northern India such as brick kilns that burn coal and therefore emit much black carbon, power plants that burn coal and also heavy vehicular traffic in the cities

such as Kanpur. You give the impression in the manuscript that there is only biomass burning going on in the region, which is both false and misleading.

AR: Added sentence to mention other aerosol sources.

Lines 754-756: Also, it is likely that a much smaller sample size in MAM contributes to the difference with JJA since a few unusual cases in spring season may significantly affect the average.

AR: Revised the sentence to mention sample size.

Lines 764-768: Please note that NO2 absorption has not been adequately accounted for in your retrievals in the urban regions where NO2 column amounts are highest. Therefore, it is important that you mention that your retrieved SSA are likely biased low especially at 380 and 440 nm where the NO2 absorption is highest. Maps of NO2 from OMI and TROPOMI clearly show high NO2 amounts over urban regions therefore affecting all of your retrievals for all urban sites, but especially so for China.

AR: Revised manuscript now includes correction for the SSA retrievals to account for the NO2 gas absorption.

Lines 785-786: This probably due to biomass burning aerosols in SON mixing with urban aerosols, not due to a change in the urban aerosol absorption as you seem to be implying. Please rephrase this sentence.

AR: Revised the sentence to mention mixing carbonaceous and urban aerosol samples.

Lines 793-794: The AE=1.3 strongly suggests a mixture of fine and coarse mode aerosols. Even 10-20% of the AOD from coarse mode particles (soil dust, etc) can result in substantial flattening of the SSA spectra. This seems a more likely explanation than your suggestion of a mixture of black carbon and organic carbon, which does not seem to make much sense since both are fine mode particle types.

AR: Revised the sentence to suggest mixture of dust and carbonaceous aerosols.

Lines 796-797: Averaging 13 sites together in Europe over a vast geographic area is not a very rigorous approach. In fact you point out at the end of this paragraph that three of these sites apparently had significantly higher absorption in SON than the other sites. I would suggest some further discussion of the range of SSA values over these 13 sites.

AR: Like the other regions, we added a brief description on the aerosol sources over the sites spread across the Europe. Our results indicate high aerosol absorption during SON for the sites at the Ispra, Modena and Rome located over Northern to Central Italy. It is likely that the increase in aerosol absorption noted during SON is caused by mixture of pollution and carbon amounts from wood burning for domestic heating. However, we do not have retrievals over other sites in the Europe for SON and DJF months to compare the range of values.

Line 805: change 'observing' to absorbing' here.

AR: Corrected as 'highly absorbing'.

Line 806: There is always a mixture of organic and black carbon from fossil fuel combustion, so this sentence essentially tells us nothing.
AR: Revised the sentence.

Lines 812-813: Yes, aerosol humidification is summer results in a large shift in the fine mode particle size to larger particles relative to winter. Since you only apply a yearly mean aerosol size distribution it seems likely that you are underestimating the winter-summer difference in SSA since the larger particles in summer scatter light much more efficiently.
AR: We use seasonal average climatology (NOT yearly mean) of particle sizes for the LUTs as mentioned in section 3.1. Our results indicate wintertime (DJF) aerosols noted over NE China are more absorbing than those noted during summer (JJA) – this is consistent.

Lines 840-841: This is incorrect. The cerrado vegetation type dominates as a source of biomass burning aerosol only at the Cuiaba site, not 'at most sites considered here'.
AR: Corrected the sentence for cerrado vegetation at the Cuiaba site.

Lines 853-854: It should also be mentioned here that the uncertainty of AAE for such a narrow wavelength interval of 354-388 nm is very high. It is higher than your estimated values in Section 3 since you assumed that AOD in both wavelengths was perfect (no error in AOD was assumed).
AR: Revised the sentence to mention that uncertainty in our computation of AAE354-388 is high. As mentioned in the above responses, uncertainty in AAE is estimated using the ensemble of uncertainties in the retrieval of SSA, which includes AOD and several other variables.

Line 861: This can be true for China where the fine mode particle size is very large due to aging and humidification processes (thereby reducing the AE value) but for the Sahel the reason for the AE is mixing with coarse mode dust.
AR: We have revised the sentence, added mixing of dust in Sahel and humidification processes in NE China as examples.

Line 940: I cannot see a valid justification for such a large and polluted megacity as Cairo to be included in the Biomass Burning section. Even when biomass burning occurs near Cairo there is certainly a mix of aerosol types since the urban aerosol sources remain strong producers of aerosol throughout the year.
AR: The section title is renamed now as 'Carbonaceous aerosols'.

Lines 948-950: Again, I find that sites in NE China are well known to be dominated by urban/industrial pollution. Including this Chinese region in the Biomass Burning section is very problematic. Your classification system between biomass burning (that you call carbonaceous half the time) and urban is dubious at best. This ambiguity in these two classifications needs to be discussed in the text so that the reader can be aware of the large overlap between these two aerosol types in your analysis. Even if there is a month or season with biomass burning in

the NE China region the overall aerosol could only be described as mixed since the urban aerosol loading is still very high.

AR: We agree, the sites over NE China are dominated by urban/industrial pollution throughout and biomass-burning emissions are adding an additional aerosol burden only in a season. We clearly mentioned the aerosol typing used here is not robust.

Line 968: Replacing 'carbonaceous' with 'biomass burning' here would be clearer for the reader, especially since you sometimes classify urban aerosol as carbonaceous (see comment above).

AR: Subtitle 'biomass burning' is now replaced with 'carbonaceous aerosols' throughout the manuscript.

Line 973: Please provide urban site names here.

AR: Added site names in the sentence.

Line 975: Also provide site names here.

AR: Added site names in the sentence.

**References**

Krotkov, N. A., Herman, J. R., Cede, A. and Labow, G.: Partitioning between aerosol and NO2 absorption in the UV spectral region, in Ultraviolet Ground- and Space-based Measurements, Models, and Effects V, edited by G. Bernhard, J. R. Slusser, J. R. Herman, and W. Gao, p. 588601., 2005.

Colarco, P. R., Gassó, S., Ahn, C., Buchard, V., Dasilva, A. M. and Torres, O.: Simulation of the Ozone Monitoring Instrument aerosol index using the NASA Goddard Earth Observing System aerosol reanalysis products, Atmos. Meas. Tech., 10(11), 4121–4134, doi:10.5194/amt-10-4121-2017, 2017.

Vandaele, A. C., Hermans, C., Simon, P. C., Carleer, M., Colin, R., Fally, S., Mérienne, M. F., Jenouvrier, A. and Coquart, B.: Measurements of the NO2 absorption cross-section from 42,000 cm-1 to 10,000 cm-1 (238-1000 nm) at 220 K and 294 K, J. Quant. Spectrosc. Radiat. Transf., 59(3–5), 171–184, doi:10.1016/S0022-4073(97)00168-4, 1998.

RC – Referee comments are in Black
AR – Authors response(s) are in Blue

We sincerely thank the reviewer(s) for providing detailed comments and suggestions wherever applicable that helped make substantial improvement of the manuscript.

**Review Report - 2**

I would like to commend the effort made by authors in addressing the comments and restructuring the analysis based on the review of the previous version of manuscript. Overall, the quality of presentation and scientific soundness of the analysis has improved a lot. Before the publication, please make sure that the texts do not mask the data points in the figures.

I have a few minor comments and is given below.
L#78: L_0 is not defined
AR: $L_0$ is atmospheric path radiance. We have now defined in the text.

L#84: Where is the dependence of omega_0 in Eq. 1
AR: In general, the solution of RTE is obtained by expressing the radiance terms as a product of single scattering albedo, phase function and optical depth through (Fourier series of polynomials) cosine function of zenith and azimuth angle of light propagation.

L#408-409: The latest estimation of TOA radiance measurement uncertainties for MODIS on Aqua exists. Just a comment.
AR: Added appropriate reference.

L#455: From the data, it is evident that there exists a spectral dependency, even though the slope is small.
AR: Removed 'spectrally invariant' and revised the sentence accordingly.

L#805: Do you mean predominant? or a typo for highly absorbing.
AR: Corrected as 'highly absorbing'.

**Figures**
F5: Explain the acronyms used in the figure labels in the caption. Also, explain what dashed and solid lines are.
AR: Revised the caption and legend for figure 5.

F6: The legends in the first subplot are blocking the data points in the figure.
AR: Revised figures to show all data points and legend clearly.

F7: Several data points are missing due to the range of y values plotted. Would you mind making sure that data is not masked when you replot the figures for final publication?
AR: Revised figures to show all data points and legend clearly.